# Adversarially Robust Out-of-Distribution Detection Using Lyapunov-Stabilized Embeddings

**Hossein Mirzaei & Mackenzie W. Mathis**
École Polytechnique Fédérale de Lausanne (EPFL)
`hossein.mirzaeisadeghlou@epfl.ch, mackenzie.mathis@epfl.ch`

## Abstract

Despite significant advancements in out-of-distribution (OOD) detection, existing methods still struggle to maintain robustness against adversarial attacks, compromising their reliability in critical real-world applications. Previous studies have attempted to address this challenge by exposing detectors to auxiliary OOD datasets alongside adversarial training. However, the increased data complexity inherent in adversarial training, and the myriad of ways that OOD samples can arise during testing, often prevent these approaches from establishing robust decision boundaries. To address these limitations, we propose AROS, a novel approach leveraging neural ordinary differential equations (NODEs) with Lyapunov stability theorem in order to obtain robust embeddings for OOD detection. By incorporating a tailored loss function, we apply Lyapunov stability theory to ensure that both in-distribution (ID) and OOD data converge to stable equilibrium points within the dynamical system. This approach encourages any perturbed input to return to its stable equilibrium, thereby enhancing the model's robustness against adversarial perturbations. To not use additional data, we generate fake OOD embeddings by sampling from low-likelihood regions of the ID data feature space, approximating the boundaries where OOD data are likely to reside. To then further enhance robustness, we propose the use of an orthogonal binary layer following the stable feature space, which maximizes the separation between the equilibrium points of ID and OOD samples. We validate our method through extensive experiments across several benchmarks, demonstrating superior performance, particularly under adversarial attacks. Notably, our approach improves robust detection performance from 37.8% to **80.1%** on CIFAR-10 vs. CIFAR-100 and from 29.0% to **67.0%** on CIFAR-100 vs. CIFAR-10. Code and pre-trained models are available at `https://github.com/AdaptiveMotorControlLab/AROS`.

## 1 Introduction

Deep neural networks have demonstrated remarkable success in computer vision, achieving significant results across a wide range of tasks. However, these models are vulnerable to adversarial examples — subtly altered inputs that can lead to incorrect predictions (1; 2; 3). As a result, designing a defense mechanism has emerged as a critical task. Various strategies have been proposed, and adversarial training has become one of the most widely adopted approaches (4; 5; 6). Recently, Neural Ordinary Differential Equations (NODEs) have attracted attention as a defense strategy by leveraging principles from control theory. By leveraging the dynamical system properties of NODEs, and imposing stability constraints, these methods aim to enhance robustness with theoretical guarantees. However, they have been predominantly studied in the context of classification tasks (7; 8; 9; 10; 11; 12; 13; 14; 15), and not in out-of-distribution (OOD) detection.

OOD detection is a safety-critical task that is crucial for deploying models in the real world. In this task, training is limited to in-distribution (ID) data, while the inference task involves identifying OOD samples, i.e., samples that deviate from the ID data (16; 17). Recent advancements have demonstrated impressive performance gains across various detection benchmarks (18; 19; 20; 21). However, a significant challenge arises concerning the robustness of OOD detectors against adversarial attacks. An adversarial attack on a detector involves introducing minor perturbations to test samples, causing the detector to predict OOD as ID samples or vice versa. Yet, a robust OOD detector is imperative,

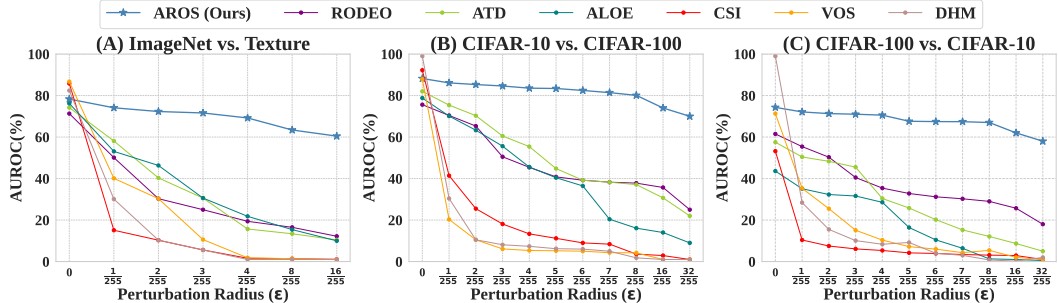

Figure 1: **OOD detection performance for various models under different perturbation magnitudes.** The perturbations are generated using PGD$^{1000}$ ($\ell_\infty$) attack targeting both test ID and OOD samples. (A) ImageNet is used as the ID dataset, while the Texture dataset is used as the OOD during test time. (B) CIFAR-10 is utilized as the ID, with CIFAR-100 as the OOD. (C) CIFAR-100 is used as the ID, with CIFAR-10 as the OOD. A perfect detector achieves an AUROC of 100%, a random detector scores 50%, and a fully compromised detector under attack scores 0%. Notably, no other model achieves detection performance **above random** (i.e., greater than 50% AUROC) at $\epsilon = \frac{8}{255}$.

especially in scenarios like medical diagnostics and autonomous driving (22; 23; 24; 25; 26). Recently, several approaches have sought to address this challenge by first demonstrating that relying solely on ID data is insufficient for building adversarially robust detectors (27; 28; 29; 23; 30; 26; 31; 32; 33; 34; 35; 36; 37; 38). Consequently, new methods propose incorporating copious amounts of auxiliary OOD data in conjunction with adversarial training to improve the detector's robustness. While effective, a significant gap remains between detector performance on clean data and their robustness against adversarial attacks (see Figure 1, Tables 1, 2a, and 2b).

This performance gap primarily arises from the wide variety of potential OOD samples encountered during testing. Relying exclusively on an auxiliary dataset to generate perturbed OOD data can bias the model toward specific OOD instances, thereby compromising the detector's ability to generalize to unseen OOD data during inference (16; 39; 40; 41; 42; 43). This limitation is particularly pronounced in adversarial settings, where adversarial training demands a higher level of data complexity compared to standard training (44; 45; 46; 5; 47). Additionally, the collection of auxiliary OOD data is a costly process, as it must be carefully curated to avoid overlap with ID semantics to ensure that the detector is not confused by data ambiguities (39; 41). Finally, as our empirical analysis reveals, existing OOD detection methods are vulnerable even to non-adversarial perturbations – a concerning issue for open-world applications, where natural factors such as lighting conditions or sensor noise can introduce significant variability (48) (see Table 3).

**Our Contribution:** We propose AROS (**A**dversarially **R**obust **O**OD Detection through **S**tability), a novel approach that leverages NODEs with the Lyapunov stability theorem (Figure 2). This constraint asserts that small perturbations near stable equilibrium points decay over time, allowing the system state to converge back to equilibrium. By ensuring that both ID and OOD data are stable equilibrium points of the detector, the system's dynamics mitigate the effects of perturbations by guiding the state back to its equilibrium. Instead of using extra OOD image data, we craft fake OOD samples in the embedding space by estimating the ID boundary. Additionally, we show that adding an orthogonal binary layer increases the separation between ID and OOD equilibrium points, enhancing robustness. We evaluate AROS under both adversarial and clean setups across various datasets, including large-scale datasets such as ImageNet (49) and real-world medical imaging data (i.e., ADNI (22)), and compare it to previous state-of-the-art methods. Under adversarial scenarios, we apply strong attacks, including PGD$^{1000}$ (44), AutoAttack (50), and Adaptive AutoAttack (51).

## 2 PRELIMINARIES

**Out-of-Distribution Detection.** In an OOD detection setup, it is assumed that there are two sets: an ID dataset and an OOD dataset. We denote the ID dataset as $\mathcal{D}^{\text{in}}$, which consists of pairs $(\mathbf{x}^{\text{in}}, y^{\text{in}})$, where $\mathbf{x}^{\text{in}}$ represents the ID data, and $y^{\text{in}} \in \mathcal{Y}^{\text{in}} := \{1, \ldots, K\}$ denotes the class label. Let $\mathcal{D}^{\text{out}}$ represent the OOD dataset, containing pairs $(\mathbf{x}^{\text{out}}, y^{\text{out}})$, where $y^{\text{out}} \in \mathcal{Y}^{\text{out}} :=$

$\{K+1, \ldots, K+O\}$, and $\mathcal{Y}^{\text{out}} \cap \mathcal{Y}^{\text{in}} = \emptyset$ (52; 18). In practice, different datasets are often used for $\mathcal{D}^{\text{in}}$ and $\mathcal{D}^{\text{out}}$. Alternatively, another scenario is called open-set recognition, where a subset of classes within a dataset is considered as ID, while the remaining classes are considered as OOD (53; 16; 54; 55). A trained model $\mathcal{F}$ assigns an OOD score $S_{\mathcal{F}}$ to each test input, with higher scores indicating a greater likelihood of being OOD.

**Adversarial Attack on OOD Detectors.** Adversarial attacks involve perturbing an input sample $x$ to generate an adversarial example $x^*$ that maximizes the loss function $\ell(x^*; y)$. The perturbation magnitude is constrained by $\epsilon$ to ensure that the alteration remains imperceptible. Formally, the adversarial example is defined as $x^* = \arg\max_{x'} \ell(x'; y)$, subject to $\|x - x^*\|_p \leq \epsilon$, where $p$ denotes the norm (e.g., $p = 2, \infty$) (56; 3; 44). A widely used attack method is Projected Gradient Descent (PGD) (44), which iteratively maximizes the loss by following the gradient sign of $\ell(x^*; y)$ with a step size $\alpha$. For adversarial evaluation (28; 34; 37), we adapt this approach by targeting the OOD score $S_{\mathcal{F}}(x)$. Specifically, the adversarial attack aims to mislead the detector by increasing the OOD score for ID samples and decreasing it for OOD samples, causing misprediction:

$$x_0^* = x, \quad x_{t+1}^* = x_t^* + \alpha \cdot \text{sign}\left(\mathbb{I}(y) \cdot \nabla_x S_{\mathcal{F}}(x_t^*)\right), \quad x^* = x_n^*,$$

where $n$ is the number of steps, and $\mathbb{I}(y) = +1$ if $y \in \mathcal{Y}^{\text{in}}$ and $-1$ if $y \in \mathcal{Y}^{\text{out}}$.

**Neural ODE and Stability.** In the NODE framework, the input and output are treated as two distinct states of a continuous dynamical system, whose evolution is described by trainable layers parameterized by weights $\phi$ and denoted as $h_{\phi}$. The state of the neural ODE, represented by $Z$, evolves over time according to these dynamics, establishing a continuous mapping between the input and output (57; 58; 59). The relationship between the input and output states is governed by the following differential equations: $\quad \frac{dz(t)}{dt} = h_{\phi}(z(t), t), \quad z(0) = z_{\text{input}}, \quad z(T) = z_{\text{output}}.$

## 3 RELATED WORK

**OOD Detection Methods.** Existing OOD detection methods can be broadly categorized into post-hoc and training-based approaches. Post-hoc methods involve training a classifier on ID data and subsequently using statistics from the classifier's outputs or intermediate representations to identify OOD samples. For instance, Hendrycks et al. (52) propose using the maximum softmax probability distributions (MSP) as a metric. The MD method (60) leverages the Mahalanobis Distance in the feature space, and OpenMax (61) recalibrates classification probabilities to improve OOD detection. Training-based methods, modify the training process to enhance OOD detection capabilities. Such modifications can include defining additional loss functions, employing data augmentation techniques, or incorporating auxiliary networks. Examples of training-based methods designed for standard OOD detection include VOS (39), DHM (19), CATEX (62), and CSI (63). On the other hand, ATOM (30), ALOE (28), ATD (34), and RODEO (37) have been developed specifically for robust detection. For detailed descriptions of these methods, please refer to Appendix A1.

**Stable NODE for Robustness.** TiSODE (64) introduces a time-invariant steady NODE to constrain trajectory evolution by keeping the integrand close to zero. Recent works employ Lyapunov stability theory to develop provable safety certificates for neural network systems, particularly in classification tasks. PeerNets (9) was among the first to use control theory and dynamical systems to improve robustness. Kolter et al. (65) designed a Lyapunov function using neural network architectures to stabilize a base dynamics model's equilibrium. ASODE (66) uses non-autonomous NODEs with Lyapunov stability constraints to mitigate adversarial perturbations in slowly time-varying systems. LyaDEQ (67) introduces a new module based on ICNN (68) into its pipeline, leveraging deep equilibrium models and learning a Lyapunov function to enhance stability. SODEF (69) enhances robustness against adversarial attacks by applying regularizers to stabilize the behavior of NODE under the time-invariant assumption. In Table 4a, we analyze these stability-based classifiers as OOD detectors and highlight the potential of Lyapunov's theorem as a framework for robust OOD detection, and show our method's ability to improve performance over these excellent baselines.

## 4 PROPOSED METHOD

**Motivation.** A robust detector should be resistant to shifting ID test samples to OOD, and vice versa, under adversarial attack. A common approach for developing robust OOD detectors involves

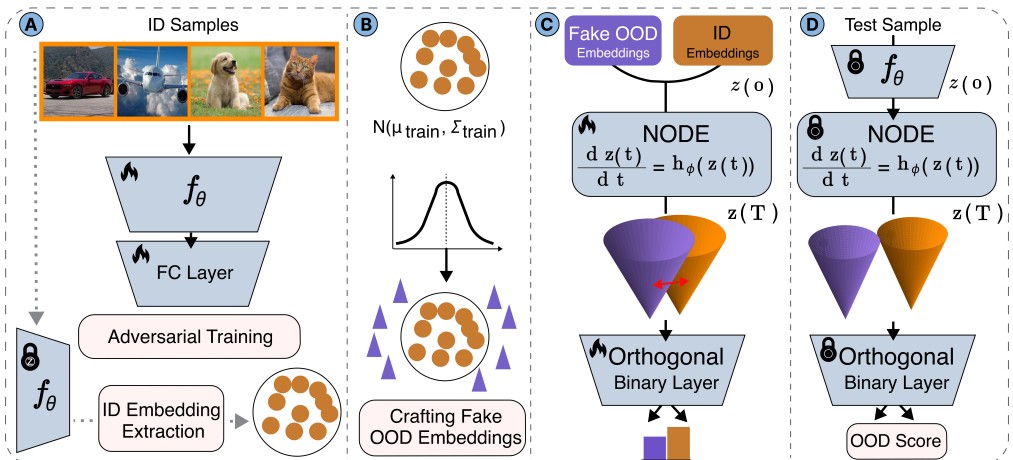

Figure 2: **An illustration of AROS**. **(A)** To obtain robust initial features for OOD detection, we perform adversarial training on a classifier using only ID samples. **(B)** We estimate the ID distribution within the embedding space and generate fake OOD embeddings as a proxy for real OOD data. This enables the creation of two balanced classes of samples: ID and fake OOD. **(C)** The model incorporates a NODE layer $h_\phi$ and an Orthogonal Binary Layer $B_\eta$. Using these two classes, we train the pipeline with the loss function $\mathcal{L}_{\text{SL}}$ to stabilize the system dynamics. **(D)** During inference, an input passes through the feature extractor $f_\theta$, NODE $h_\phi$, and Orthogonal Binary Layer $B_\eta$, and the resulting likelihood from $B_\eta$ serves as the OOD score. The complete algorithmic workflow of AROS can be found in Appendix A2.

employing adversarial training on ID data, combined with an auxiliary real OOD dataset, to expose the detector to potential vulnerable perturbations. The core intuition is that adversarial training on ID data alone, without an accompanying OOD dataset, leaves the detector susceptible to perturbations that alter the boundary between ID and OOD data during testing (28; 29; 30; 23; 34; 26; 31; 33; 36; 35; 37). Beyond the unsatisfactory performance of the prior approach, there are further challenges with this strategy. A key issue is the cost of preparing an auxiliary dataset disjoint from the ID data, along with ensuring that the selected OOD images adequately cover the boundary between ID and OOD samples—a critical factor for such frameworks (70; 37; 30; 24). Moreover, adversarial training of neural networks is notably more data-intensive than standard setups, further increasing complexity (45; 46; 5; 47). There is also the concern that exclusively relying on perturbed OOD data may introduce biases toward specific OOD examples (39; 41). To address these challenges, we propose AROS, which utilizes provable stability theorems in the embedding space to develop a robust OOD detector without requiring exposure to perturbed OOD image data.

**Overview of AROS.** AROS ensures that perturbed input samples remain close to their non-perturbed counterparts in the feature space by leveraging the Lyapunov stability theorem (71; 72; 73; 74). By using a NODE, we consider the model as a dynamical system and design it so that ID and OOD samples converge to distinct stable equilibrium points of that system. This approach prevents significant deviations in the output when adversarial perturbations are applied. However, since OOD data is unavailable, we craft fake OOD samples in the embedding space by estimating the boundaries of the ID distribution and sampling from the corresponding low-likelihood regimes. To further avoid any misprediction between OOD and ID data caused by perturbations, we maximize the distance between their equilibrium points by leveraging an orthogonal binary layer for classification. In the following, we will thoroughly explain each proposed component, highlighting the benefits of AROS.

## 4.1 FAKE EMBEDDING CRAFTING STRATEGY

There have been efforts to utilize synthetic features, primarily under clean scenarios (39; 41; 75; 70). However, for adversarial settings, prior work has often relied on a large pre-trained model and additional data. In contrast, our approach limits information to ID samples, proposing to craft OOD data from ID data in the embedding space. These generated OOD samples are subsequently utilized in the training step.

We employ a well-trained encoder to transform ID training data into robust embedding spaces. To achieve these embeddings, we first adversarially train a classifier on ID training samples using cross-entropy loss $\mathcal{L}_{\text{CE}}$ and the PGD[10]($l_\infty$) attack. By removing the last fully connected layer from the classifier, we utilize the remaining encoder, denoted as $f_\theta$, to extract ID embeddings $r$, where $r = f_\theta(x)$ from an ID training sample $x$ (Figure 2A). Specifically, by considering $\mathcal{D}_{\text{train}}^{\text{in}}$ with $K$ classes, we estimate their distribution as a $K$ class-conditional Gaussian distribution, a well-known approach in the detection literature (39; 76; 77; 78; 79; 80). We then select fake embeddings $r$ from the feature space corresponding to class $j$ such that $r \sim \mathcal{N}(\hat{\mu}_j, \hat{\Sigma}_j)$ satisfies:

$$\frac{1}{(2\pi)^{d/2}|\hat{\Sigma}_j|^{1/2}} \exp\left(-\tfrac{1}{2}(r - \hat{\mu}_j)^T \hat{\Sigma}_j^{-1}(r - \hat{\mu}_j)\right) < \beta, \tag{1}$$

where, $\beta$ serves as a likelihood threshold, and we set that to a small value (e.g., 0.001) (Figure 2B). Additionally, we conduct an ablation study to evaluate the impact of different values of $\beta$ and discuss practical considerations (see Appendix A3.3).Our comprehensive ablation experiments demonstrate the consistent performance of AROS across varying $\beta$ values. Note, $d$ is the dimensionality of the feature vectors $r$, and $j = 1, \ldots, K$. The terms $\hat{\mu}_j$ and $\hat{\Sigma}_j$ represent the mean vector and covariance matrix of the $j$-th class of ID training samples in feature space, respectively:

$$\hat{\mu}_j = \tfrac{1}{n_j} \sum_{i:y_i=j} f_\theta(x_i), \quad \hat{\Sigma}_j = \tfrac{1}{n_j-1} \sum_{i:y_i=j} (f_\theta(x_i) - \hat{\mu}_j)(f_\theta(x_i) - \hat{\mu}_j)^T, \tag{2}$$

where $n_j$ is the number of samples in class $j$. By sampling equally across each class of $\mathcal{D}_{\text{train}}^{\text{in}}$, we generate a set of synthetic, "fake" OOD embeddings (Figure 2C), denoted as $r_{\text{OOD}}$. We then construct a balanced training set by taking the union of the embeddings of ID samples and the OOD embeddings, defining it as: $X_{\text{train}} = \left\{ f_\theta(\mathcal{D}_{\text{train}}^{\text{in}}) \cup r_{\text{OOD}} \right\}$. We define the labels $y$ for this set as 0 for ID and 1 for fake OOD embeddings.

## 4.2 LYAPUNOV STABILITY FOR ROBUST OOD DETECTION

As mentioned, several approaches have been proposed to apply Lyapunov's theorem to deep networks in practice, including methods such as LyaDEQ (67), ASODE (66), and SODEF (69). Here, we utilize their framework to define the objective function and also benchmark our approach to these baselines. Amongst them, SODEF adopts a time-invariant (69; 64) assumption, which makes stability analysis more practical, as the behavior of the neural ODE depends solely on the state $z(t)$, independent of the specific time that the state is reached. This assumption implies that the equilibrium points of the NODE remain constant over time, facilitating a more tractable analysis of how perturbations evolve around these points (64; 81; 69). This is supported by our experiments in Table 4a, which highlight SODEF's superior robustness. Consequently, we adopt the time-invariant framework and use their approach to define the loss function. In order to gain intuition for our approach, we provide the basic mathematical overview of how we leverage the Lyapunov theorems. In this study, as a practical consideration, we assume that the networks utilized have continuous first derivatives with respect to the input $z(0)$, which has been shown to be a reasonable assumption (82).

For a given dynamic system $\frac{dz(t)}{dt} = h_\phi(z(t))$, a state $z^\star$ is an equilibrium point of system if $z^\star$ satisfies $h(z^\star) = 0$. An equilibrium point is stable if the trajectories starting near $z^\star$ remain around it all the time. More formally:

**Definition 1:** (Lyapunov stability (83)). An equilibrium $z^\star$ is said to be stable in the sense of Lyapunov if, for every $\varepsilon > 0$, there exists $\delta > 0$ such that, if $\|z(0) - z^\star\| < \delta$, then $\|z(t) - z^\star\| < \varepsilon$ for all $t \geq 0$. If $z^\star$ is stable, and $\lim_{t\to\infty} \|z(t) - z^\star\| = 0$, $z^\star$ is said to be asymptotically stable.

**Theorem 1:** (Hartman–Grobman Theorem (84)). *Consider a time-invariant system with continuous first derivatives, represented by $\frac{d\mathbf{z}(t)}{dt} = h(\mathbf{z}(t))$. For a fixed point $\mathbf{z}^*$, if the Jacobian matrix $\nabla h$ evaluated at $\mathbf{z}^*$ has no eigenvalues with a real part equal to zero, the behavior of the original nonlinear dynamical system can be analyzed by studying the linearization of the system around this fixed point. The linearized system is given by $\frac{d\mathbf{z}'(t)}{dt} = \mathbf{A}\mathbf{z}'(t)$, where $\mathbf{A}$ is the Jacobian matrix evaluated at $\mathbf{z}^*$. This allows for a simplified analysis of the local dynamics in the vicinity of $\mathbf{z}^*$.*

**Theorem 2:** (Lyapunov Stability Theorem (83)) *The equation $\frac{d\mathbf{z}'(t)}{dt} = \mathbf{A}\mathbf{z}'(t)$, is asymptotically stable if and only if all eigenvalues of $\mathbf{A}$ have negative real parts.*

**Theorem 3:** (Levy–Desplanques Theorem (85)) *Let $A = [a_{ij}]$ be an $n$-dimensional square matrix and suppose it is strictly diagonally dominant, i.e., $|a_{ii}| \geq \sum_{i \neq j} |a_{ij}|$ and $a_{ii} \leq 0$ for all $i$. Then every eigenvalue of $A$ has a negative real part.*

*Definition 1* introduces the concept of asymptotic stability. Building on this, *Theorem 1* demonstrates that the behavior of a nonlinear, time-invariant system near a fixed point can be effectively analyzed through its linearization. *Theorem 2* then establishes a key condition for the asymptotic stability of linear systems: all eigenvalues of the system matrix must have negative real parts. To facilitate the verification of this stability condition, *Theorem 3* provides a practical criterion based on the matrix's eigenvalues. In the subsequent section, we will introduce an objective function designed to adhere to these stability criteria.

### 4.3 ORTHOGONAL BINARY LAYER AND TRAINING STEP

We propose incorporating an orthogonal binary layer (86) denoted as $B_\eta$ after the NODE $h_\phi$ in our pipeline to maximize the distance between the equilibrium points of ID and OOD data. Intuitively, this layer prevents the misalignment of convergence between perturbed OOD data and ID data by maximizing the distance between their equilibrium points. Given the output $z$ from the $h_\phi$, the orthogonal binary layer $B_\eta$ applies a transformation using weights $w$ such that $w^T w = I$, ensuring orthogonality. Although Lyapunov stability encourages perturbed inputs to converge to neighborhoods of their unperturbed counterparts, the infinite-depth nature of NODE (87) makes them susceptible to degraded activations due to exploding or vanishing gradients (88). The introduction of an orthogonal layer mitigates this risk. Moreover, encouraging orthogonality within neural networks has demonstrated multiple benefits, such as preserving gradient norms and enforcing low Lipschitz constants—both of which contribute to enhanced robustness (89; 90; 91).

To satisfy the aforementioned conditions, we optimize the following empirical Lagrangian $\mathcal{L}_{\text{SL}}$ with training data $(X_{\text{train}}, y)$:

$$\mathcal{L}_{\text{SL}} = \min_{\phi, \eta} \frac{1}{|X_{\text{train}}|} \left( \ell_{\text{CE}}(B_\eta(h_\phi(X_{\text{train}})), y) + \gamma_1 \|h_\phi(X_{\text{train}})\|_2 + \gamma_2 \exp\left( -\sum_{i=1}^n [\nabla h_\phi(X_{\text{train}})]_{ii} \right) \right.$$

$$\left. + \gamma_3 \exp\left( \sum_{i=1}^n \left( -|[\nabla h_\phi(X_{\text{train}})]_{ii}| + \sum_{j \neq i} |[\nabla h_\phi(X_{\text{train}})]_{ij}| \right) \right) \right) \tag{3}$$

Note that here, $X_{\text{train}}$ serves as the initial hidden state, i.e., $z(0)$, for the NODE layer. The first term, $\ell_{\text{CE}}$, is a cross-entropy loss function. The second term forces $z(0)$ to be near the equilibrium points, while the remaining terms ensure strictly diagonally dominant derivatives, as described in Theorem 3. The $\exp(.)$ function is selected as a monotonically increasing function with a minimum bound to limit the unbounded influence of the two regularizers, preventing them from dominating the loss. We set $\gamma_1 = 1$ to balance the first regularization term with $\ell_{\text{CE}}$, and $\gamma_2 = \gamma_3 = 0.05$ to assign small, equal values that effectively enforce stability without overpowering the other terms. By setting $\gamma_2$ and $\gamma_3$ equal, we ensure that both stability conditions contribute equally. Details of the ablation study on these hyperparameters, along with other training step specifics, are provided in Appendices A3.3.2 and A4. By optimizing this objective function, the model learns Lyapunov-stable representations where ID and OOD equilibrium points are well-separated in the feature space after the NODE. The $B_\eta$ captures the probability distribution over the binary classes (ID vs. fake OOD), and for the OOD score of an input $x$, we use its probability assigned to the OOD class (Figure 2D).

## 5 EXPERIMENTS

Here we present empirical evidence to validate the effectiveness of our method under various setups, including adversarial attacks, corrupted inputs (non-adversarial perturbations), and clean inputs (non-perturbed scenarios). We note that the backbone architecture for the methods considered is the same as described in Table 1.

First, we adversarially train a classifier on ID data and then use it to map the data into a robust embedding space. A Gaussian distribution is fitted around these embeddings, and low-likelihood regions of the distribution are sampled to create fake OOD data as a proxy for OOD test samples. We

*Table 1.* Performance of OOD detection methods under clean evaluation, random corruption (Gaussian noise), and PGD ($l_\infty$) adversarial attack with 1000 steps and $\frac{8}{255}$, as well as AutoAttack and Adaptive AutoAttack (AA), measured by AUROC (%). A clean evaluation is one where no attack is made on the data. For corruption evaluation, Gaussian noise from the ImageNet-C (48) benchmark was used. The best results are highlighted in **bold**, and the second-best results are underlined in each row.

[†] These methods leveraged auxiliary datasets and these [*] used large pretrained models as part of their pipeline.

| Dataset | | Attack | Method | | | | | | | | |
|---|---|---|---|---|---|---|---|---|---|---|---|
| $\mathcal{D}_{in}$ | $\mathcal{D}_{out}$ | | VOS (ResNet) | DHM (WideResNet) | CATEX[*] (CLIP-ViT) | CSI (ResNet) | ATOM[†] (DenseNet) | ALOE[†] (WideResNet) | ATD[†*] (WideResNet) | RODEO[†*] (CLIP-ViT) | AROS (WideResNet) |
| CIFAR10 | CIFAR100 | Clean | 87.9 | **100.0** | 88.3 | 92.2 | 94.2 | 78.8 | 82.0 | 75.6 | 88.2 |
| | | Corruption | 56.2 | 57.7 | 60.4 | 54.7 | 57.3 | 54.5 | 59.2 | 58.6 | **84.3** |
| | | PGD$^{1000}$ | 4.2 | 1.8 | 0.8 | 3.6 | 1.6 | 16.1 | 37.1 | 37.8 | **80.1** |
| | | AutoAttack | 0.0 | 1.2 | 0.0 | 0.4 | 0.5 | 14.8 | 36.2 | 35.9 | **78.9** |
| | | AdaptiveAA | 0.0 | 0.0 | 1.7 | 0.0 | 0.0 | 11.5 | 34.8 | 32.3 | **76.4** |
| CIFAR100 | CIFAR10 | Clean | 71.3 | **100.0** | 85.1 | 53.2 | 87.5 | 43.6 | 57.5 | 61.5 | 74.3 |
| | | Corruption | 53.8 | 58.2 | 57.4 | 50.1 | 55.3 | 56.1 | 56.0 | 54.9 | **71.8** |
| | | PGD$^{1000}$ | 5.4 | 0.0 | 4.0 | 2.8 | 2.0 | 1.3 | 12.1 | 29.0 | **67.0** |
| | | AutoAttack | 2.6 | 0.0 | 0.3 | 0.9 | 0.0 | 0.0 | 10.5 | 28.3 | **66.5** |
| | | AdaptiveAA | 0.0 | 1.4 | 0.0 | 0.0 | 0.0 | 0.2 | 9.4 | 26.7 | **65.2** |

then demonstrate that time invariance, which establishes that the NODE's behavior does not explicitly depend on time, leads to more stable behavior of the detector under adversarial attacks (see Section 5). Consequently, we leverage Lyapunov stability regularization under a time-invariant assumption for training. However, a potential challenge arises when ID and OOD equilibrium points are located near each other. As a remedy, we introduce an orthogonal binary layer (86; 92; 93) that enhances the separation between ID and OOD data by increasing the distance between their neighborhoods of Lyapunov-stable equilibrium. This enhances the model's robustness against shifting adversarial samples from OOD to ID and vice versa. Finally, we use the orthogonal binary layer's confidence output as the OOD score during inference.

**Experimental Setup & Datasets.** We evaluated OOD detection methods under both adversarial and clean scenarios (see Tables 1 and 2a). Each experiment utilized two disjoint datasets: one as the ID dataset and the other as the OOD test set. For Table 1, CIFAR-10 or CIFAR-100 (94) served as the ID. Table 2a extends the evaluation to ImageNet-1k as the ID, with OOD being comprised of Texture (95), SVHN (96), iNaturalist (97), Places365 (98), LSUN (99), and iSUN (100).

An OSR (101) setup was also tested, in which each experiment involved a single dataset that was randomly split into ID (60%) and OOD (40%) subclasses, with results averaged over 10 trials. Datasets used for OSR included CIFAR-10, CIFAR-100, ImageNet-1k, MNIST (102), FMNIST (103), and Imagenette (104) (Table 2b). Additionally, models were evaluated on corrupted data using the CIFAR-10-C and CIFAR-100-C benchmarks (48) (see Table 3). Specifically, both the ID and OOD data were perturbed with corruptions that did not alter semantics but introduced slight distributional shifts during testing. Further details on the datasets are provided in Appendix A5.

**Evaluation Details.** For adversarial evaluation, all ID and OOD test data were perturbed by using a fully end-to-end PGD ($l_\infty$) attack targeting their OOD scores (as described in Section 2). We used $\epsilon = \frac{8}{255}$ for low-resolution images and $\epsilon = \frac{4}{255}$ for high-resolution images. The PGD attack steps denoted as $M$ were set to 1000, with 10 random initializations sampled from the interval $(-\epsilon, \epsilon)$. The step size for the attack was set to $\alpha = 2.5 \times \frac{\epsilon}{M}$ (4). Additionally, we considered AutoAttack and Adaptive AutoAttack (Table 1). Details on how these attacks are tailored for the detection task can be found in Appendix A4. As the primary evaluation metric, we used AUROC, representing the area under the receiver operating characteristic curve. Additionally, we used AUPR and FPR95 as supplementary metrics, with results presented in Table 4b. AUPR represents the area under the precision-recall curve, while FPR95 measures the false positive rate when the model correctly identifies 95% of the true positives.

**Reported and Re-Evaluated Results.** Some methods may show different results here compared to those reported in their original papers (30; 28) due to our use of stronger attacks we incorporated

*Table 2a.* Performance of OOD detection methods under clean evaluation and $\text{PGD}^{1000}(l_\infty)$ measured by AUROC (%). The perturbation budget $\epsilon$ is set to $\frac{8}{255}$ for low-resolution datasets and $\frac{4}{255}$ for high-resolution datasets. The table cells denote results in the 'Clean/$\text{PGD}^{1000}$' format.

| Dataset | | Method | | | | | | | | |
|---|---|---|---|---|---|---|---|---|---|---|
| $\mathcal{D}_{in}$ | $\mathcal{D}_{out}$ | VOS | DHM | CATEX | CSI | ATOM | ALOE | ATD | RODEO | AROS (Ours) |
| **CIFAR-10** | **CIFAR-100** | 87.9/4.2 | **100.0**/1.8 | 88.3/0.8 | 92.2/3.6 | 94.2/1.6 | 78.8/16.1 | 82.0/37.1 | 75.6/37.8 | 88.2/**80.1** |
| | **SVHN** | 93.3/2.8 | **100.0**/4.5 | 91.6/2.3 | 97.4/1.7 | 89.2/4.7 | 83.5/26.6 | 87.9/39.0 | 83.0/38.2 | 93.0/**86.4** |
| | **Places** | 89.7/5.2 | **99.6**/0.0 | 90.4/4.7 | 93.6/0.1 | 98.7/5.6 | 85.1/21.9 | 92.5/59.8 | 96.2/70.2 | 90.8/**83.5** |
| | **LSUN** | 98.0/7.3 | **100.0**/2.6 | 95.1/0.8 | 97.7/0.0 | 99.1/1.0 | 98.7/50.7 | 96.0/68.1 | 99.0/**85.1** | 90.6/82.4 |
| | **iSUN** | 94.6/0.5 | 99.1/2.8 | 93.2/4.4 | 95.4/3.6 | **99.5**/2.5 | 98.3/49.5 | 94.8/65.9 | 97.7/78.7 | 88.9/**81.2** |
| **CIFAR-100** | **CIFAR-10** | 71.3/5.4 | **100.0**/2.6 | 85.1/4.0 | 53.2/0.7 | 87.5/2.0 | 43.6/1.3 | 57.5/12.1 | 61.5/29.0 | 74.3/**67.0** |
| | **SVHN** | 92.6/3.2 | **100.0**/0.8 | 94.6/5.7 | 90.5/4.2 | 92.8/5.3 | 74.0/18.1 | 72.5/27.6 | 76.9/31.4 | 81.5/**70.6** |
| | **Places** | 75.5/0.0 | **100.0**/3.9 | 87.3/1.4 | 73.6/0.0 | 94.8/3.0 | 75.0/12.4 | 83.3/40.0 | 93.0/66.6 | 77.0/**69.2** |
| | **LSUN** | 92.9/5.7 | **100.0**/1.6 | 94.0/8.9 | 63.4/1.8 | 96.6/1.5 | 98.7/50.7 | 96.0/68.1 | 98.1/63.1 | 74.3/**68.1** |
| | **iSUN** | 70.2/4.5 | 99.6/3.6 | 81.2/0.0 | 81.4/3.0 | 96.4/1.4 | **98.3**/49.5 | 94.8/65.9 | 95.1/65.6 | 72.8/**67.9** |
| **ImageNet-1k** | **Texture** | 86.7/0.8 | 82.4/0.0 | 92.7/0.0 | 85.8/0.6 | **88.9**/7.3 | 76.2/21.8 | 74.2/15.7 | 71.3/19.4 | 78.3/**69.2** |
| | **iNaturalist** | 94.5/0.0 | 80.7/0.0 | 97.9/2.0 | 85.2/1.7 | 83.6/10.5 | 78.9/19.4 | 72.5/12.6 | 72.7/15.0 | 84.6/**75.3** |
| | **Places** | 90.2/0.0 | 76.2/0.4 | 90.5/0.0 | 83.9/0.2 | 84.5/12.8 | 78.6/15.3 | 75.4/17.5 | 69.2/18.5 | 76.2/**68.1** |
| | **LSUN** | 91.9/0.0 | 82.5/0.0 | 92.9/0.4 | 78.4/1.9 | 85.3/11.2 | 77.4/16.9 | 68.3/15.1 | 70.4/16.2 | 79.4/**69.0** |
| | **iSUN** | 92.8/2.7 | 81.6/0.0 | 93.7/0.0 | 77.5/0.0 | 80.3/14.1 | 75.3/11.8 | 76.6/15.8 | 72.8/17.3 | 80.3/**71.6** |
| | *Mean* | 88.1/2.8 | **93.4**/1.6 | 91.2/2.3 | 83.3/1.5 | 91.4/5.6 | 81.4/25.5 | 81.6/37.4 | 82.1/44.4 | 82.0/**74.0** |

*Table 2b.* Performance (Clean/$\text{PGD}^{1000}$) of OOD detection methods under clean and $\text{PGD}^{1000}(l_\infty)$, measured by AUROC (%), on the OSR setup, which splits one dataset's classes randomly to create $\mathcal{D}_{in}$ and $\mathcal{D}_{out}$.

| Dataset | Method | | | | | | | | |
|---|---|---|---|---|---|---|---|---|---|
| | VOS | DHM | CATEX | CSI | ATOM | ALOE | ATD | RODEO | AROS (Ours) |
| **MNIST** | 86.3/4.8 | 92.6/0.4 | 92.3/1.9 | 93.6/6.1 | 74.8/4.1 | 79.5/37.3 | 68.7/56.5 | **97.2**/85.0 | 94.4/**86.3** |
| **FMNIST** | 78.1/2.0 | 85.9/0.0 | 87.0/0.4 | 84.6/1.2 | 64.3/4.2 | 72.6/28.5 | 59.6/42.1 | **87.7**/65.3 | 84.1/**72.6** |
| **CIFAR-10** | 74.7/0.0 | 90.8/0.0 | **95.1**/0.0 | 91.4/0.6 | 68.3/5.0 | 52.4/25.6 | 49.0/32.4 | 79.6/62.7 | 78.8/**69.5** |
| **CIFAR-100** | 63.5/0.0 | 78.6/0.0 | **91.9**/0.0 | 86.7/1.9 | 51.4/2.6 | 49.8/18.2 | 50.5/36.1 | 64.1/35.3 | 67.0/**58.2** |
| **Imagenette** | 76.7/0.0 | 84.2/0.0 | **96.4**/1.6 | 92.8/0.0 | 63.5/8.2 | 61.7/14.2 | 63.8/28.4 | 70.6/39.4 | 78.2/**67.5** |
| **ADNI** | 73.5/4.1 | 69.4/5.2 | **86.9**/0.1 | 82.1/0.0 | 66.9/2.3 | 64.0/11.0 | 68.3/33.9 | 75.5/24.6 | 80.9/**61.7** |
| *Mean* | 75.5/1.8 | 83.6/0.9 | **91.6**/0.7 | 88.5/1.6 | 64.9/4.4 | 63.3/22.5 | 60.0/38.2 | 79.1/52.1 | 80.6/**69.3** |

for evaluation, or the more challenging benchmarks used. For example, ALOE (28) considered a lower perturbation budget for evaluation (i.e., $\frac{1}{255}$), and the ATD (34) and RODEO (37) benchmarks used CIFAR-10 vs. a union of several datasets, rather than CIFAR-10 vs. CIFAR-100. The union set included datasets such as MNIST, which is significantly different from CIFAR-10, leading to a higher reported robust performance.

**Results Analysis.** Without relying on additional datasets or pretrained models, AROS significantly outperforms existing methods in adversarial settings, achieving up to a 40% improvement in AUROC and demonstrating competitive results under clean setups (see Table 2b). Specifically, AROS also exhibits greater robustness under various corruptions, further underscoring its effectiveness in OOD detection. We further verify our approach through an extensive ablation study of various components in AROS (see Section 6).

We note the superiority of AROS compared to representative methods in terms of robust OOD detection. Notably, AROS, *without* relying on pre-trained models or extra datasets, improves adversarial robust OOD detection performance from 45.9% to **74.0%**. In the OSR setup, the results

*Table 3.* Performance of OOD detection methods under various types of non-adversarial perturbations, referred to as image corruptions, as introduced in the CIFAR-10-C and CIFAR-100-C datasets (48), measured by AUROC (%). Specifically, test inputs, including both ID and OOD, are perturbed with a particular corruption in each experiment.

| Dataset | | Methods | Corruption | | | | | | | | | | | | | | | Mean |
|---|---|---|---|---|---|---|---|---|---|---|---|---|---|---|---|---|---|---|
| $\mathcal{D}_{in}$ | $\mathcal{D}_{out}$ | | Gauss. | Shot | Impulse | Defocus | Glass | Motion | Zoom | Snow | Frost | Fog | Bright | Contrast | Elastic | Pixel | JPEG | |
| CIFAR-10-C | CIFAR-100-C | VOS | 56.2 | 67.5 | 76.5 | 77.7 | 73.9 | 78.7 | 76.3 | 72.0 | 54.1 | 77.0 | 58.5 | 79.1 | 81.2 | 83.6 | 74.4 | 72.5 |
| | | DHM | 57.7 | 78.7 | 72.4 | 75.4 | 75.6 | 73.9 | 77.5 | 75.8 | 70.8 | 56.8 | 74.5 | 58.0 | 77.4 | 78.4 | 80.6 | 72.2 |
| | | CATEX | 62.4 | 80.9 | 73.0 | 78.4 | 76.3 | 78.6 | 81.3 | 79.9 | 78.9 | 58.3 | 80.0 | 54.0 | 79.0 | 80.5 | 82.4 | 74.9 |
| | | CSI | 54.7 | 58.0 | 58.7 | 62.9 | 61.7 | 69.0 | 65.9 | 77.2 | 69.2 | 74.8 | 91.9 | 65.8 | 74.2 | 62.6 | 74.9 | 68.1 |
| | | ATOM | 57.3 | 75.5 | 63.6 | 70.7 | 72.2 | 69.9 | 74.6 | 77.2 | 76.5 | 55.3 | 80.5 | 54.1 | 74.7 | 77.4 | 80.8 | 70.7 |
| | | ALOE | 54.5 | 76.4 | 64.0 | 71.5 | 73.0 | 70.9 | 75.5 | 78.2 | 77.9 | 56.3 | 81.5 | 54.0 | 76.9 | 79.3 | 82.1 | 71.5 |
| | | ATD | 59.2 | 79.2 | 71.0 | 76.7 | 76.9 | 75.6 | 79.5 | 78.2 | 74.9 | 59.5 | 77.8 | 59.5 | 79.0 | 80.8 | 82.9 | 73.7 |
| | | RODEO | 58.6 | 76.0 | 68.5 | 73.5 | 73.8 | 72.1 | 75.5 | 74.5 | 70.9 | 74.5 | 57.7 | 75.3 | 76.8 | 79.5 | 71.0 | |
| | | AROS | 84.3 | 76.5 | 79.2 | 83.8 | 77.3 | 82.0 | 81.3 | 83.4 | 84.0 | 84.0 | 84.7 | 83.3 | 80.7 | 79.6 | 82.5 | 81.8 |
| CIFAR-100-C | CIFAR-10-C | VOS | 53.8 | 55.7 | 65.6 | 58.2 | 47.1 | 51.4 | 57.6 | 53.9 | 59.0 | 57.2 | 56.5 | 54.8 | 48.2 | 59.4 | 51.1 | 55.3 |
| | | DHM | 58.2 | 59.9 | 64.0 | 57.7 | 48.9 | 58.0 | 57.4 | 57.6 | 58.5 | 57.9 | 58.1 | 58.3 | 49.8 | 55.6 | 56.7 | 57.1 |
| | | CATEX | 57.4 | 60.2 | 65.7 | 59.6 | 64.9 | 62.9 | 59.3 | 67.5 | 61.4 | 59.8 | 60.0 | 64.2 | 56.8 | 57.5 | 58.6 | 61.0 |
| | | CSI | 50.1 | 48.8 | 50.6 | 47.8 | 47.5 | 46.9 | 46.8 | 50.6 | 50.3 | 51.8 | 49.9 | 52.2 | 42.9 | 48.0 | 47.7 | 48.8 |
| | | ATOM | 55.3 | 51.2 | 53.1 | 50.2 | 49.9 | 49.2 | 49.6 | 53.1 | 52.8 | 54.4 | 52.4 | 54.8 | 45.0 | 50.4 | 50.8 | 51.5 |
| | | ALOE | 56.1 | 53.4 | 62.8 | 54.5 | 51.8 | 54.9 | 54.1 | 54.4 | 55.6 | 54.8 | 52.7 | 56.4 | 47.8 | 51.7 | 53.2 | 54.3 |
| | | ATD | 56.0 | 57.4 | 61.5 | 57.5 | 44.8 | 57.1 | 54.2 | 56.9 | 58.3 | 55.2 | 53.7 | 57.5 | 49.3 | 50.8 | 56.0 | 55.1 |
| | | RODEO | 54.9 | 58.1 | 60.6 | 56.4 | 51.0 | 60.5 | 58.9 | 58.4 | 57.9 | 54.6 | 57.4 | 52.3 | 52.7 | 53.5 | 51.2 | 55.9 |
| | | AROS | 71.8 | 74.8 | 67.7 | 59.6 | 72.6 | 73.9 | 65.7 | 68.5 | 64.4 | 59.8 | 75.0 | 64.2 | 72.8 | 69.5 | 58.6 | 67.9 |

*Table 4a.* Comparison of post-hoc OOD detection methods using different classifiers trained with various strategies and evaluated with multiple scoring functions. The comparison (Clean/PGD$^{1000}$) is conducted under clean and PGD$^{1000}$ conditions, measured by AUROC (%).

| Classifier | Posthoc Method | CIFAR-10 | | CIFAR-100 | |
|---|---|---|---|---|---|
| | | CIFAR-100 | SVHN | CIFAR-10 | SVHN |
| Standard | MSP | 87.9/0.0 | 91.8/1.4 | 75.4/0.2 | 71.4/3.6 |
| | MD | 88.5/4.3 | 99.1/0.6 | 75.0/1.9 | 98.4/0.6 |
| | OpenMax | 86.4/0.0 | 94.7/2.8 | 77.6/0.0 | 93.9/4.2 |
| AT | MSP | 79.3/16.0 | 85.1/19.7 | 67.2/10.7 | 74.6/11.3 |
| | MD | 81.4/25.6 | 88.2/27.5 | 71.8/15.0 | 81.5/19.7 |
| | OpenMax | 82.4/27.8 | 86.5/26.9 | 80.0/16.4 | 75.4/22.9 |
| ODENet | MSP | 84.2/10.6 | 89.3/15.4 | 69.7/12.5 | 76.1/23.8 |
| | MD | 80.7/9.1 | 84.6/13.0 | 66.4/14.8 | 72.9/16.4 |
| | OpenMax | 83.8/14.2 | 87.4/20.9 | 70.3/15.6 | 75.6/18.2 |
| LyaDEQ | MSP | 77.5/56.5 | 83.7/58.5 | 69.1/48.0 | 69.4/53.3 |
| | MD | 79.1/56.9 | 82.0/56.5 | 60.3/ 53.4 | 69.3/54.2 |
| | OpenMax | 76.0/47.4 | 77.5/56.5 | 67.8/57.1 | 73.3/ 58.0 |
| ASODE | MSP | 76.3/56.3 | 80.5/62.5 | 64.6/44.9 | 64.6/58.9 |
| | MD | 74.9/49.5 | 76.1/54.4 | 59.3/ 52.0 | 72.1/55.1 |
| | OpenMax | 72.6/44.2 | 75.9/57.9 | 66.1/52.1 | 80.5/ 50.4 |
| SODEF | MSP | 83.5/61.9 | 86.4/65.3 | 67.2/53.1 | 73.7/60.4 |
| | MD | 75.4/57.7 | 81.9/64.2 | 65.8/58.4 | 71.8/62.5 |
| | OpenMax | 82.8/65.3 | 86.4/69.1 | 66.3/56.6 | 75.2/64.9 |
| AROS | N/A | 88.2/80.1 | 93.0/86.4 | 74.3/67.0 | 82.5/70.6 |

*Table 4b.* Performance of OOD detection methods under clean and PGD$^{1000}$, measured by AUPR↑ (%) and FPR95 ↓(%) metrics. The perturbation budget $\epsilon$ is set to $\frac{8}{255}$. The table cells present results in the 'Clean/PGD$^{1000}$' format.

| Method | Metric | CIFAR-10 | | CIFAR-100 | |
|---|---|---|---|---|---|
| | | CIFAR-100 | SVHN | CIFAR-10 | SVHN |
| VOS | AUPR↑ | 85.8/0.0 | 90.4/6.2 | 75.8/0.0 | 93.9/7.6 |
| | FPR95↓ | 35.2/100.0 | 38.2/99.8 | 48.7/100.0 | 41.5/98.2 |
| DHM | AUPR↑ | 100.0/0.3 | 100.0/4.8 | 100.0/0.0 | 100.0/3.2 |
| | FPR95↓ | 0.2/99.2 | 0.0/98.5 | 1.1/100.0 | 0.4/99.7 |
| CATEX | AUPR↑ | 89.5/0.4 | 93.1/7.6 | 84.2/0.0 | 96.6/1.3 |
| | FPR95↓ | 36.6/99.1 | 27.3/95.6 | 42.8/100.0 | 37.1/98.4 |
| CSI | AUPR↑ | 93.4/0.0 | 98.2/4.6 | 65.8/0.0 | 82.9/0.4 |
| | FPR95↓ | 40.6/100.0 | 37.4/99.1 | 65.2/100.0 | 42.6/97.5 |
| ATOM | AUPR↑ | 97.9/5.8 | 98.3/11.6 | 89.3/5.1 | 94.6/7.2 |
| | FPR95↓ | 24.0/96.4 | 12.7/93.1 | 38.6/98.0 | 29.2/97.9 |
| ALOE | AUPR↑ | 80.4/21.7 | 86.5/27.3 | 54.8/9.2 | 85.1/18.6 |
| | FPR95↓ | 38.6/89.2 | 45.1/93.7 | 72.8/96.1 | 57.4/84.8 |
| ATD | AUPR↑ | 81.9/44.6 | 85.3/53.7 | 61.4/27.2 | 68.3/26.1 |
| | FPR95↓ | 47.3/86.2 | 42.4/83.0 | 68.2/94.8 | 59.0/91.9 |
| RODEO | AUPR↑ | 83.5/47.0 | 88.2/51.6 | 72.8/26.5 | 81.7/42.9 |
| | FPR95↓ | 42.9/81.3 | 49.6/75.4 | 65.3/89.0 | 61.8/83.5 |
| AROS | AUPR↑ | 87.2/80.5 | 97.2/91.4 | 71.0/65.3 | 72.4/66.8 |
| | FPR95↓ | 39.3/45.2 | 15.5/27.0 | 54.2/67.8 | 46.3/62.7 |

increased from 52.1% to **69.3%**. Similar gains are observed in robustness against corruptions, as shown in Table 3.

For instance, performance improved from 72.5% to **81.8%** on the CIFAR-10-C vs. CIFAR-100-C setup, and from 61.0% to **67.9%** on the CIFAR-100-C vs. CIFAR-10-C benchmark. Meanwhile, AROS achieves competitive results in clean scenarios (82.0%) compared to state-of-the-art methods like DHM (93.4%), though it should be noted that DHM performs near zero under adversarial attacks. The trade-off between robustness and clean performance is well-known in the field (5; 105; 44), and AROS offers the best overall balance among existing methods. Furthermore, we demonstrate that by using pre-trained models or auxiliary data, AROS's clean performance can be further improved (see Appendix A3). Moreover, we provide additional experiments in Appendix A3 to support our claims.

**Classifier Training Strategies for Robust OOD Detection.** We assessed the impact of different training strategies on the robust OOD detection performance of various classifiers, including those trained with standard training, adversarial training (AT), and NODE-based methods such as ODENet,

*Table 5.* An ablation study (Clean/PGD$^{1000}$), measured by AUROC (%), on our method with the exclusion of different components while keeping all others intact. The left side is the configurations.

| Config | Components | | | | | | CIFAR10 | | CIFAR100 | | ImageNet-1k | |
|---|---|---|---|---|---|---|---|---|---|---|---|---|
| | Adv. Trained Backbone | Fake Sampling | Orthogonal Binary Layer | Extra Data | $\mathcal{L}_{\text{CE}}$ | $\mathcal{L}_{\text{SL}}$ | CIFAR100 | SVHN | CIFAR10 | SVHN | Texture | iNaturalist |
| A | ✓ | ✓ | ✓ | - | ✓ | - | 81.4/17.6 | 86.9/23.5 | 68.4/12.7 | 79.0/16.2 | 76.4/18.8 | 82.7/20.3 |
| B | - | ✓ | ✓ | - | - | ✓ | 90.1/56.7 | 93.8/51.5 | 75.2/41.8 | 82.0/47.5 | 81.9/36.0 | 84.9/48.6 |
| C | ✓ | ✓ | - | - | - | ✓ | 85.6/67.3 | 88.2/74.6 | 66.9/57.1 | 78.4/63.3 | 75.4/60.7 | 79.8/70.2 |
| D | ✓ | - | ✓ | - | - | ✓ | 85.3/76.5 | 89.4/78.1 | 70.5/61.3 | 74.4/62.5 | 76.1/67.4 | 81.3/72.7 |
| E *(Ours)* | ✓ | ✓ | ✓ | - | - | ✓ | 88.2/80.1 | 93.0/86.4 | 74.3/67.0 | 81.5/70.6 | 78.3/69.2 | 84.6/75.3 |
| F *(Ours+Data)* | ✓ | ✓ | ✓ | ✓ | - | ✓ | 90.4/81.6 | 94.2/87.9 | 75.7/68.1 | 82.2/71.8 | 79.2/70.4 | 85.1/76.8 |

LyaDEQ, SODEF, and ASODE. To utilize these classifiers as OOD detectors, various post-hoc score functions were applied, as described in Section 3. The results are presented in Table 4a. In brief, adversarially trained classifiers exhibit enhanced robustness compared to standard training but still fall short of optimal performance. Furthermore, the time-invariance assumption in SODEF leads to improved robust performance relative to ODENet, LyaDEQ, and ASODE by effectively constraining the divergence between output states, which motivated us to explore similar frameworks. Notably, AROS demonstrates superior performance compared to all these approaches.

**Implementation details.** We use a WideResNet-70-16 model as $f_\theta$ (106) and train it for 200 epochs on classification using PGD$^{10}$. For the integration of $h_\phi$, an integration time of $T = 5$ is applied. To implement the orthogonal layer $B_\eta$, we utilize the `geotorch.orthogonal` library. Training with the loss $\mathcal{L}_{\text{SL}}$ is performed over 100 epochs. We used SGD as the optimizer, employing a cosine learning rate decay schedule with an initial learning rate of 0.05 and a batch size of 128. See Appendix A3 for more details and additional ablation studies on different components of AROS.

# 6 ABLATION STUDY

**AROS Components.** To verify the effectiveness of AROS, we conducted ablation studies across various datasets. The corresponding results are presented in Table 5. In each experiment, individual components were replaced with alternative ones, while the remaining elements were held constant. In *Config A*, we ignored the designed loss function $\mathcal{L}_{\text{SL}}$ and instead utilized the cross-entropy loss function $\mathcal{L}_{\text{CE}}$ for binary classification. *Config B* represents the scenario in which we train the classifier in the first step without adversarial training on the ID data, instead using standard training. This reduces robustness as $f_\theta$ becomes more susceptible to perturbations within ID classes, ultimately making the final detector more vulnerable to attacks. In *Config C*, the orthogonal binary layer was replaced with a regular binary layer. In *Config D*, rather than estimating the ID distribution and sampling OOD data in the embedding space, we substituted this process by creating random Gaussian noise in the embedding space as fake OOD data. This removes the conditioning of the fake OOD distribution on the ID data and, as a result, makes them unrelated. This is in line with previous works that have shown that related and nearby auxiliary OOD samples are more useful (43; 24). *Config E* represents our default pipeline. Finally, in *Config F*, we extended AROS by augmenting the fake OOD embedding data with additional OOD images (i.e., Food-101 (107)) alongside the proposed fake OOD strategy. Specifically, we transformed these additional OOD images into the embedding space using $f_\theta$ and combined them with the crafted fake embeddings, which led to enhanced performance.

# 7 CONCLUSIONS

In this paper we introduce AROS, a framework for improving OOD detection under adversarial attacks. By leveraging Lyapunov stability theory, AROS drives ID and OOD samples toward stable equilibrium points to mitigate adversarial perturbations. Fake OOD samples are generated in the embedding space, and a tailored loss function is used to enforce stability. Additionally, an orthogonal binary layer is employed to enhance the separation between ID and OOD equilibrium points. Limitations and future directions can be found in the Appendix A6.

## 8 ACKNOWLEDGMENTS

The authors thank the SNSF (Grant No. 320030-227871) and EPFL for funding. MWM is the Bertarelli Foundation Chair of Integrative Neuroscience.

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
