# Appendix

## A1 REVIEW OF RELATED WORK

In our experimental evaluation we compare our results with works that have proposed a defense mechanism for their OOD detection method, as well as those that have not. We will describe each separately below. Subsequently, we will detail the post-hoc score functions.

**Adversarially Robust OOD Detection Methods.** Previous adversarially robust methods for OOD detection have built on the outlier exposure (27) technique and conducted adversarial training on a union of exposed outliers and ID samples. These methods primarily aim to enhance the outlier exposure technique by improving the diversity of the extra OOD image dataset. To tackle this, ALOE leverages an auxiliary OOD dataset and perturbs the samples to adversarially maximize the KL-divergence between the model's output and a uniform distribution. It is important to note that the adversarial evaluation in their pipeline was less rigorous than that in our approach, due primarily to the limited budget allocated for adversarial attacks. ATOM, in contrast, selectively samples informative outliers rather than using random outliers. The adversarial evaluation of ATOM is also not entirely standard, as it exclusively targets OOD test samples. However, a more comprehensive approach would involve attacking both ID and OOD test samples to ensure enhanced robustness. Meanwhile, ATD employs a GAN to generate auxiliary OOD images for adversarial training, instead of relying on an external dataset. RODEO aims to demonstrate that enhancing the diversity of auxiliary OOD images, while maintaining their stylistic and semantic alignment with ID samples, will improve robustness. Consequently, they utilize a pretrained CLIP model (108) and a diffusion model for OOD image synthesis (43; 109; 54; 110; 111; 112).

**Standard OOD Detection Methods.** CSI enhances the outlier detection task by building on standard contrastive learning through the introduction of 'distributionally-shifted augmentations'—transformations that encourage the model to treat augmented versions of a sample as OOD. This approach enables the model to learn representations that more effectively distinguish between ID and OOD samples. The paper also proposes a detection score based on the contrastive features learned through this training scheme, which demonstrates effectiveness across various OOD detection scenarios, including one-class, multi-class, and labeled multi-class settings. DHM involves modeling the joint density of data and labels in a single forward pass. By factorizing this joint density into three sources of uncertainty (aleatoric, distributional, and parametric), DHMs aim to distinguish in-distribution samples from OOD samples. To achieve computational efficiency and scalability, the method employs weight normalization during training and utilizes normalizing flows (NFs) to model the probability distribution of the features. The key idea is to use bi-Lipschitz continuous mappings, enabled by spectral normalization, which allows the use of state-of-the-art deep neural networks for learning expressive and geometry-preserving representations of data. VOS uses virtual outliers to regularize the model's decision boundary and improve its ability to distinguish between ID and OOD data. The framework includes an unknown-aware training objective that uses contrastive learning to shape the uncertainty surface between the known data and the synthesized outliers. This method is effective for both object detection and image classification. CATEX use of two hierarchical contexts—perceptual and spurious—to describe category boundaries more precisely through automatic prompt tuning in vision-language models like CLIP. The perceptual context distinguishes between different categories (e.g., cats vs. apples), while the spurious context helps identify samples that are similar but not ID (e.g., distinguishing cats from panthers). This hierarchical structure helps create more precise category boundaries.

**Post-hoc Methods for OOD Detection.** Post-hoc methods for OOD detection are approaches applied after training a classifier on ID samples. These methods utilize information from the classifier to indicate OOD detection. A simple but effective approach is the Maximum Softmax Probability (MSP) method. Applied to a $K$-class classifier $f_c$, MSP returns $\max_{c \in \{1,2,...,K\}} f_c(x)$ as the likelihood that the sample $x$ belongs to the ID set. In contrast, OpenMax replaces the softmax layer with a calibrated layer that adjusts the logits by fitting a class-wise probability model, such as the Weibull distribution. Another perspective on OOD detection is to measure the distance of a sample to class-conditional distributions. The Mahalanobis distance (MD) is a prominent method for this. For an ID set with $K$ classes, MD-based approaches fit a class-conditional Gaussian distribution $\mathcal{N}(\mu_k, \Sigma)$ to the pre-logit features $z$. The mean vector and covariance matrix are calculated as follows:

$$\mu_k = \frac{1}{N} \sum_{i:y_i=k} z_i, \quad \Sigma = \frac{1}{N} \sum_{k=1}^{K} \sum_{i:y_i=k} (z_i - \mu_k)(z_i - \mu_k)^T, \quad k = 1, 2, ..., K.$$

The MD for a sample $z'$ relative to class $k$ is defined as: $MD_k(z') = (z' - \mu_k)^T \Sigma^{-1} (z' - \mu_k)$.

The final score MD used for OOD detection is given by: $\text{score}_{\text{MD}}(x') = -\min_k \{MD_k(z')\}$

## A2 ALGORITHM

Here we provide pseudocode for our proposed AROS framework, designed for adversarially robust OOD detection. We begin by leveraging adversarial training on ID data to obtain robust feature representations, utilizing the well-known practice of adversarial training with 10-step PGD. These features are then used to fit class-conditional multivariate Gaussians, from which we sample low-likelihood regions to generate fake OOD embeddings, effectively creating a proxy for real OOD data in the embedding space. By constructing a balanced training set of ID and fake OOD embeddings, we then employ a stability-based objective using a NODE pipeline, coupled with an orthogonal binary layer. This layer maximizes the separation between the equilibrium points of ID and OOD samples, promoting a robust decision boundary under perturbations. During inference, we compute the OOD score based on the orthogonal binary layer's output, enabling the model to reliably distinguish ID samples from OOD samples, even in the presence of adversarial attacks.

---

**Algorithm 1** Adversarially Robust OOD Detection through Stability (AROS)

---

**Require:** ID training samples $\mathcal{D}_{\text{train}}^{\text{in}}$ consisting of $N$ samples spanning $k$ classes, a $k$-class classifier $f_\theta$, a time-invariant NODE $h_\phi$, an orthogonal binary layer $B_\eta$, an $\ell_\infty$ norm constraint, perturbation budget $\epsilon$ set to $\frac{4}{255}$ for low-resolution and $\frac{8}{255}$ for high-resolution, Lyapunov-based loss $\mathcal{L}_{\text{ST}}$.

**Ensure:** Adversarially Robust OOD Detector

1:
2: **Step A-1. Adversarial Training of Classifier $f_\theta$ on ID:**
3: **for** each sample $(x, y) \in \mathcal{D}_{\text{train}}^{\text{in}}$ **do**
4:     Generate adversarial example $x^*$ using PGD[10] to maximize the cross-entropy loss:
5:     $x^* \leftarrow x + \alpha \cdot \text{sign}(\nabla_x \mathcal{L}_{\text{CE}}(f_\theta(x), y))$, constrained by $\epsilon$
6:     Train the classifier $f_\theta$ on $(x^*, y)$ to improve robustness
7: **end for**
8: **Step A-2. Feature Extraction:**
9: Map ID samples to the robust embedding space: $\text{ID}_{\text{features}} = f_\theta(\mathcal{D}_{\text{train}}^{\text{in}})$
10:
11: **Step B. Fake OOD Embedding Generation:**
12: Fit a class-conditional multivariate Gaussian $\mathcal{N}(\mu_j, \Sigma_j)$ on $\text{ID}_{\text{features}}$ for each class $j$
13: **for** $0 \le j < K$ **do**
14:     **for** $0 \le i < \left\lceil \frac{N}{K} \right\rceil$ **do**
15:         Sample synthetic OOD embeddings $r$ from the low-likelihood regions of $\mathcal{N}(\mu_j, \Sigma_j)$
16:         $r_{\text{OOD}} \leftarrow r_{\text{OOD}} \cup \{r\}$
17:     **end for**
18: **end for**
19: Construct a balanced training set $X_{\text{train}} = \text{ID}_{\text{features}} \cup r_{\text{OOD}}$
20:
21: **Step C. Lyapunov Stability Objective Function:**
22: Create the pipeline $B_\eta(h_\phi(.))$ and train it on $X_{\text{train}}$ using $\mathcal{L}_{\text{ST}}$
23:
24: **Step D. Inference:**
25: **for** each test sample $x_{\text{test}}$ **do**
26:     Compute the OOD score based on the probability output of the orthogonal binary layer for each input image $x$: OOD score$(x) = f_\theta(B_\eta(h_\phi(x)))[1]$
27: **end for**

---

## A3 SUPPLEMENTARY EXPERIMENTAL RESULTS AND DETAILS FOR AROS

### A3.1 ADDITIONAL EXPERIMENTAL RESULTS

Each experiment discussed in the main text was repeated 10 times, with the reported results representing the mean of these trials. Here, we provide the standard deviation across these runs, as summarized in Table 6a.

**Table 6a: Standard deviation of AROS performance under both clean and PGD, across 10 repeated experiments** 'Clean/PGD$^{1000}$'.

| AROS | CIFAR-10 | | | CIFAR-100 | | | ImageNet-1k | | |
|---|---|---|---|---|---|---|---|---|---|
| | CIFAR-100 | SVHN | LSUN | CIFAR-10 | SVHN | LSUN | Texture | iNaturalist | LSUN |
| Mean Performance | 88.2 / 80.1 | 93.0 / 86.4 | 90.6 / 82.4 | 74.3 / 67.0 | 81.5 /7 0.6 | 74.3 / 68.1 | 78.3 / 69.2 | 84.6 / 75.3 | 79.4 / 69.0 |
| Standard Deviation | ± 0.9 / ± 1.6 | ± 0.7 / ± 1.3 | ± 0.6/± 0.9 | ± 1.3 / ± 1.8 | ± 1.4 / ± 2.0 | ± 1.8 / ± 2.3 | ± 2.5 / ± 3.1 | ± 1.9 / ± 3.0 | ± 2.8 / ± 3.7 |

#### A3.1.1 COMPARATIVE ANALYSIS OF AROS AND DIFFUSION-BASED PURIFICATION METHODS

We present additional experiments to further demonstrate the effectiveness of AROS. We compare AROS's performance against diffusion-based (purification) methods, with results detailed in Table 7a.

Purification techniques in adversarial training aim to enhance model robustness by 'purifying' or 'denoising' adversarial examples prior to feeding them into the model. The primary goal is to mitigate adversarial perturbations through preprocessing, often by leveraging neural networks or transformation-based methods to restore perturbed inputs to a state resembling clean data. A common approach involves training a generative diffusion model on the original training samples, which is then utilized as a purification module (113; 114; 115). In contrast, certain approaches categorize stable NODE-based methods as non-diffusion-based strategies for improving robustness (116).

However, in the context of OOD detection, purification using diffusion methods (117) may not be effective. This is primarily because diffusion models trained on ID samples tend to shift the features of unseen OOD data towards ID features during the reverse process. Such a shift can mistakenly transform OOD samples into ID, compromising both OOD detection performance and clean robustness (118). To emphasize this distinction, we compare our approach against purification using diffusion-based models. Specifically, we adopt the AdvRAD (119) setup for OOD detection and present a comparison with our method in Table 7a.

**Table 7a: Comparison of AROS and AdvRAD under clean and PGD$^{1000}$($l_\infty$) evaluation**, measured by AUROC (%). The table cells denote results in the 'Clean/PGD$^{1000}$' format. The perturbation budget $\epsilon$ is set to $\frac{8}{255}$ for low-resolution datasets and $\frac{4}{255}$ for high-resolution datasets.

| Method | CIFAR-10 | | | CIFAR-100 | | | ImageNet-1k | | |
|---|---|---|---|---|---|---|---|---|---|
| | CIFAR-100 | SVHN | LSUN | CIFAR-10 | SVHN | LSUN | Texture | iNaturalist | LSUN |
| AdvRAD | 61.0/49.0 | 68.5/52.7 | 66.8/50.7 | 54.8/49.4 | 60.1/52.0 | 54.8/50.2 | 57.7/51.0 | 62.4/56.5 | 58.5/50.9 |
| AROS | 88.2/80.1 | 93.0/86.4 | 90.6/ 82.4 | 74.3/67.0 | 81.5/70.6 | 74.3/68.1 | 78.3/ 69.2 | 84.6/75.3 | 79.4/69.0 |

#### A3.1.2 AROS UNDER ADDITIONAL ADVERSARIAL ATTACKS

In this section, we further emphasize the robustness of AROS by evaluating its performance under different attacks. First, we conducted additional experiments considering different perturbation norms, specifically substituting the $l_\infty$ norm with the $l_2$ norm in PGD$^{1000}$, setting $\epsilon = \frac{128}{255}$. The outcomes of this evaluation are presented in Table 8a. See Figure 3 for an illustration of some clean and perturbed samples.

Next, we evaluated the robustness of AROS under PGD$^{1000}$($l_\infty$) with a higher perturbation budget (i.e., $\epsilon = \frac{16}{255}$), as shown in Table 8b. Furthermore, the performance of AROS under AutoAttack (AA) and adaptive AA is reported in Table 9a.

Additionally, evaluating a model against a variety of transfer-based adversarial attacks is crucial for understanding its robustness in real-world scenarios, where adversarial examples crafted for one

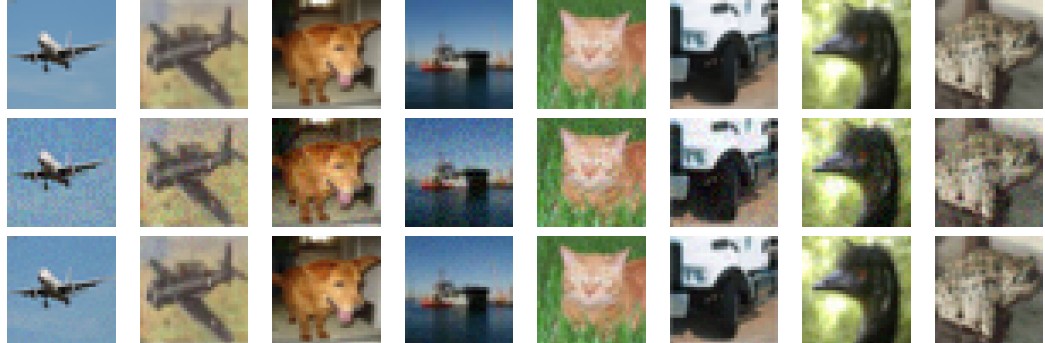

Figure 3: Visualization of clean and perturbed images from the CIFAR-10 dataset to illustrate the impact of perturbations on semantic content. The first row depicts clean images, while the second and third rows show images perturbed with $L_\infty$ norm of $\frac{8}{255}$ and $L_2$ norm of $\frac{128}{255}$, respectively. Despite the added perturbations, the semantic content of the images remains unchanged, demonstrating that robustness expectations from models under these perturbations are fair.

model can successfully deceive others. These attacks simulate diverse and challenging conditions by targeting features such as generalization, input transformations, and model invariance, providing a comprehensive assessment of the model's resilience. Such evaluations reveal potential vulnerabilities, measure performance under adversarial conditions, and offer insights into the model's ability to generalize across different attack strategies.

Motivated by this, we evaluate our method, as well as DHM and RODEO, against several adversarial attacks, including DTA (120), DeCoWA (121), and SASD-WS (122). We selected DHM and RODEO for comparison due to their strong performance in clean detection and robust detection, respectively. For these experiments, we utilized the implementation provided in the `https://github.com/Trustworthy-AI-Group/TransferAttack` repository. The results of this evaluation are presented in Table 9b.

The consistent and superior performance of AROS against the aforementioned attacks underscores its effectiveness in adversarial scenarios.

**Table 8a: Evaluation of AROS under PGD$^{1000}(l_2)$,** where $\epsilon = \frac{128}{255}$ is used for low-resolution datasets and $\epsilon = \frac{64}{255}$ for high-resolution datasets. The results are measured by AUROC (%).

| Attack | CIFAR-10 | | | CIFAR-100 | | | ImageNet-1k | | |
|---|---|---|---|---|---|---|---|---|---|
| | CIFAR-100 | SVHN | LSUN | CIFAR-10 | SVHN | LSUN | Texture | iNaturalist | LSUN |
| PGD$^{1000}(l_2)$ | 81.6 | 87.1 | 83.7 | 67.4 | 71.0 | 69.3 | 70.5 | 75.9 | 69.8 |

**Table 8b: : Evaluation of AROS under PGD$^{1000}(l_\infty)$,** where $\epsilon = \frac{16}{255}$ is used for all datasets, including both high- and low-resolution ones. The results are reported in terms of AUROC (%).

| Attack | CIFAR-10 | | | CIFAR-100 | | | ImageNet-1k | | |
|---|---|---|---|---|---|---|---|---|---|
| | CIFAR-100 | SVHN | LSUN | CIFAR-10 | SVHN | LSUN | Texture | iNaturalist | LSUN |
| PGD$^{1000}(l_\infty)$ | 70.3 | 78.4 | 72.6 | 58.2 | 63.9 | 60.4 | 60.5 | 66.8 | 62.7 |

### A3.2 DETECTING BACKDOORED SAMPLES

We conducted an experiment to evaluate our method's ability to detect clean samples (without triggers) as ID and poisoned samples as OOD. The results are presented in this section, with STRIP (123) included as a baseline method for identifying Trojaned samples.

**Table 9a: Evaluation of AROS under AutoAttack and Adaptive AA.** The perturbation budget $\epsilon$ is set to $\frac{8}{255}$ for low-resolution datasets and $\frac{4}{255}$ for high-resolution datasets.

| Attack | CIFAR-10 | | | CIFAR-100 | | | ImageNet-1k | | |
|---|---|---|---|---|---|---|---|---|---|
| | CIFAR-100 | SVHN | LSUN | CIFAR-10 | SVHN | LSUN | Texture | iNaturalist | LSUN |
| AutoAttack | 78.9 | 83.4 | 80.2 | 66.5 | 70.2 | 68.9 | 67.4 | 73.6 | 67.1 |
| Adaptive AA | 76.4 | 82.9 | 78.6 | 65.2 | 67.4 | 68.3 | 66.1 | 70.5 | 66.9 |

**Table 9b: Comparison of AROS, DHM, and RODEO under Transfer-Based Attacks.** The perturbation budget $\epsilon$ is set to $\frac{16}{255}$. Results compare AROS, DHM, and RODEO across DTA, DeCoWA, and SASD-WS attacks.

| Attack | Methods | CIFAR-10 | | | CIFAR-100 | | | ImageNet-1k | | |
|---|---|---|---|---|---|---|---|---|---|---|
| | | CIFAR-100 | SVHN | LSUN | CIFAR-10 | SVHN | LSUN | Texture | iNaturalist | LSUN |
| | AROS | **84.7** | **86.4** | 88.2 | **71.5** | **76.6** | **71.7** | **76.2** | **83.5** | **75.9** |
| DTA | RODEO | 61.6 | 68.5 | **89.3** | 51.1 | 63.1 | 70.9 | 61.1 | 60.6 | 59.2 |
| | DHM | 72.9 | 78.3 | 78.9 | 75.4 | 79.1 | 74.1 | 55.9 | 59.2 | 57.6 |
| | AROS | **83.8** | **85.2** | **86.8** | **69.6** | **77.2** | **69.4** | **72.6** | **79.5** | **74.8** |
| DeCoWA | RODEO | 57.8 | 64.6 | 80.4 | 43.0 | 57.7 | 67.5 | 54.7 | 54.8 | 52.5 |
| | DHM | 60.1 | 64.3 | 64.9 | 68.9 | 62.1 | 60.8 | 54.9 | 58.4 | 56.0 |
| | AROS | **82.3** | **84.9** | **81.7** | **67.6** | **75.0** | **67.7** | **72.4** | **77.5** | **71.2** |
| SASD-WS | RODEO | 49.5 | 53.4 | 79.4 | 44.7 | 56.5 | 66.1 | 53.2 | 46.9 | 47.0 |
| | DHM | 64.6 | 69.3 | 61.3 | 56.3 | 67.8 | 59.7 | 53.1 | 51.3 | 49.2 |

It is important to highlight a key distinction in the context of OOD detection, which underpins our study. In traditional OOD detection scenarios, ID and OOD datasets typically differ at the semantic level (e.g., CIFAR-10 versus SVHN). However, detecting backdoored samples presents a unique challenge: the presence of triggers modifies clean images at the pixel level rather than the semantic level. Consequently, this task often hinges on texture-level differences rather than semantic distinctions, necessitating models designed specifically to leverage this inductive bias effectively.

Despite these challenges, our results demonstrate that AROS achieves strong performance in detecting poisoned samples. This underscores the versatility and effectiveness of AROS, even when applied to the nuanced problem of backdoor detection. The results are presented in Table 10a.

**Table 10a: Effectiveness of the Proposed Method for Detecting Backdoor Attack Samples** across CIFAR-10, CIFAR-100, and GTSRB datasets. The results are presented for different backdoor attack methods, demonstrating the performance of AROS and STRIP.

| Method | Backdoor Attack | CIFAR-10 | CIFAR-100 | GTSRB |
|---|---|---|---|---|
| | Badnets | **80.3** | 67.5 | 72.8 |
| AROS | Wanet | **62.7** | **58.9** | **54.4** |
| | SSBA | **57.2** | 72.6 | **66.0** |
| | Badnets | 79.2 | **86.0** | **87.1** |
| STRIP | Wanet | 39.5 | 48.5 | 35.6 |
| | SSBA | 36.4 | 68.5 | 64.1 |

### A3.2.1 ARCHITECTURE COMPARISON & ENHANCING CLEAN PERFORMANCE WITH TRANSFER LEARNING

Additionally, we investigate the influence of replacing our default backbone architecture, WideResNet, with alternative architectures (Table 11a). We also explore transfer learning techniques to enhance clean performance by leveraging robust pre-trained classifiers.

Lastly, while AROS demonstrates a significant improvement of up to 40% in adversarial robustness, it shows a performance gap of approximately 10% when compared to state-of-the-art clean detection methods. Although this trade-off between robustness and clean performance is well-documented in the literature (5; 105; 44), our aim is to enhance clean performance. A promising approach would involve enhancing clean performance while preserving robustness by leveraging transfer learning through distillation from a large, robust pretrained model, rather than training the classifier

from scratch, as is currently done in the pipeline. Specifically, by utilizing adversarially pretrained classifiers on ImageNet (124), we aim to improve our clean performance. Results in lower part of Table 11a indicate that leveraging such pretrained models can improve clean performance by 6%. Moreover, we also consider using different architectures trained from scratch to further demonstrate the robustness of AROS across various backbones.

**Table 11a: Ablation study on different backbone architectures & Transfer learning.** Results are reported under clean and $PGD^{1000}(l_\infty)$ evaluations, measured by AUROC (%). Each table cell presents results in the 'Clean/$PGD^{1000}$' format.

[†] Denotes that the backbone is adversarially pretrained on ImageNet.

| Backbone ($f_\theta$) | CIFAR-10 | | | CIFAR-100 | | | ImageNet-1k | | |
|---|---|---|---|---|---|---|---|---|---|
| | CIFAR-100 | SVHN | LSUN | CIFAR-10 | SVHN | LSUN | Texture | iNaturalist | LSUN |
| WideResNet (default) | 88.2/80.1 | 93.0/86.4 | 90.6/ 82.4 | 74.3/67.0 | 81.5/70.6 | 74.3/68.1 | 78.3/ 69.2 | 84.6/75.3 | 79.4/69.0 |
| PreActResNet | 83.5/74.6 | 92.1/82.4 | 87.9/79.6 | 69.6/62.2 | 76.7/65.2 | 69.9/62.7 | 73.4/69.3 | 82.2/73.6 | 76.8/65.6 |
| ResNet18 | 82.2/74.4 | 90.5/80.7 | 84.5/81.0 | 70.7/63.7 | 79.0/66.4 | 74.6/66.4 | 77.4/64.4 | 81.8/72.3 | 78.7/67.8 |
| ResNet50 | 86.5/79.2 | 92.5/86.1 | 90.8/82.7 | 74.5/66.4 | 81.7/69.6 | 72.8/68.3 | 77.9/67.4 | 82.7/75.1 | 77.9/68.5 |
| WideResNet [†] | 91.2/75.2 | 97.4/80.9 | 92.1/77.9 | 80.5/60.1 | 87.8/70.0 | 76.7/62.9 | - | - | - |
| PreActResNet[†] | 93.1/79.5 | 96.3/84.4 | 96.5/76.6 | 75.4/62.2 | 83.1/66.7 | 79.4/64.9 | - | - | - |
| ResNet18[†] | 90.2/78.0 | 97.9/83.9 | 97.5/78.6 | 76.2/63.7 | 83.0/65.2 | 78.0/62.2 | - | - | - |
| ResNet50[†] | 90.6/76.8 | 95.5/82.3 | 94.6/74.7 | 79.5/64.9 | 85.7/69.7 | 78.7/45.5 | - | - | - |
| ViT-b-16[†] | 94.3/71.6 | 97.8/79.7 | 94.6/76.8 | 85.2/61.0 | 87.6/66.9 | 82.7/59.2 | - | - | - |

## A3.3  HYPERPARAMETER ABLATION STUDY

### A3.3.1  ABLATION STUDY ON HYPERPARAMETER OF FAKE GENERATION ($\beta$)

To craft fake samples, we fit a GMM to the embedding space of ID samples. The objective is to sample from the GMM such that the likelihood of the samples is low, ensuring they do not belong to the ID distribution and are located near its boundaries. This approach generates near-OOD samples, which are valuable for understanding the distribution manifold and improving the detector's performance.

In practice, an encoder is used as a feature extractor to obtain embeddings of ID samples. A GMM is then fitted to these embeddings, and the likelihood of each training sample under the GMM is computed. The embeddings are subsequently sorted based on their likelihoods, and the $\beta$-th minimum likelihood of the training samples is used as a threshold. Random samples are drawn from the GMM, and their likelihoods are compared against this threshold. If a sample's likelihood is lower than the threshold, it is retained as fake OOD; otherwise, it is discarded.

For instance, setting $\beta = 0.1$ ensures that the crafted fake OOD samples have a likelihood of belonging to the ID distribution that is lower than 90% of the ID samples. Conversely, setting $\beta = 0.5$ corresponds to randomly sampling from the ID distribution, rather than targeting low-likelihood regions, which leads to poor performance. An ablation study in our manuscript explores the effects of varying $\beta$ values on model performance.

The sensitivity of AROS to this hyperparameter is analyzed in detail. As shown in Table 12a, AROS demonstrates consistent performance for small values of $\beta$. Our experimental results indicate that selecting $\beta$ values in the range $[0.0, 0.1]$ achieves optimal performance, highlighting the robustness of AROS to changes in $\beta$.

**Table 12a: Ablation study on the $\beta$ hyperparameter.** Results are reported under clean and $\text{PGD}^{1000}(l_\infty)$ evaluations, measured by AUROC (%). Each table cell presents results in the 'Clean/$\text{PGD}^{1000}$' format.

| Hyperparameter | CIFAR-10 | | | CIFAR-100 | | | ImageNet-1k | | |
|---|---|---|---|---|---|---|---|---|---|
| | CIFAR-100 | SVHN | LSUN | CIFAR-10 | SVHN | LSUN | Texture | iNaturalist | LSUN |
| $\beta = 0.001$(default) | 88.2/80.1 | 93.0/86.4 | 90.6/ 82.4 | 74.3/67.0 | 81.5/70.6 | 74.3/68.1 | 78.3/ 69.2 | 84.6/75.3 | 79.4/69.0 |
| $\beta = 0.01$ | 87.3/80.2 | 92.2/85.3 | 89.4/80.7 | 74.7/65.8 | 80.2/68.9 | 73.1/66.6 | 78.1/69.5 | 83.0/74.2 | 79.0/68.5 |
| $\beta = 0.025$ | 86.3/79.6 | 93.8/85.0 | 89.3/83.1 | 72.7/67.5 | 81.7/70.8 | 74.3/68.0 | 76.7/67.7 | 84.7/75.3 | 79.8/69.7 |
| $\beta = 0.05$ | 86.5/80.1 | 92.5/86.1 | 90.8/82.7 | 74.5/66.4 | 81.7/69.6 | 72.8/68.3 | 77.9/67.4 | 82.7/75.1 | 77.9/68.5 |
| $\beta = 0.075$ | 88.4/78.4 | 92.7/85.2 | 89.1/82.3 | 72.4/65.7 | 81.5/69.4 | 72.5/67.9 | 78.6/69.7 | 84.8/73.5 | 78.8/69.3 |
| $\beta = 0.1$ | 86.6/79.5 | 92.0/85.4 | 90.9/80.9 | 72.3/65.8 | 80.3/69.4 | 73.3/66.7 | 77.8/67.5 | 84.1/74.9 | 78.2/67.8 |
| $\beta = 0.25$ | 79.4/67.5 | 89.0/79.5 | 78.7/69.4 | 67.7/54.3 | 69.5/60.3 | 70.1/62.7 | 62.6/65.0 | 65.2/57.9 | 70.9/62.0 |
| $\beta = 0.5$ | 65.4/53.0 | 77.4/62.3 | 64.7/54.6 | 57.5/40.9 | 52.4/49.2 | 49.1/41.9 | 55.8/48.9 | 67.4/59.7 | 57.9/53.8 |

### A3.3.2  ABLATION STUDY ON HYPERPARAMETERS OF THE OBJECTIVE FUNCTION ($\gamma$)

In our proposed method, we introduce the empirical loss function $\mathcal{L}_{\text{SL}}$ as follows:

$$\mathcal{L}_{\text{SL}} = \min_{\phi,\eta} \frac{1}{|X_{\text{train}}|} \Bigg( \ell_{\text{CE}}(B_\eta(h_\phi(X_{\text{train}})), y) + \gamma_1 \|h_\phi(X_{\text{train}})\|_2$$

$$+ \gamma_2 \exp\left( -\sum_{i=1}^{n} [\nabla h_\phi(X_{\text{train}})]_{ii} \right)$$

$$+ \gamma_3 \exp\left( \sum_{i=1}^{n} \left( -|[\nabla h_\phi(X_{\text{train}})]_{ii}| + \sum_{j\neq i} |[\nabla h_\phi(X_{\text{train}})]_{ij}| \right) \right) \Bigg) \Bigg) \quad (1)$$

Here, $\gamma_1$ controls the regularization term $\|h_\phi(X_{\text{train}})\|_2$, which encourages the system's state to remain near the equilibrium point. This term helps mitigate the effect of perturbations by ensuring that trajectories stay close to the equilibrium. The hyperparameters $\gamma_2$ and $\gamma_3$ weight the exponential terms designed to enforce Lyapunov stability conditions by ensuring that the Jacobian matrix satisfies strict diagonal dominance, as per Theorem 3 in the paper. Specifically, $\gamma_2$ weights the term

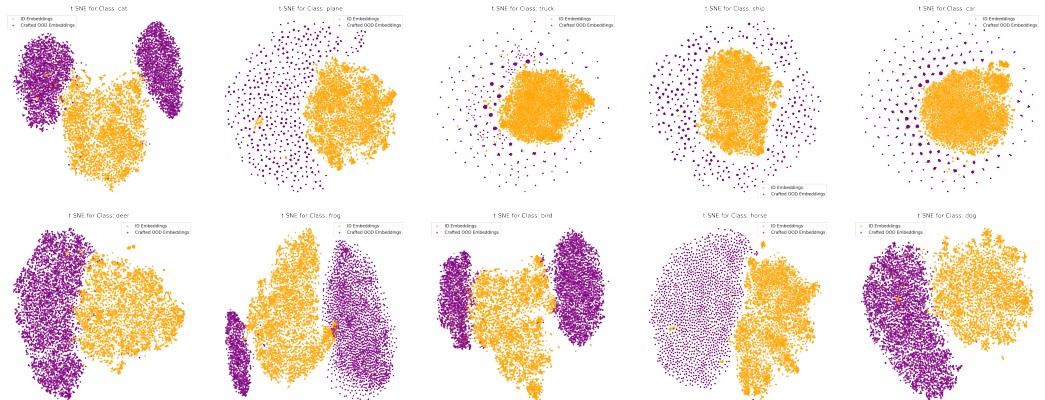

Figure 4: t-SNE visualization of CIFAR-10 embeddings and the corresponding crafted OOD embeddings for each class. Orange points represent the ID embeddings for each class, while purple points represent the synthetic OOD embeddings crafted using a GMM. The visualization highlights the separability between ID and OOD embeddings. The crafted embeddings are positioned near the boundaries of the ID concepts, emphasizing they are near OOD samples and they coverage the OOD space. The $\beta$ hyperparameter used in this experiment is set to 0.001.

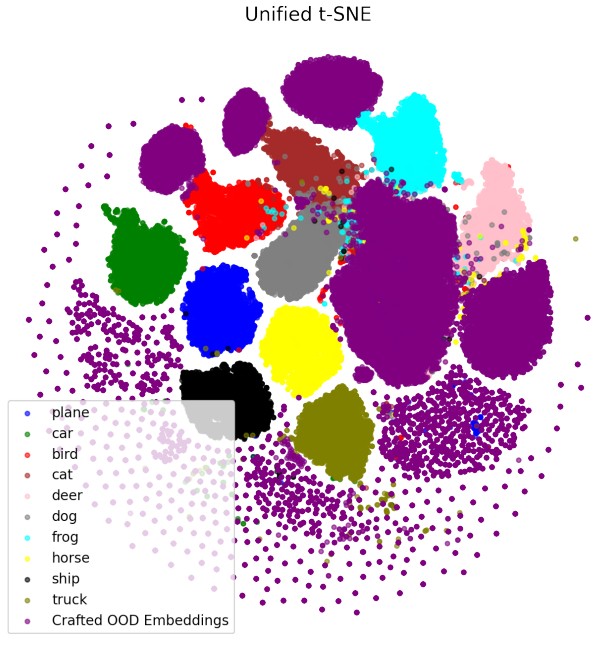

Figure 5: Unified t-SNE visualization of embeddings for all CIFAR-10 classes and their corresponding crafted OOD embeddings. Each color represents a specific CIFAR-10 class, while the purple points represent the synthetic OOD embeddings crafted using a GMM. The figure demonstrates the clustering of ID embeddings for each class and the distinct distribution of crafted OOD embeddings. The $\beta$ hyperparameter used in this experiment is set to 0.01. Highlighting both 0.01 and 0.001 leads to crafting effective fake OOD samples, lying out of ID set.

$\exp\left(-\sum_{i=1}^{n}[\nabla h_\phi(X_{\text{train}})]_{ii}\right)$, encouraging the diagonal entries of the Jacobian to be negative with

large magnitudes. $\gamma_3$ weights the term

$$\exp\left(\sum_{i=1}^{n}\left(-\left|[\nabla h_\phi(X_{\text{train}})]_{ii}\right| + \sum_{j \neq i}\left|[\nabla h_\phi(X_{\text{train}})]_{ij}\right|\right)\right),$$

promoting strict diagonal dominance by penalizing large off-diagonal entries relative to the diagonal entries.

We set $\gamma_1 = 1$ to balance the influence of the regularization term with the primary cross-entropy loss. This ensures sufficient regularization without overwhelming the classification objective. For $\gamma_2$ and $\gamma_3$, we choose $\gamma_2 = \gamma_3 = 0.05$ Inspired by related works that incorporate stability terms into the training process (8; 9; 10; 11; 12; 13; 14; 15).

We will now discuss why we select $\gamma_2$ and $\gamma_3$ equally:

- **Balanced Contribution**: Both regularization terms serve complementary purposes in enforcing the stability conditions. Equal weighting ensures that neither term dominates, maintaining a balanced emphasis on both negative diagonal dominance and strict diagonal dominance.
- **Simplified Hyperparameter Tuning**: Setting $\gamma_2$ and $\gamma_3$ equal reduces the hyperparameter search space, simplifying the tuning process without sacrificing performance.
- **Empirical Validation**: Experiments showed that equal values yield robust performance, and deviating from this balance did not provide significant benefits.

Regarding the effect of $\gamma_1$:

- **Low Values** ($\gamma_1 < 1$): Reduces the emphasis on keeping the state near the equilibrium, making the system more susceptible to perturbations.
- **High Values** ($\gamma_1 > 1$): Overly constrains the state to the equilibrium point, potentially limiting the model's capacity to learn discriminative features.
- **Chosen Value** ($\gamma_1 = 1$): Offers a good balance, ensuring sufficient regularization without compromising learning.

Similarly, the effect of $\gamma_2$ and $\gamma_3$ is as follows:

- **Low Values** ($< 0.05$): Diminish the impact of the stability constraints, reducing robustness.
- **High Values** ($> 0.05$): Overemphasize the stability terms, potentially hindering the optimization of the primary classification loss.
- **Equal Values** ($\gamma_2 = \gamma_3$): Ensure balanced enforcement of both stability conditions, leading to optimal performance.

Our stability framework draws inspiration from prior works on deep equilibrium-based models (8; 9; 10; 11; 12; 13; 14; 15), which proposed similar regularization and hyperparameter tuning techniques. To evaluate the robustness of AROS with respect to these hyperparameters, we conducted extensive ablation studies, holding all components constant while varying the values of $\gamma_1$, $\gamma_2$, and $\gamma_3$. These experiments demonstrate that AROS consistently performs well across a broad range of hyperparameter values, including extreme cases (e.g., $\gamma_1 = 2$). The results of this analysis, presented in Table 13a, confirm the robust performance of AROS under varying hyperparameter configurations.

### A3.4 Time Complexity

The computational complexity of our model is reported in Table 14a.

### A3.5 Discussion on the Fake Generation Strategy

It has been shown that utilizing auxiliary realistic OOD samples is generally effective for improving OOD detection performance. However, this strategy comes with several challenges, as discussed before.

**Table 13a: An ablation study on the hyperparameters** $\gamma_3$ and $\gamma_2$ and $\gamma_1$. Results are reported under clean and $\text{PGD}^{1000}(l_\infty)$ evaluations, measured by AUROC (%). Each table cell presents results in the 'Clean/$\text{PGD}^{1000}$' format.

| Hyperparameter | CIFAR-10 | | | CIFAR-100 | | | ImageNet-1k | | |
|---|---|---|---|---|---|---|---|---|---|
| | CIFAR-100 | SVHN | LSUN | CIFAR-10 | SVHN | LSUN | Texture | iNaturalist | LSUN |
| $\gamma_1 = 1, \gamma_2 = \gamma_3 = 0.05$ (default) | 88.2/80.1 | 93.0/86.4 | 90.6/82.4 | 74.3/67.0 | 81.5/70.6 | 74.3/68.1 | 78.3/69.2 | 84.6/75.3 | 79.4/69.0 |
| $\gamma_1 = 1, \gamma_2 = \gamma_3 = 0.025$ | 86.4/78.2 | 90.4/86.3 | 88.6/80.4 | 74.9/64.1 | 78.8/68.7 | 72.8/68.3 | 79.2/69.7 | 83.3/76.0 | 78.8/67.5 |
| $\gamma_1 = 1, \gamma_2 = \gamma_3 = 0.075$ | 87.5/80.3 | 92.1/87.2 | 88.9/81.6 | 72.7/67.7 | 80.8/68.7 | 73.2/68.8 | 76.4/69.9 | 83.4/73.4 | 80.2/69.9 |
| $\gamma_1 = 1, \gamma_2 = \gamma_3 = 0.1$ | 86.9/78.6 | 93.6/82.1 | 85.8/78.2 | 70.0/65.8 | 80.6/66.2 | 72.6/67.3 | 79.3/69.6 | 82.8/74.0 | 78.9/67.2 |
| $\gamma_1 = 1, \gamma_2 = \gamma_3 = 0.25$ | 86.2/78.7 | 94.3/82.3 | 87.1/77.9 | 70.2/65.2 | 81.3/65.8 | 73.9/66.7 | 78.6/70.1 | 82.8/74.1 | 78.9/68.2 |
| $\gamma_1 = 1, \gamma_2 = \gamma_3 = 0.5$ | 85.2/76.9 | 93.6/81.5 | 85.0/76.0 | 69.7/64.2 | 78.1/64.4 | 72.6/65.2 | 77.0/66.7 | 83.6/74.4 | 78.2/67.3 |
| $\gamma_1 = \gamma_2 = \gamma_3 = 1$ | 84.4/75.3 | 90.5/80.5 | 81.5/73.4 | 69.8/62.2 | 77.5/65.3 | 70.1/64.4 | 73.5/67.3 | 82.7/70.9 | 76.7/65.4 |
| $\gamma_1 = \gamma_2 = \gamma_3 = 0.25$ | 85.4/77.4 | 94.6/82.9 | 86.3/76.9 | 69.9/64.4 | 79.9/65.1 | 74.4/65.3 | 77.2/68.6 | 85.4/76.0 | 79.2/66.8 |
| $\gamma_1 = 0.5, \gamma_2 = \gamma_3 = 0.05$ | 86.0/78.6 | 92.7/82.5 | 86.3/77.7 | 69.7/66.1 | 81.2/65.5 | 70.9/67.1 | 78.8/68.6 | 83.0/73.7 | 78.8/67.1 |
| $\gamma_1 = 1, \gamma_2 = 0.1, \gamma_3 = 0.05$ | 84.0/76.3 | 91.5/81.5 | 81.7/76.4 | 70.1/63.4 | 76.2/63.8 | 69.8/67.7 | 77.4/69.5 | 83.2/71.0 | 78.5/64.8 |
| $\gamma_1 = 1, \gamma_2 = 0.05, \gamma_3 = 0.1$ | 83.5/77.5 | 90.2/78.0 | 81.4/75.1 | 67.2/63.5 | 76.7/66.2 | 67.5/60.4 | 74.3/70.0 | 83.6/74.1 | 74.2/61.7 |
| $\gamma_1 = 2, \gamma_2 = 0.5, \gamma_3 = 0.5$ | 80.4/74.0 | 93.5/80.7 | 81.8/70.9 | 65.2/66.0 | 77.3/60.0 | 65.0/60.1 | 78.4/63.3 | 80.3/70.6 | 71.6/59.5 |

**Table 14a: Time Complexity of Model Steps** for CIFAR-10, CIFAR-100, and ImageNet-1k, measured on a NVIDIA RTX A5000 GPU (on a workstations running Ubuntu 20.04, Intel Core i9-10900X: 10 cores, 3.70 GHz, 19.25 MB cache; within Docker).

| Step | CIFAR-10 | CIFAR-100 | ImageNet-1k |
|---|---|---|---|
| Step 1: Adversarial Training of Classifier | 15 hours | 15 hours | 180 hours |
| Step 2: Crafting Fake OOD Data | 7 hours | 7 hours | 150 hours |
| Step 3: Training with Stability Loss ($\mathcal{L}_{\text{SL}}$) | 8 hours | 8 hours | 100 hours |

First, in certain scenarios, access to an external realistic OOD dataset may not be feasible, and acquiring such data can be challenging. Even when a suitable dataset is available, it must be processed to remove ID concepts to prevent the detector from being misled. This preprocessing step is both time-consuming and computationally expensive. Additionally, studies highlight a potential risk of bias being introduced into the detector when trained on specific auxiliary datasets. Such reliance on a particular realistic dataset may undermine the detector's ability to generalize effectively to diverse OOD samples. These issues become even more pronounced in adversarial training setups, where the complexity of the required data is significantly higher. Motivated by these challenges, this study proposes an alternative strategy that does not depend on the availability of an auxiliary OOD dataset. Notably, our approach is flexible and can incorporate auxiliary OOD datasets as additional information if they are available. To validate this, we conducted an experiment assuming access to a realistic OOD dataset (i.e., Food-101). In this scenario, we computed embeddings of the real OOD samples and used them alongside crafted fake OOD embeddings during training. The results, presented in Table 4 (Setup A), demonstrate improved performance compared to using fake OOD embeddings alone. Furthermore, related studies have shown that in adversarial training, using samples near the decision boundary of the distribution improves robustness by encouraging compact representations. This boundary modeling is critical for enhancing the model's robustness, especially against adversarial attacks that exploit vulnerabilities near the decision boundary. In light of this, our approach shifts focus from generating "realistic" OOD data to estimating low-likelihood regions of the in-distribution. We generate fake "near" OOD data that is close to the ID boundary, which is particularly beneficial for adversarial robustness.

For better intuition regarding usefulness of auxiliary near OOD samples here We will provide a simple example that highlights the effectiveness of near-distribution crafted OOD samples in the adversarial setup. We assume that the feature space is one-dimensional, i.e. $\mathbb{R}$, and the ID class is sampled according to a uniform distribution $U(0, a - \epsilon)$, with $a > 0$, and $\epsilon < a$. We assume that the OOD class is separable with a safety margin of $2\epsilon$ from the ID class to allow a perfectly accurate OOD detector under input perturbations of at most $\epsilon$ at inference. For instance, we let $U(a + \epsilon, b)$ be the feature distribution under the OOD class. The goal is to leverage crafted fake OOD samples to find a robust OOD detector under the $\ell_2$ bounded perturbations of norm $\epsilon$. We assume that the crafted OOD samples data distribution is not perfectly aligned with the anomaly data, e.g. crafted OOD

Figure 6: A depiction in a one-dimensional feature space where the crafted fake OOD samples form a subset of actual OOD data. The gray area represents feasible thresholds separating ID (purple) and fake OOD data (orange). $r$ indicates the shift in exposed fake OOD samples from the real OOD samples. Bold gray lines represent perfect test AUROC thresholds. **Left**: In standard scenarios, perfect thresholds are abundant, even if the fake OOD samples are distant from the ID data. **Middle**: With adversarial training, the feasible thresholds decrease due to the maximum margin constraint, impacting the perfect thresholds. Large deviations in the crafted fake OOD data reduce the set of perfect thresholds. **Right**: For adversarial testing, the overlap between feasible and perfect thresholds narrows to point $a$, emphasizing the importance of near-OOD properties in adversarial contexts.

samples comes from $U(a + r, c)$, with $r \geq \epsilon$. It is evident that the optimal robust decision boundary under infinite training samples that separates the ID and crafted OOD samples would be a threshold $k$ satisfying $a \leq k \leq a + r - \epsilon$. The test adversarial error rate to separate ID and OOD classes is $\frac{1}{2}.\mathbb{I}(k \geq a + \epsilon).\left(\frac{k-a-\epsilon}{b-a-\epsilon} + \frac{\min(k+\epsilon,b)-k}{b-a-\epsilon}\right) + \frac{1}{2}.\mathbb{I}(a < k < a + \epsilon)\frac{\min(k+\epsilon,b)-a-\epsilon}{b-a-\epsilon}$, assuming that the classes are equally probable a prior. It is obvious the adversarial error rate would be zero for $k = a$. But otherwise, if $k \geq a + \epsilon$ the classifier incurs classification error in two cases; in intervals $(a + \epsilon, k)$ (even without any attack), and $(k, \min(k + \epsilon, b))$ in which a perturbation of $-\epsilon$ would cause classification error. Also if $a < k < a + \epsilon$, classification error only happens at $(a + \epsilon, \min(k + \epsilon, b))$. Now, for the crafted OOD samples to be near-distribution, $r \to \epsilon$, which forces $k$ to be $a$ in the training, and makes the test adversarial error zero. Otherwise, the adversarial error is proportional to $k$, for $k$ being distant from $b$. Therefore, in the worst case, if $k = a + r - \epsilon$, we get an adversarial error proportional to $r$. As a result, minimizing $r$, which makes the crafted OOD samples near-distribution, would be an effective strategy in making the adversarial error small. Refer to the figure 6 for further intuition and clarity.

To provide a more practical intuition about our crafted fake OOD samples, we present t-SNE visualizations of the embedding space. These visualizations demonstrate that the crafted fake data are positioned near the ID samples and effectively cover their boundary. Please refer to Figures 4 and 5.

To further demonstrate the superiority of our strategy for crafting fake OOD samples—a simple yet effective technique—we conducted an ablation study by replacing our proposed method with alternative approaches. Please refer to Table 15a for the results.

**Table 15a: Ablation study on different fake OOD crafting strategies.** Results are measured by AUROC (%). The perturbation budget $\epsilon$ is set to $\frac{8}{255}$ for low-resolution datasets and $\frac{4}{255}$ for high-resolution datasets. Table cells present results in the 'Clean/PGD$^{1000}$' format. We evaluate alternative OOD data synthesis strategies, such as random Gaussian and uniform noise, while keeping other components fixed.

| Fake Crafting | CIFAR-10 | | | CIFAR-100 | | | ImageNet-1k | | |
|---|---|---|---|---|---|---|---|---|---|
| Strategy | CIFAR-100 | SVHN | LSUN | CIFAR-10 | SVHN | LSUN | Texture | iNaturalist | LSUN |
| *AROS* | 88.2/80.1 | 93.0/86.4 | 90.6/82.4 | 74.3/67.0 | 81.5/70.6 | 74.3/68.1 | 78.3/69.2 | 84.6/75.3 | 79.4/69.0 |
| Random Gaussian Noise | 85.3/76.5 | 89.4/78.1 | 82.5/77.1 | 70.5/61.3 | 74.4/62.5 | 71.5/64.6 | 76.1/67.4 | 81.3/72.7 | 75.8/67.3 |
| Random Uniform Noise | 81.2/74.3 | 87.1/79.5 | 84.8/75.4 | 65.6/59.6 | 73.8/63.5 | 65.8/60.1 | 70.0/63.8 | 78.8/67.3 | 71.6/62.8 |

### A3.6 ORTHOGONAL BINARY LAYER

To demonstrate the superiority of the orthogonal binary layer used in our pipeline, we conducted an ablation study. In this study, we fixed all other components and replaced the orthogonal binary layer with a regular fully connected (FC) layer, then compared the results across different budget levels. The results of this comparison are presented in Table 16a. Additionally, we visualized the embedding space corresponding to the regular FC layer and the orthogonal binary layer in Figure 7.

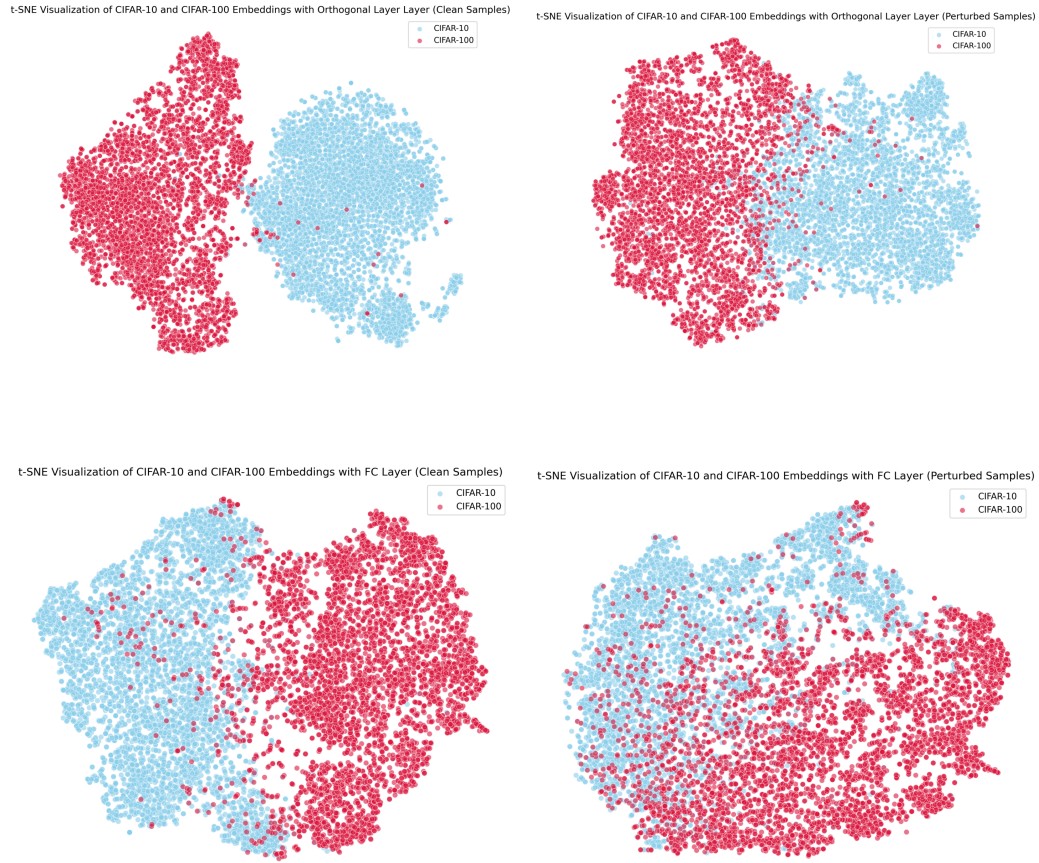

Figure 7: t-SNE visualizations of CIFAR-10 (ID) and CIFAR-100 (OOD) embeddings, illustrating the effect of an orthogonal binary layer compared to a regualr fully connected (FC) layer under clean and perturbed sample scenarios. Top row: Embeddings with the orthogonal binary layer for clean (left) and perturbed (right) samples, showing enhanced ID-OOD separation. Bottom row: Embeddings with the FC layer for clean (left) and perturbed (right) samples, demonstrating reduced separation in adversarial settings. These results highlight the orthogonal binary layer's effectiveness in preserving ID-OOD distinction, in both clean and adversarial conditions.

**Table 16a:** Comparison of performance between the orthogonal binary layer (Ortho.) and the regular fully connected (FC) layer across different datasets and perturbation levels ($\epsilon$). The results demonstrate that the orthogonal binary layer consistently outperforms the regular FC layer in both clean ($\epsilon = \frac{0}{255}$) and adversarial scenarios, with varying levels of perturbation ($\epsilon = \frac{2}{255}, \frac{4}{255}, \frac{8}{255}$).

| $\epsilon$ | Methods | CIFAR-10 | | | CIFAR-100 | | | ImageNet-1k | | |
|---|---|---|---|---|---|---|---|---|---|---|
| | | CIFAR-100 | SVHN | LSUN | CIFAR-10 | SVHN | LSUN | Texture | iNaturalist | LSUN |
| $\frac{0}{255}$ (Clean) | FC | 85.6 | 88.2 | 87.5 | 66.9 | 78.4 | 71.5 | 75.4 | 79.8 | 76.1 |
| | Ortho. | 88.2 | 93.0 | 90.6 | 74.3 | 81.5 | 74.3 | 78.3 | 84.0 | 79.4 |
| $\frac{2}{255}$ | FC | 82.0 | 84.9 | 84.1 | 63.6 | 75.8 | 68.6 | 73.2 | 76.9 | 72.2 |
| | Ortho. | 85.5 | 90.7 | 88.8 | 72.2 | 79.9 | 72.1 | 75.7 | 82.7 | 76.6 |
| $\frac{4}{255}$ | FC | 75.2 | 78.1 | 80.3 | 57.9 | 71.6 | 63.5 | 68.2 | 72.9 | 66.7 |
| | Ortho. | 83.1 | 87.5 | 85.2 | 69.5 | 74.5 | 70.4 | 71.6 | 78.3 | 68.0 |
| $\frac{8}{255}$ | FC | 67.3 | 74.6 | 79.5 | 57.1 | 63.3 | 62.4 | 60.7 | 70.2 | 65.1 |
| | Ortho. | 80.1 | 86.4 | 82.4 | 67.0 | 70.6 | 68.1 | 69.2 | 75.3 | 69.0 |

These visualizations highlight the enhanced separability provided by the orthogonal binary layer in both clean and adversarial scenarios.

The orthogonal binary layer $B_\eta$ is designed to apply a transformation to the NODE output $h_\phi(z)$, where the weights $W$ of the layer are constrained to be orthogonal ($W^T W = I$). This constraint encourages maximal separation between the equilibrium points of ID and OOD data by ensuring that the learned representations preserve distinct directions in the feature space. Specifically, the layer operates as:

$$B_\eta(z) = Wz + b, \text{ subject to } W^T W = I,$$

where $W$ represents the weight matrix, $b$ is the bias term, and the orthogonality constraint is enforced during training using a regularization term.

To clarify its role in $L_{SL}$, we now explicitly annotate $B_\eta$ as the mapping responsible for projecting the NODE output $h_\phi(z)$ into a binary classification space (ID vs. OOD).

The $L_{SL}$ with explicit reference to $W$ and $b$ would be as follows:

$$L_{SL} = \min_{\phi, w} \frac{1}{|X_{\text{train}}|} \Bigg( \ell_{\text{CE}}((W h_\phi(X_{\text{train}}) + b), y) + \gamma_1 \| \|h_\phi(X_{\text{train}})\| \|_2$$

$$+ \gamma_2 \exp\Bigg( -\sum_{i=1}^{n} [\nabla h_\phi(X_{\text{train}})]_{ii} \Bigg)$$

$$+ \gamma_3 \exp\Bigg( \sum_{i=1}^{n} (-|[\nabla h_\phi(X_{\text{train}})]_{ii}|$$

$$+ \sum_{j \neq i} |[\nabla h_\phi(X_{\text{train}})]_{ij}|) \Bigg) \Bigg)$$

where the orthogonality constraint $W^T W = I$ is enforced via regularization during optimization.

## A4 Details of Evaluation and Experimental Setup

OOD detection can be framed as a binary classification task, where training samples are confined to a single set (the ID data), and during testing, input samples from OOD set must be identified. A key challenge is that OOD data is not as clearly defined as ID data; any semantic content absent from the ID distribution is considered OOD. The primary objective in OOD detection is to develop a pipeline capable of assigning meaningful OOD scores to input samples, where a higher score suggests that the model perceives the input as having a greater likelihood of being OOD. An optimal OOD detector would produce score distributions for OOD and ID samples that are fully separated, with no overlap, ensuring clear differentiation between the two.

Formally, the OOD detection decision can be performed as follows:

$$G_\zeta(\text{x}) = \begin{cases} \text{ID} & \text{if } S_\mathcal{F}(x) \leq \zeta \\ \text{OOD} & \text{if } S_\mathcal{F}(x) > \zeta \end{cases},$$

where $\zeta$ is a threshold parameter.

Thus, the scoring function $S_\mathcal{F}$ is central to the performance of the OOD detector. In the setup of adversarial evaluation, PGD and other attack methods, we adversarially target $S_\mathcal{F}$ and perturb the inputs as described in Section 2. Intuitively, these attacks aim to shift ID samples closer to OOD and vice versa. This approach is fair and consistent with existing OOD detection frameworks, as all such methods define an OOD score function. Furthermore, this adversarial strategy has been previously explored in related work. In the following, we will provide details on AutoAttack and adaptive AutoAttack in our evaluation.

**AutoAttack.** AutoAttack is an ensemble of six attack methods: APGD with Cross-Entropy loss, APGD with Difference of Logits Ratio (DLR) loss, APGDT, FAB (125), multi-targeted FAB (126), and Square Attack (127). However, the DLR loss-based attacks assume that the target model is a classifier trained on more than two classes. In OOD detection, the problem is more akin to binary classification, as we are only distinguishing between OOD and ID classes. Therefore, we excluded these specific DLR-based attacks in our adaptation of AutoAttack and instead used an ensemble of the remaining attacks.

**Adaptive AA.** We also employed Adaptive AutoAttack (AA) for evaluation, an attack framework designed to efficiently and reliably approximate the lower bound of a model's robustness. Adaptive AA integrates two main strategies: Adaptive Direction Initialization to generate better starting points for adversarial attacks, and Online Statistics-Based Discarding to prioritize attacking easier-to-perturb images. Unlike standard AutoAttack, Adaptive AA adapts its attack directions based on the model's specific defense properties and dynamically allocates iterations to improve the efficiency of successful attacks. For the perturbation budget, we used a similar budget to that considered for PGD, and for other hyperparameters, we used their default values. For the implementation of adversarial attacks, we utilized the `Torchattacks` library in PyTorch.

### A4.1 DETAILS OF THE REPORTED RESULTS

To report and evaluate previous works, we reproduced their results using the official GitHub repositories. For the clean evaluation setup, if our reproduced results differed from the originally reported values, we reported the higher value. If no results were available, we presented our reproduced results. A similar approach was taken for adversarial evaluation: if their evaluation details and considered datasets matched ours, we used their reported results. However, if the attacks details in their evaluation differed or if they did not consider our datasets or benchmarks, we reported our reproduced results.

### A4.2 METRICS

Our main results are reported using AUROC for comparison across methods, as it is more commonly utilized in the detection literature. However, in Table 4b, we also compare our method's performance using other metrics, including AUPR and FPR95. Below, we provide an explanation of each of these metrics.

**AUROC.** The Area Under the Receiver Operating Characteristic Curve (AUROC) is a metric that evaluates classification performance by measuring the trade-off between the True Positive Rate (TPR) and the False Positive Rate (FPR) across various threshold settings. TPR, also known as recall or sensitivity, measures the proportion of actual positives correctly identified, while FPR measures the proportion of actual negatives incorrectly classified as positives. The AUROC, which ranges from 0.5 (random classifier) to 1.0 (perfect classifier), provides an aggregate performance measure.

**AUPR.** The Area Under the Precision-Recall Curve (AUPR) is particularly useful for imbalanced datasets where the positive class is rare. The AUPR captures the trade-off between precision (the proportion of predicted positives that are actual positives) and recall. A high AUPR indicates a model with both high precision and recall, providing valuable insights in scenarios where the dataset is skewed.

**FPR95.** False Positive Rate at 95% True Positive Rate (FPR95), which assesses the model's ability to correctly identify ID samples while rejecting OOD samples. Specifically, FPR95 is the false positive rate when the true positive rate is set to 95%, indicating how often OOD samples are misclassified as ID. Lower values of FPR95 indicate better OOD detection capabilities, as the model more accurately differentiates between ID and OOD samples. Together, these metrics offer a comprehensive understanding of model performance and robustness in classification and OOD detection tasks.

## A5 ADDITIONAL DETAILS ABOUT THE DATASETS

In our experiments, we considered several datasets. Our perturbation budget is determined based on the image size of the ID training set. It is set to $\frac{4}{255}$ for scenarios where the ID training set consists of

high-resolution images (e.g., ImageNet-1K), and $\frac{8}{255}$ for high-resolution datasets (e.g., CIFAR-10 and CIFAR-100). In all experiments, the ID and OOD datasets contain disjoint semantic classes. If there is an overlap, we exclude those semantics from the OOD test set. In all experiments, the ID and OOD datasets contain disjoint semantic classes. If there is an overlap, we exclude those semantics from the OOD test set. In the following, we provide a brief explanation of the datasets used.

**CIFAR-10 and CIFAR-100.** are benchmark datasets commonly used for image classification tasks. CIFAR-10 consists of 60,000 color images of size $32 \times 32$ pixels across 10 classes, with 6,000 images per class. CIFAR-100 is similar but contains 100 classes with 600 images each, providing a more fine-grained classification challenge.

**ImageNet-1k.** ImageNet-1K contains 1,281,167 training images, 50,000 validation images and 100,000 test images. This dataset was compiled to facilitate research in computer vision by providing a vast range of images to develop and test algorithms, particularly in the areas of object recognition, detection, and classification.

**Texture Dataset.** Texture Dataset is designed for studying texture recognition in natural images. It comprises diverse textures captured in the wild, enabling research on classifying and describing textures under varying conditions.

**Street View House Numbers (SVHN).** SVHN is a real-world image dataset for developing machine learning and object recognition algorithms with minimal data preprocessing. It contains over 600,000 digit images obtained from house numbers in Google Street View images.

**iNaturalist.** The iNaturalist Species Classification and Detection Dataset aims to address real-world challenges in computer vision by focusing on large-scale, fine-grained classification and detection. This dataset, consisting of 859,000 images from over 5,000 different species of plants and animals, is noted for its high class imbalance and the visual similarity of species within its collection. The images are sourced globally, contributed by a diverse community through the iNaturalist platform, and verified by multiple citizen scientists.

**Places365.** Places365 is a scene-centric dataset containing over 10 million images spanning 365 scene categories. It is designed for scene recognition tasks and aids in understanding contextual information in images.

**Large-scale Scene Understanding (LSUN).** The Large-scale Scene Understanding (LSUN) challenge is designed to set a new standard for large-scale scene classification and comprehension. It features a classification dataset that includes 10 scene categories, such as dining rooms, bedrooms, conference rooms, and outdoor churches, among others. Each category in the training dataset comprises an extensive range of images, from approximately 120,000 to 3,000,000 images. The dataset also provides 300 images per category for validation and 1,000 images per category for testing.

**iSUN.** The iSUN dataset is a large-scale eye tracking dataset that leverages natural scene images from the SUN database. It was specifically developed to address the limitations of small, in-lab eye tracking datasets by enabling large-scale data collection through a crowdsourced, webcam-based eye tracking system deployed on Amazon Mechanical Turk.

**MNIST.** The MNIST dataset (Modified National Institute of Standards and Technology dataset) is a large collection of handwritten digits that is widely used for training and testing in the field of machine learning. It contains 70,000 images of handwritten digits from 0 to 9, split into a training set of 60,000 images and a test set of 10,000 images. Each image is a 28x28 pixel grayscale image.

**Fashion-MNIST (FMNIST).** FMNIST dataset is a collection of article images designed to serve as a more challenging replacement for the traditional MNIST dataset. It consists of 70,000 grayscale images divided into 60,000 training samples and 10,000 test samples, each image having a resolution of 28x28 pixels. The dataset contains 10 different categories of fashion items such as T-shirts/tops, trousers, pullovers, dresses, coats, sandals, shirts, sneakers, bags, and ankle boots.

**Imagenette.** Imagenette is a subset of ImageNet consisting of ten easily classified classes. It was released to encourage research on smaller datasets that require less computational resources, facilitating experimentation and algorithm development.

**CIFAR-10-C and CIFAR-100-C**. CIFAR-10-C and CIFAR-100-C are corrupted versions of CIFAR-10 and CIFAR-100, respectively. They include various common corruptions and perturbations such

as noise, blur, and weather effects, used to evaluate the robustness of image classification models against real-world imperfections.

**ADNI Neuroimaging Dataset.** The Alzheimer's Disease Neuroimaging Initiative (ADNI) dataset is a large-scale collection of neuroimaging data aimed at tracking the progression of Alzheimer's disease. In our study, we categorize the dataset into six classes based on cognitive status and disease progression. The classes CN (Cognitively Normal) and SMC (Subjective Memory Concerns) are designated as ID, representing individuals without significant neurodegenerative conditions or with only minor memory concerns. The other four classes—AD (Alzheimer's Disease), MCI (Mild Cognitive Impairment), EMCI (Early Mild Cognitive Impairment), and LMCI (Late Mild Cognitive Impairment)—are treated as OOD, encompassing various stages of cognitive decline related to Alzheimer's. The dataset contains 6,000 records for each class, providing a comprehensive representation of the spectrum of cognitive health and Alzheimer's disease. A summary of each class is as follows:

- **AD**: Patients diagnosed with Alzheimer's disease, showing significant cognitive decline.
- **CN**: Individuals with normal cognitive functioning, serving as a control group.
- **MCI**: A stage of cognitive decline that lies between normal aging and more advanced impairment, often preceding Alzheimer's disease.
- **EMCI**: Patients with early signs of mild cognitive impairment, indicating the initial onset of cognitive issues.
- **LMCI**: Patients with symptoms indicative of more advanced mild cognitive impairment, closer to Alzheimer's in severity.
- **SMC**: Individuals reporting memory concerns but performing normally on cognitive assessments.

We used the middle slice of MRI scans from ADNI phases 1, 2, and 3. We should note that the ADNI dataset was the only one we split into ID and OOD using the strategy mentioned earlier. For the other datasets in our OSR experiments, we randomly divided each dataset into ID and OOD at a ratio of 0.6 and 0.4, respectively.

### A5.1   DETAILS ON DATASET CORRUPTIONS

**Corruptions**: The first type of corruption is Gaussian noise, which commonly appears in low-light conditions. Shot noise, also known as Poisson noise, results from the discrete nature of light and contributes to electronic noise. Impulse noise, similar to salt-and-pepper noise but in color, often arises due to bit errors. Defocus blur happens when an image is out of focus. Frosted glass blur is caused by the appearance of "frosted glass" on windows or panels. Motion blur occurs due to rapid camera movement, while zoom blur happens when the camera moves swiftly towards an object. Snow is an obstructive form of precipitation, and frost occurs when lenses or windows accumulate ice crystals. Fog obscures objects and is often simulated using the diamond-square algorithm. Brightness is affected by the intensity of daylight, and contrast levels change based on lighting conditions and the object's color. Elastic transformations distort small image regions by stretching or contracting them. Pixelation results from upscaling low-resolution images. Lastly, JPEG compression, a lossy format, introduces noticeable compression artifacts.

## A6   LIMITATIONS AND FUTURE RESEARCH

We considered classification image-based benchmarks, that while difficult, do not cover the full breadth of real-world attacks in situations such as more complex open-set images or video, i.e., time-series data. Future work should test our proposed method in video streaming data (time-series), as inherently we leverage stability properties in dynamical systems that could be very attractive for such settings. Namely, because video frames are often highly correlated, one can even leverage the prior stability points in time in order to make better future predictions. This could also lower the compute time.

Moreover, moving to tasks beyond classification could be very attractive. Concretely, one example is pose estimation. This is a keypoint detection task that can often be cast as a panoptic segmentation

task, where each keypoint needs to be identified in the image and grouped appropriately (128; 129). By using AROS, one could find OOD poses could not only improve data quality, but alert users to wrongly annotated data. Another example is in brain decoding where video frame or scene classification is critical, and diffusion models are becoming an attractive way to leverage generative models (130). Given AROS ability in OOD detection, this could be smartly used to correct wrong predictions. In summary, adapting AROS to these other data domains could further extend its applicability.

## A7   THEORETICAL INSIGHT AND BACKGROUND

In this section, we provide additional background on the theorems utilized in the main text. The structure of this background section is inspired by (131). Specifically, we present proofs for the theorems referenced in the main text, including the Hartman-Grobman Theorem, and offer a theoretical justification for our proposed objective function, $\mathcal{L}_{\text{SL}}$. Additionally, we include Figures 8 and 9 as empirical support for the proposed loss function. The first figure illustrates how $\mathcal{L}_{\text{SL}}$ ensures that the real parts of the eigenvalues become negative, indicating stability, while the second figure highlights the stable decrease of the proposed loss function throughout the training process. Inspired by previous works leveraging control theory in deep learning, we set $T = 5$ as the integration time for the neural ODE layer and employ the Runge-Kutta method of order 5 as the solver. This choice ensures a balance between computational efficiency and robustness, allowing the ODE dynamics to stabilize feature representations effectively and mitigate the impact of adversarial perturbations.

### A7.1   BACKGROUND

The Hartman-Grobman Theorems is among the most powerful tools in dynamical systems (132). The Hartman-Grobman theorem allows us to depict the local phase portrait near certain equilibria in a nonlinear system using a similar, simpler linear system derived by computing the system's Jacobian matrix at the equilibrium point.

**Why does linearization at fixed points reveal behavior around the fixed point?**

For an $n$-dimensional linear system of differential equations ($\dot{\mathbf{x}} = A\mathbf{x}$) with a fixed point at the origin, we can classify behaviors such as saddle points, spirals, cycles, stars, and nodes based on the eigenvalues of the matrix $A$. These behaviors are well-understood in the linear case. However, for nonlinear systems, analyzing the behavior becomes more challenging. Fortunately, the situation is not entirely intractable. By calculating the Jacobian matrix, or "total derivative," $J$, of the system and evaluating it at the fixed point, we obtain a linear approximation characterized by the matrix $J$. The Hartman-Grobman theorem states that, within a neighborhood of the fixed point, if all eigenvalues of $J$ have nonzero real parts, we can infer qualitative properties of the solutions to the nonlinear system. These include whether trajectories converge to or diverge from the equilibrium point and whether they spiral or behave like a node.

## Definitions

### Definition 7.1: Homeomorphism

A function $h : X \rightarrow Y$ is called a homeomorphism between $X$ and $Y$ if it is a continuous bijection (both one-to-one and onto) with a continuous inverse (denoted $h^{-1}$). The existence of a homeomorphism implies that $X$ and $Y$ have analogous structures, as $h$ and $h^{-1}$ preserve the neighborhood relationships of points. Topologists often describe this concept as a process of stretching and bending without tearing.

### Definition 7.2: Topological Conjugacy

Consider two maps $f : X \rightarrow X$ and $g : Y \rightarrow Y$. A map $h : X \rightarrow Y$ is called a topological semi-conjugacy if it is continuous, onto, and satisfies $h \circ f = g \circ h$, where $\circ$ denotes function composition (sometimes written as $h(f(\mathbf{x})) = g(h(\mathbf{x}))$ for $\mathbf{x}$ in $X$). Furthermore, $h$ is a topological conjugacy if it is a homeomorphism between $X$ and $Y$ (i.e., $h$ is also one-to-one and has a continuous inverse). In this case, $X$ and $Y$ are said to be homeomorphic.

### Definition 7.3: Hyperbolic Fixed Point

A hyperbolic fixed point of a system of differential equations is a point where all eigenvalues of the Jacobian evaluated at that point have nonzero real parts.

### Definition 7.4: Cauchy Sequence

For the purposes of this document, we will provide a non-technical definition. A Cauchy sequence of functions is a series $x_k = x_1, x_2, \ldots$ such that the functions become increasingly similar as $k \to \infty$.

### Definition 7.5: Flow

Let $\dot{\mathbf{x}} = F(\mathbf{x})$ be a system of differential equations with initial condition $\mathbf{x}_0$. Provided that the solutions exist and are unique (conditions given by the existence and uniqueness theorem; see, for example, ((133), pg. 149), the flow $\phi(t; \mathbf{x}_0)$ of $F(\mathbf{x})$ provides the spatial solution over time starting from $\mathbf{x}_0$. An important property of flows is that small changes in initial conditions in phase space lead to continuous changes in flows, due to the continuity of the vector field in $\mathbb{R}^n$.

### Definition 7.6: Orbit/Trajectory

The set of all points in the flow $\phi(t; \mathbf{x}_0)$ for the differential equations $\dot{\mathbf{x}} = F(\mathbf{x})$ is called the "orbit" or "trajectory" of $F(\mathbf{x})$ with initial condition $\mathbf{x}_0$. We denote the orbit as $\phi(\mathbf{x}_0)$. When considering only $t \geq 0$, we refer to the "forward orbit" or "forward trajectory."

### A7.2   THEOREM AND PROOF

**Theorem 7.1 The Hartman-Grobman Theorem**  *Let $\mathbf{x} \in \mathbb{R}^n$. Consider the nonlinear system $\dot{\mathbf{x}} = f(\mathbf{x})$ with flow $\phi_t$ and the linear system $\dot{\mathbf{x}} = A\mathbf{x}$, where $A$ is the Jacobian $Df(\mathbf{x}^*)$ of $f$ at a hyperbolic fixed point $\mathbf{x}^*$. Assume that we have appropriately shifted $\mathbf{x}^*$ to the origin, i.e., $\mathbf{x}^* = \mathbf{0}$.*

*Suppose $f$ is $C^1$ on some $E \subset \mathbb{R}^n$ with $\mathbf{0} \in E$. Let $I_0 \subset \mathbb{R}$, $U \subset \mathbb{R}^n$, and $V \subset \mathbb{R}^n$ be neighborhoods containing the origin. Then there exists a homeomorphism $H : U \to V$ such that, for all initial points $\mathbf{x}_0 \in U$ and all $t \in I_0$,*

$$H \circ \phi_t(\mathbf{x}_0) = e^{At} H(\mathbf{x}_0).$$

*Thus, the flow of the nonlinear system is homeomorphic to the flow $e^{At}$ of the linear system given by the fundamental theorem for linear systems.*

**Proof**

Essentially, this theorem states that the nonlinear system $\dot{\mathbf{x}} = f(\mathbf{x})$ is locally homeomorphic to the linear system $\dot{\mathbf{x}} = A\mathbf{x}$. To prove this, we begin by expressing $A$ as the matrix

$$\begin{pmatrix} P & 0 \\ 0 & Q \end{pmatrix}$$

where $P$ and $Q$ are sub-matrices of $A$ such that the real parts of the eigenvalues of $P$ are negative, and those of $Q$ are positive. Finding such a matrix $A$ may require finding a new basis for our linear system using linear algebra techniques. For more details, see section 1.8 on Jordan forms of matrices in Perko (134).

Consider the solution $\mathbf{x}(t, \mathbf{x}_0) \in \mathbb{R}^n$ given by

$$\mathbf{x}(t, \mathbf{x}_0) = \phi_t(\mathbf{x}) = \begin{pmatrix} \mathbf{y}(t, \mathbf{y}_0, \mathbf{z}_0) \\ \mathbf{z}(t, \mathbf{y}_0, \mathbf{z}_0) \end{pmatrix}$$

with $\mathbf{x}_0 \in \mathbb{R}^n$ given by Consider the solution:

$$\mathbf{x}(t, \mathbf{x}_0) = \phi_t(\mathbf{x}) = \begin{pmatrix} \mathbf{y}(t, \mathbf{y}_0, \mathbf{z}_0) \\ \mathbf{z}(t, \mathbf{y}_0, \mathbf{z}_0) \end{pmatrix}$$

where $\mathbf{x}_0 \in \mathbb{R}^n$ is given by

$$\mathbf{x}_0 = \begin{pmatrix} \mathbf{y}_0 \\ \mathbf{z}_0 \end{pmatrix}$$

with $\mathbf{y}_0 \in E^S$ (the stable subspace of $A$) and $\mathbf{z}_0 \in E^U$ (the unstable subspace of $A$). The stable and unstable subspaces of $A$ are defined as the spans of the eigenvectors of $A$ corresponding to eigenvalues with negative and positive real parts, respectively. Define

$$\widetilde{\mathbf{Y}}(\mathbf{y}_0, \mathbf{z}_0) = \mathbf{y}(1, \mathbf{y}_0, \mathbf{z}_0) - e^P \mathbf{y}_0,$$
$$\widetilde{\mathbf{Z}}(\mathbf{y}_0, \mathbf{z}_0) = \mathbf{z}(1, \mathbf{y}_0, \mathbf{z}_0) - e^Q \mathbf{z}_0.$$

Here, $\widetilde{\mathbf{Y}}$ and $\widetilde{\mathbf{Z}}$ are functions of the trajectory with initial condition $\mathbf{x}_0$ evaluated at $t = 1$. If $\mathbf{x}_0 = \mathbf{0}$, then $\mathbf{y}_0 = \mathbf{z}_0 = \mathbf{0}$, leading to $\widetilde{\mathbf{Y}}(\mathbf{0}) = \widetilde{\mathbf{Z}}(\mathbf{0}) = \mathbf{0}$ and thus $D\widetilde{\mathbf{Y}}(\mathbf{0}) = D\widetilde{\mathbf{Z}}(\mathbf{0}) = \mathbf{0}$ since $\mathbf{x}_0$ is at the fixed point $\mathbf{0}$. Since $f$ is $C^1$ on $E$, it follows that $\widetilde{\mathbf{Y}}$ and $\widetilde{\mathbf{Z}}$ are also $C^1$ on $E$. Knowing that $D\widetilde{\mathbf{Y}}$ and $D\widetilde{\mathbf{Z}}$ are zero at the origin and that $\widetilde{\mathbf{Y}}$ and $\widetilde{\mathbf{Z}}$ are continuously differentiable, we can define a region around the origin such that $\|\mathbf{y}_0\|^2 + \|\mathbf{z}_0\|^2 \leq s_0^2$ for some sufficiently small $s_0 \in \mathbb{R}$, where the norms of $D\widetilde{\mathbf{Y}}$ and $D\widetilde{\mathbf{Z}}$ are each less than some real number $a$:

$$\|D\widetilde{\mathbf{Y}}(\mathbf{y}_0, \mathbf{z}_0)\| \leq a,$$
$$\|D\widetilde{\mathbf{Z}}(\mathbf{y}_0, \mathbf{z}_0)\| \leq a.$$

We now apply the mean value theorem: Let $Y$ and $Z$ be smooth functions such that $Y = Z = 0$ when $\|\mathbf{y}_0\|^2 + \|\mathbf{z}_0\|^2 \geq s_0^2$, and $Y = \widetilde{Y}$ and $Z = \widetilde{Z}$ when $\|\mathbf{y}_0\|^2 + \|\mathbf{z}_0\|^2 \leq (s_0^2/2)$. Then the mean value theorem gives us

$$|Y| \leq a\sqrt{\|\mathbf{y}_0\|^2 + \|\mathbf{z}_0\|^2} \leq a(\|\mathbf{y}_0\| + \|\mathbf{z}_0\|),$$
$$|Z| \leq a\sqrt{\|\mathbf{y}_0\|^2 + \|\mathbf{z}_0\|^2} \leq a(\|\mathbf{y}_0\| + \|\mathbf{z}_0\|).$$

Let $B = e^P$ and $C = e^Q$. With proper normalization (see (132)), we have $b = \|B\| < 1$ and $c = \|C^{-1}\| < 1$. We will now prove the existence of a homeomorphism $H$ from $U$ to $V$ satisfying $H \circ T = L \circ H$ using the method of successive approximations. Define the transformations $L, T$, and $H$ as follows:

$$L(\mathbf{y}, \mathbf{z}) = \begin{pmatrix} B\mathbf{y} \\ C\mathbf{z} \end{pmatrix} = e^{A\mathbf{x}}, \tag{7.1}$$

$$T(\mathbf{y}, \mathbf{z}) = \begin{pmatrix} B\mathbf{y} + Y(\mathbf{y}, \mathbf{z}) \\ C\mathbf{z} + Z(\mathbf{y}, \mathbf{z}) \end{pmatrix}, \tag{7.2}$$

$$H(\mathbf{x}) = \begin{pmatrix} \Phi(\mathbf{y}, \mathbf{z}) \\ \Psi(\mathbf{y}, \mathbf{z}) \end{pmatrix}. \tag{7.3}$$

From equations (2.1)-(2.3) and our desired relation $H \circ T = L \circ H$, we obtain

$$B\Phi = \Phi(B\mathbf{y} + Y(\mathbf{y}, \mathbf{z}), C\mathbf{z} + Z(\mathbf{y}, \mathbf{z})),$$
$$C\Psi = \Psi(B\mathbf{y} + Y(\mathbf{y}, \mathbf{z}), C\mathbf{z} + Z(\mathbf{y}, \mathbf{z})).$$

We define successive approximations for $\Psi$ recursively by

$$\Psi_0 = \mathbf{z}, \tag{7.4}$$

$$\Psi_{k+1} = C^{-1}\Psi_k(B\mathbf{y} + Y(\mathbf{y}, \mathbf{z}), C\mathbf{z} + Z(\mathbf{y}, \mathbf{z})), \quad k \in \mathbb{N}_0. \tag{7.5}$$

This implies that we can increasingly approximate the function $\Phi$ by following the recursion relations defined in equations (7.4)-(7.5). By induction, it follows that all $\Psi_k$ are continuous because the flow $\phi_t$ is continuous, which means $\Psi_0$ is continuous. Since $C^{-1}$ is continuous, $\Psi_1$ is also continuous, and by induction, $\Psi_k$ is continuous for all $k \in \mathbb{N}_0$. Additionally, it follows that $\Psi_k(\mathbf{y}, \mathbf{z}) = \mathbf{z}$ whenever $|\mathbf{y}| + |\mathbf{z}| \geq 2s_0$ (134).

It can be shown by induction (134) that

$$|\Psi_j(\mathbf{y}, \mathbf{z}) - \Psi_{j-1}(\mathbf{y}, \mathbf{z})| \leq Mr^j(|\mathbf{y}| + |\mathbf{z}|)^\sigma$$

where $j = 1, 2, \ldots, r = c[2\max(a, b, c)]^\sigma$, $c < 1$, and $\sigma \in (0, 1)$ such that $r < 1$. This leads to the conclusion that $\Psi_k(\mathbf{y}, \mathbf{z})$ forms a Cauchy sequence of continuous functions. These functions converge uniformly as $k \to \infty$, and we denote the limiting function by $\Psi(\mathbf{y}, \mathbf{z})$. As with the $\Psi_k$, it holds that $\Psi(\mathbf{y}, \mathbf{z}) = \mathbf{z}$ for $|\mathbf{y}| + |\mathbf{z}| \geq 2s_0$.

A similar argument applies for $B\Phi = \Phi(B\mathbf{y} + Y(\mathbf{y}, \mathbf{z}), C\mathbf{z} + Z(\mathbf{y}, \mathbf{z}))$, which can be rewritten as $B^{-1}\Phi(\mathbf{y}, \mathbf{z}) = \Phi(B^{-1}\mathbf{y} + Y_1(\mathbf{y}, \mathbf{z}), C^{-1}\mathbf{z} + Z_1(\mathbf{y}, \mathbf{z}))$, where $T^{-1}$ defines $Y_1$ and $Z_1$ as follows:

$$T^{-1}(\mathbf{y}, \mathbf{z}) = \begin{pmatrix} B^{-1}\mathbf{y} + Y_1(\mathbf{y}, \mathbf{z}) \\ C^{-1}\mathbf{z} + Z_1(\mathbf{y}, \mathbf{z}) \end{pmatrix}.$$

We can then solve for $\Phi$ in the same manner as we solved for $\Psi$ earlier, starting with $\Phi_0 = \mathbf{y}$. After performing the calculations necessary to find $\Psi$ and $\Phi$, we obtain the homeomorphism $H : \mathbb{R}^n \to \mathbb{R}^n$ given by

$$H = \begin{pmatrix} \Phi \\ \Psi \end{pmatrix}. \tag{7.6}$$

### A7.3 Analysis and Justification of $\mathcal{L}_{\text{SL}}$

Consider a nonautonomous initial value ODE problem: $\frac{d\mathbf{z}(t)}{dt} = \mathbf{h}(\mathbf{z}(t), t), t \geq t_0; \quad \mathbf{z}(t_0) = \mathbf{z}_0$.

Let $\mathbf{s}(\mathbf{z}_0, t_0, t)$ denote the solution of the ODE corresponding to the initial input $\mathbf{z}_0$ at time $t_0$.

**Definition 7.7** *(Equilibrium (135))* A vector $\mathbf{x}^*$ is called an equilibrium of a system if $\mathbf{h}(\mathbf{x}^*, t) = 0, \forall t \geq 0$.

**Definition 7.8** *(Stability (136))* A constant vector $\mathbf{x}^* \in \mathbb{R}^d$ is a stable equilibrium point for a system if, for every $\epsilon > 0$ and every $t_0 \in \mathbb{R}^+$, there exists $\delta(\epsilon, t_0)$ such that for each $\mathbf{z}_0 \in B_\delta(\mathbf{x}^*)$, it holds that $\|\mathbf{s}(\mathbf{z}_0, t_0, t) - \mathbf{x}^*\| < \epsilon, \forall t \geq t_0$. Where $B_\delta(\mathbf{x}^*) = \{\mathbf{x} \in \mathbb{R}^d : \|\mathbf{x} - \mathbf{x}^*\| < \delta\}$.

**Definition 7.9** *(Attractivity (136))* A constant vector $\mathbf{x}^* \in \mathbb{R}^d$ is an attractive equilibrium point for (1) if for every $t_0 \in \mathbb{R}^+$, there exists $\delta(t_0) > 0$ such that for every $\mathbf{z}_0 \in B_\delta(\mathbf{x}^*)$,

$$\lim_{t \to +\infty} \|\mathbf{s}(\mathbf{z}_0, t_0, t) - \mathbf{x}^*\| = 0.$$

**Definition 7.10** *(Asymptotic stability (136))* A constant vector $\mathbf{x}^* \in \mathbb{R}^d$ is said to be asymptotically stable if it is both stable and attractive. Note that if an equilibrium is exponentially stable, it is also asymptotically stable with exponential convergence.

Our goal is to make the contaminated instance $\hat{\mathbf{x}}$ converge to the clean instance $\mathbf{x}$. To achieve this evolution, we impose constraints on the ODE to output $\mathbf{z}(T) = \mathbf{x}$ when the input is $\mathbf{z}(0) = \hat{\mathbf{x}} \in B_\delta(\mathbf{x})$. To ensure that

$$\lim_{t \to +\infty} \|\mathbf{s}(\hat{\mathbf{x}}, t) - \mathbf{x}\| = 0,$$

where $\hat{\mathbf{x}} \in B_\delta(\mathbf{x})$, we make all $\mathbf{x} \in \mathcal{X}$ asymptotically stable equilibrium points.

**Theorem 7.2** *Suppose the perturbed instance $\hat{\mathbf{x}}$ is produced by adding a perturbation smaller than $\delta$ to the clean instance. If all the clean instances $\mathbf{x} \in \mathcal{X}$ are asymptotically stable equilibrium points of ODE (1), then there exists $\delta > 0$ such that for each contaminated instance $\hat{\mathbf{x}} \in \{\hat{\mathbf{x}} : \hat{\mathbf{x}} \in \hat{\mathcal{X}}, \hat{\mathbf{x}} \notin \mathcal{X}\}$, there exists $\mathbf{x} \in \mathcal{X}$ satisfying*

$$\lim_{t \to +\infty} \|\mathbf{s}(\hat{\mathbf{x}}, t) - \mathbf{x}\| = 0.$$

**Proof:**

According to the definition of asymptotic stability, a constant vector of a system is asymptotically stable if it is both stable and attractive. Based on the definition of stability of (1), for every $\epsilon > 0$ and every $t_0 \in \mathbb{R}^+$, there exists $\delta_1 = \delta(\epsilon, 0)$ such that

$$\forall \hat{\mathbf{x}} \in B_{\delta_1}(\mathbf{x}) \implies \|\mathbf{s}(\hat{\mathbf{x}}, t) - \mathbf{x}\| < \epsilon, \, \forall t \geq t_0.$$

Based on the attractivity definition, there exists $\delta_2 = \delta(0) > 0$ such that

$$\hat{\mathbf{x}} \in B_{\delta_2}(\mathbf{x}), \, \lim_{t \to +\infty} \|\mathbf{s}(\hat{\mathbf{x}}, t) - \mathbf{x}\| = 0.$$

We set $\delta = \min\{\delta_1, \delta_2\}$. Since the perturbed instance $\hat{\mathbf{x}}$ is produced by adding a perturbation smaller than $\delta$ to the clean instance, then for each contaminated instance $\hat{\mathbf{x}} \in \{\hat{\mathbf{x}} : \hat{\mathbf{x}} \in \hat{\mathcal{X}}, \hat{\mathbf{x}} \notin \mathcal{X}\}$, there exists a clean instance $\mathbf{x} \in \mathcal{X}$ such that $\hat{\mathbf{x}} \in B_\delta(\mathbf{x})$. Because the clean instance $\mathbf{x}$ is an asymptotically stable equilibrium point of (1), we have

$$\lim_{t \to +\infty} \|\mathbf{s}(\hat{\mathbf{x}}, t) - \mathbf{x}\| = 0.$$

This theorem guarantees that if we make the clean instance $\mathbf{x}$ an asymptotically stable equilibrium point, the ODE can reduce the perturbation and cause the perturbed instance to approach the clean instance. This can help improve the robustness of the DNN and aid it in defending against adversarial attacks.

**Theoretical Justification for Lyapunov-Stable Embedding Representations in Neural ODEs:**

We denote by $\lambda$ the pushforward measure, which is a probability distribution derived from the original distribution of the input data under the continuous feature extractor mapping $h_\phi$. The conditional probability distribution for the embeddings of each class in the training set, $l \in \{1, \ldots, L\}$, has compact support $E_l \subset \mathbb{R}^n$, as $E_l$ is closed and $h_\phi(X)$ is bounded in $\mathbb{R}^n$.

**Premise** The input data are sampled from a probability distribution defined over a compact metric space. The feature extractor $f_\theta$ is injective and continuous. Furthermore, the supports of each class in the embedding space are pairwise disjoint,

**Lemma 1.** *Given $k$ distinct points $\mathbf{z}_i \in \mathbb{R}^n$ and matrices $\mathbf{A}_i \in \mathbb{R}^{n \times n}$, for $i = 1, \ldots, k$, there exists a function $g \in \mathcal{C}^1(\mathbb{R}^n, \mathbb{R}^n)$ such that $g(\mathbf{z}_i) = 0$ and $\nabla g(\mathbf{z}_i) = \mathbf{A}_i$.*

**Proof:** The set of finite points $\{z_1, \ldots, z_k\}$ is closed, and this lemma is an immediate consequence of the Whitney extension theorem (137).

We restrict $h_\phi$ to be in $C^1(\mathbb{R}^n, \mathbb{R}^n)$ to satisfy the condition in Theorem 1. $C^1$ represents the function with first-order derivative. From (82), we also know that standard multilayer feedforward networks with as few as a single hidden layer and arbitrary bounded and non-constant activation functions are universal approximators for $C^1(\mathbb{R}^n, \mathbb{R}^n)$ functions with respect to some performance criteria, provided only that sufficiently many hidden units are available.

Suppose for each class $l = 1, \ldots, L$, the embedding feature set $E_l = \{z_1^{(l)}, \ldots, z_k^{(l)}\}$ is finite. For each $i = 1, \ldots, k$, let $A_i \in \mathbb{R}^{n \times n}$ be a strictly diagonally dominant matrix with every main diagonal entry negative, such that the eigenvalues of $A_i$ all have negative real parts. From Theorem 3, each $A_i$ is non-singular and every eigenvalue of $A_i$ has negative real part. Therefore, from Theorem 2 and Lemma 1, there exists a function $h_\phi$ such that all $z_i^{(l)}$ are Lyapunov-stable equilibrium points with

corresponding first derivative $\nabla h_\phi(z_i^{(l)}) = A_i$. This shows that if there exist only finite representation points for each class, we can find a function $h_\phi$ such that all inputs to the neural ODE layer are Lyapunov-stable equilibrium points for $h_\phi$ and

(I) $\mathbb{E}_\lambda \|h_\phi(X_{\text{train}})\|_2 = 0$,

(II) $\mathbb{E}_\lambda \big[\nabla h_\phi(X_{\text{train}})\big]_{ii} < 0$,

(III) $\mathbb{E}_\lambda \left[ \big[\nabla h_\phi(X_{\text{train}})\big]_{ii} - \sum_{j \neq i} \big[\nabla h_\phi(X_{\text{train}})\big]_{ij} \right] > 0$.

We will show that under mild conditions, for all $\epsilon > 0$, we can find a continuous function $h_\phi$ with finitely many stable equilibrium points such that conditions (II) and (III) above hold and condition (I) is replaced by $\mathbb{E}_\lambda \|h_\phi(X_{\text{train}})\|_2 < \epsilon$. This motivates the optimization constraints in (I, II, III).

**Theorem 7.3.** *Suppose Premise hold. If $\lambda$ is not a continuous measure on $E_l$ for each $l = 1, \ldots, L$, then the following holds:*

1. *The function space satisfying the constraints in (I, II, III) is non-empty for all $\epsilon > 0$.*

2. *If additionally the restriction of $\lambda$ to any open set $O \subset E_l$ is a continuous measure, then we can find such a function such that each support $E_l$ almost surely satisfies the conditions in (I, II, III).*

**Proof:** Consider $g(z) = [g^{(1)}(z^{(1)}), \ldots, g^{(n)}(z^{(n)})]$ with each $g^{(i)}(z^{(i)}) \in C^1(\mathbb{R}, \mathbb{R})$. Since $g^{(i)}(z^{(i)})$ depends only on $z^{(i)}$, $\nabla g_\theta(z)$ is a diagonal matrix with all off-diagonal elements being 0. The constraint (III) is thus immediately satisfied, and it suffices to show that there exists such an $f$ satisfying the constraints (II) and (III).

Select a point $z_l = (z_l^{(1)}, \ldots, z_l^{(n)})$ from the interior of each $E_l$, for $l = 1, \ldots, L$. Let $g^{(i)}(z^{(i)}) = -\nu(z^{(i)} - z_l^{(i)})$ on each $E_l$, where $\nu > 0$. Then $g(z)$ satisfies (II) for all $\nu > 0$, and $z_l$ is a Lyapunov-stable equilibrium point for each $l$ since $\nabla h_\phi(z)$ is a diagonal matrix with negative diagonal values. Since each $E_l \subset \mathbb{R}^n$ is compact, we have that $\forall \epsilon > 0, \exists \nu > 0$ sufficiently small such that $|f^{(i)}(z^{(i)})| < \epsilon$ for all $z \in \bigcup_l E_l$. The constraint (I) is therefore satisfied for $f(z)$ with a sufficiently small $\nu$. Since $\bigcup_l E_l$ is closed, the Whitney extension theorem (137) can be applied to extend $f(z)$ to a function in $C^1(\mathbb{R}^n, \mathbb{R}^n)$.

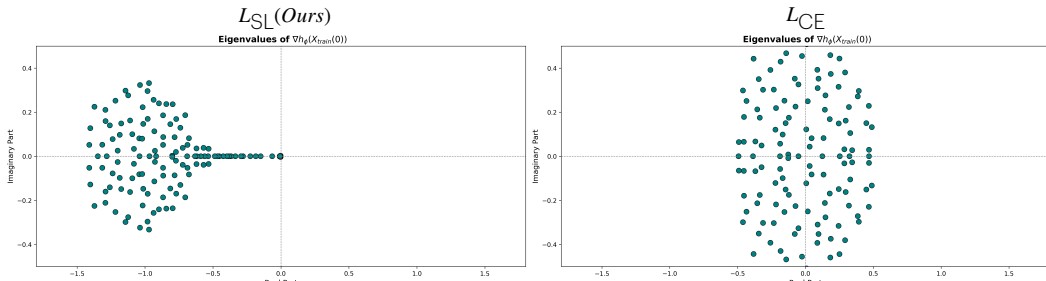

Figure 8: Eigenvalue visualization of the Jacobian matrix $\nabla h_\theta(z(0))$ for a NODE trained on CIFAR-10 using using the loss functions $\mathcal{L}_{\text{CE}}$ and $\mathcal{L}_{\text{SL}}$. The results demonstrate that $\mathcal{L}_{\text{SL}}$ encourages eigenvalues with negative real parts, indicating stability.

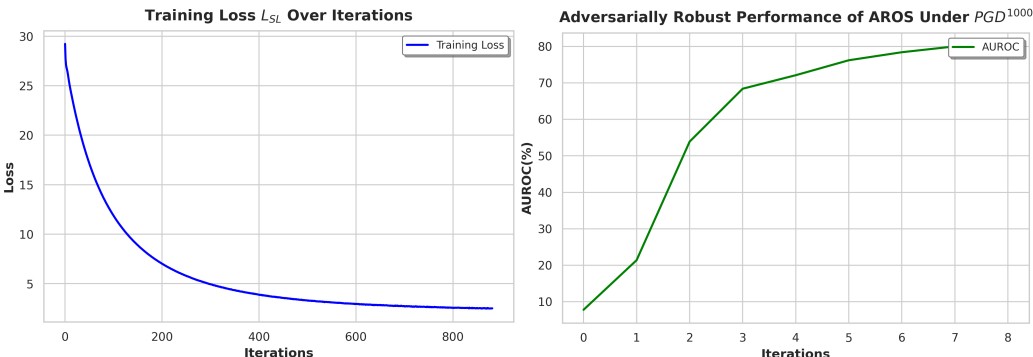

Figure 9: Training loss ($\mathcal{L}_{\text{SL}}$) and adversarial robustness performance of AROS on the CIFAR-10 vs. CIFAR-100 benchmark (CIFAR-10 served as the ID dataset). The left plot shows the convergence of the stability-based loss $\mathcal{L}_{\text{SL}}$ over iterations, demonstrating effective training. The right plot depicts the AUROC performance under PGD[1000] attacks, highlighting the adversarial robustness achieved by AROS as iterations progress.