# OpenReview forum: "Adversarially Robust Out-of-Distribution Detection Using Lyapunov-Stabilized Embeddings"
_ICLR.cc/2025/Conference — ICLR 2025 Poster_

### Official Review · Reviewer_x9hq · 2024-11-02

**Soundness:** 3
**Presentation:** 3
**Contribution:** 3
**Rating:** 6
**Confidence:** 3

**Summary:**

This paper introduces AROS, a method for enhancing the robustness of OOD detection under adversarial attacks. Using Lyapunov-stabilized embeddings within a NODE framework, AROS aims to stabilize the decision boundaries between ID and OOD samples by promoting equilibrium convergence through adversarial training. The approach includes generating synthetic OOD data in the feature space and applying an orthogonal binary layer to maximize separation between ID and OOD samples.

**Strengths:**

1. **Innovative Use of Stability Theory:** The integration of Lyapunov stability within NODEs for OOD detection is novel and well-motivated. The application of stability principles to mitigate adversarial perturbations shows theoretical grounding.

2. **Comprehensive Evaluation:** The method undergoes extensive testing across datasets (CIFAR, ImageNet, and medical data) under both adversarial and clean conditions. Performance metrics like AUROC and AUPR are presented alongside ablations, providing a thorough examination.

3. **Orthogonal Binary Layer:** The addition of an orthogonal layer to further separate equilibrium points of ID and OOD data adds robustness and addresses alignment issues in perturbed scenarios.

**Weaknesses:**

1. **Complexity of Mathematical Justifications:** The paper’s reliance on Lyapunov stability and NODEs introduces significant mathematical complexity. Equation derivations and stability proofs (such as those around equilibrium points) are not fully explained, making it challenging for readers without a control theory background.

2. **Synthetic OOD Data Generation:** While the fake OOD generation in the feature space is practical, it lacks clarity on how well it approximates real-world OOD samples. Further analysis or empirical validation of this approximation would strengthen the argument for its effectiveness.

3. **Loss Function Ambiguity:** The custom loss function, $L_{SL}$, is somewhat ambiguous in its practical effects on training stability. Further detail on parameter tuning (e.g., effects of $\gamma$-values) and ablation studies on these would clarify its role in achieving robustness.

**Questions:**

1. Could the authors clarify the theoretical guarantees of Lyapunov stability when applied within the NODE framework, particularly for high-dimensional embeddings? Specifically, could they expand on how the equilibrium points are stabilized across varying adversarial perturbation magnitudes, especially under different norm constraints? Additionally, the derivations and stability proofs related to Theorems 1 and 2 are complex. Would the authors consider including expanded derivations or references in the appendix to enhance understanding for readers without a control theory background?


2. The introduction of the orthogonal binary layer aims to enhance ID-OOD separation. Could the authors provide more empirical insights or visualizations to demonstrate the layer's effectiveness in maintaining separation, particularly in adversarial scenarios? Furthermore, an analysis of the sensitivity of this layer to different perturbation levels would strengthen the practical understanding of its robustness.


3. The authors propose generating synthetic OOD embeddings by sampling from low-likelihood regions in the ID feature space. However, how well does this synthetic data represent real-world OOD instances, which may be more complex in structure? Could the authors provide empirical evidence or a quantitative comparison to validate the representativeness of these synthetic OOD samples? Additionally, would the authors consider including an ablation or alternative generation methods to explore the impact of different OOD data synthesis strategies?

---

> ### Author Response · Authors · 2024-11-22
>
> Dear Reviewer x9hq,
>
> We are grateful for your insightful comments and feedback.  To address your concerns, we have provided comments as follows.
>
> >**Q1&W1:**
>
>
> Our proposed objective function builds upon foundational studies in control theory and equilibrium-based deep learning methods. The proofs of the theorems are supported by references to the original sources, as cited in our manuscript. However, we acknowledge the reviewer’s concerns and will actively work to enhance the manuscript by providing additional background on control theory and offering a theoretical justification for the robustness of Lyapunov stability.
>
>
>
> >**Q2:**
>
> Our manuscript includes an ablation study to investigate the effect of the orthogonal binary layer compared to a regular fully connected (FC) layer, as detailed in Table 5, setup C. In response to the reviewers' concerns, we conducted additional experiments and provided visualizations in Section A3.6, specifically in Table 16a. To ensure clarity, we have also included this table here for reference. We kindly request that reviewers refer to Section A3.6 for a detailed comparison, including the visualizations (Figure 7) illustrating the differences in performance between the orthogonal binary layer and the regular FC layer.
>
>
> | **ε**     | Methods | CIFAR-10 |        |        | CIFAR-100 |        |        | ImageNet-1k |          |        |
> |-|-|-|-|-|-|-|-|-|-|-|
> |          |         | CIFAR-100 | SVHN   | LSUN   | CIFAR-10  | SVHN   | LSUN   | Texture     | iNaturalist | LSUN  |
> | **0/255 (Clean)**     | FC          | 85.6         | 88.2          | 87.5     | 66.9     | 78.4         | 71.5     | 75.4     | 79.8        | 76.1           |
> |                       | Ortho.      | 88.2         | 93.0          | 90.6     | 74.3     | 81.5         | 74.3     | 78.3     | 84.0        | 79.4           |
> | **2/255**             | FC          | 82.0         | 84.9          | 84.1     | 63.6     | 75.8         | 68.6     | 73.2     | 76.9        | 72.2           |
> |                       | Ortho.      | 85.5         | 90.7          | 88.8     | 72.2     | 79.9         | 72.1     | 75.7     | 82.7        | 76.6           |
> | **4/255**             | FC          | 75.2         | 78.1          | 80.3     | 57.9     | 71.6         | 63.5     | 68.2     | 72.9        | 66.7           |
> |                       | Ortho.      | 83.1         | 87.5          | 85.2     | 69.5     | 74.5         | 70.4     | 71.6     | 78.3        | 68.0           |
> | **8/255**             | FC          | 67.3         | 74.6          | 79.5     | 57.1     | 63.3         | 62.4     | 60.7     | 70.2        | 65.1           |
> |                       | Ortho.      | 80.1         | 86.4          | 82.4     | 67.0     | 70.6         | 68.1     | 69.2     | 75.3        | 69.0           |
>
>
> This comparison highlights the sensitivity analysis of the orthogonal binary layer (Ortho.) and the regular fully connected (FC) layer across various datasets and perturbation levels (ϵ). The results demonstrate that the orthogonal binary layer consistently outperforms the regular FC layer in both clean (ε = 0/255) and adversarial scenarios, with varying levels of perturbation (ε = 2/255, 4/255, 8/255).

---

> ### Author Response · Authors · 2024-11-22
>
> >**W2&Q3-a:**
>
> It has been shown that utilizing auxiliary real-world OOD samples is generally effective for improving OOD detection performance. However, this strategy comes with several challenges, as discussed in our manuscript (see lines 79–89 and 190-200).
> First, in certain scenarios, access to an external real-world OOD dataset may not be feasible, and acquiring such data can be challenging. Even when a suitable dataset is available, it must be processed to remove ID concepts to prevent the detector from being misled. This preprocessing step is both time-consuming and computationally expensive. Additionally, studies highlight a potential risk of bias being introduced into the detector when trained on specific auxiliary datasets. Such reliance on a particular real-world dataset may undermine the detector's ability to generalize effectively to diverse OOD samples. These issues become even more pronounced in adversarial training setups, where the complexity of the required data is significantly higher. Motivated by these challenges, this study proposes an alternative strategy that does not depend on the availability of an auxiliary OOD dataset. Notably, our approach is flexible and can incorporate auxiliary OOD datasets as additional information if they are available.
> To validate this, we conducted an experiment assuming access to a real-world OOD dataset (i.e., Food-101). In this scenario, we computed embeddings of the real OOD samples and used them alongside crafted fake OOD embeddings during training. The results, presented in Table 4 (Setup A), demonstrate improved performance compared to using fake OOD embeddings alone.
> Furthermore, related studies have shown that in adversarial training, using samples near the decision boundary of the distribution improves robustness by encouraging compact representations. This boundary modeling is critical for enhancing the model's robustness, especially against adversarial attacks that exploit vulnerabilities near the decision boundary.
> In light of this, our approach shifts focus from generating “realistic” OOD data to estimating low-likelihood regions of the in-distribution. We generate fake "near" OOD data that is close to the ID boundary, which is particularly beneficial for adversarial robustness.
>  For better intuition regarding usefulness of auxiliary near OOD samples here we will provide a simple example that highlights the effectiveness of near-distribution crafted OOD samples in the adversarial setup.  We assume that the feature space is one-dimensional, i.e. $\mathbb{R}$, and the ID class is sampled according to a uniform distribution $U(0, a - \epsilon)$, with $a > 0$, and $\epsilon < a $. We assume that the OOD class is separable with a safety margin of $2\epsilon$ from the ID class to allow a perfectly accurate OOD detector under input perturbations of at most $\epsilon$ at inference. For instance, we let $U(a+\epsilon, b)$ be the feature distribution under the OOD class. The goal is to leverage crafted fake OOD samples to find a robust OOD detector under the $\ell_2$ bounded perturbations of norm $\epsilon$. We assume that the crafted OOD samples data distribution is not perfectly aligned with the OOD data, e.g. crafted OOD samples comes from $U(a+r, c)$, with $r \geq \epsilon$. It is evident that the optimal robust decision boundary under infinite training samples that separates the ID and crafted OOD samples would be a threshold $k$ satisfying $a \leq k \leq a + r - \epsilon$. The test adversarial error rate to separate ID and OOD classes is $\frac{1}{2}. \mathbb{I}(k \geq a+\epsilon).\left(\frac{k-a-\epsilon}{b-a-\epsilon}  + \frac{\min(k+\epsilon, b)-k}{b - a - \epsilon} \right) + \frac{1}{2}.\mathbb{I}(a < k < a + \epsilon) \frac{\min(k+\epsilon, b) - a - \epsilon}{b - a - \epsilon} $, assuming that the classes are equally probable a prior. It is obvious the adversarial error rate would be zero for $k = a$. But otherwise, if $k \geq a + \epsilon$ the classifier incurs classification error in two cases; in intervals $(a+\epsilon, k)$ (even without any attack), and $(k, \min(k+\epsilon, b))$ in which a perturbation of $-\epsilon$ would cause  classification error. Also if $a < k < a + \epsilon$, classification error only happens at $(a+\epsilon, \min(k+\epsilon, b))$. Now, for the crafted OOD samples to be near-distribution, $r \rightarrow \epsilon$, which forces $k$ to be $a$ in the training, and makes the test adversarial error zero. Otherwise, the adversarial error is proportional to $k$, for $k$ being distant from $b$. Therefore, in the worst case, if $k = a + r - \epsilon$, we get an adversarial error proportional to $r$. As a result, minimizing $r$, which makes the crafted OOD samples near-distribution, would be an effective strategy in making the adversarial error small. We added this new example to our updated manuscript PDF along with a new figure for further clarification. We kindly ask the reviewer to see A3.5 section of the Appendix.

---

> ### Author Response · Authors · 2024-11-22
>
> >**Q3-b:**
>
> >**...Additionally, would the authors consider including an ablation or alternative generation methods to explore the impact of different OOD data synthesis strategies?**
>
> Our submitted manuscript includes an ablation study where alternative generation strategies were substituted for the proposed method (see Table 5, Setup D). To further address the reviewer’s concerns, we have conducted additional experiments, the results of which can be found in Table 15a of the manuscript and are also presented here. These experiments eliminate the conditioning of the fake OOD distribution on the ID data, thereby ensuring that the generated OOD samples are unrelated to ID. This aligns with prior studies that have demonstrated the utility of related and nearby auxiliary OOD samples in improving performance.
>
>
> | **Fake Crafting Strategy** | **CIFAR-10**        |                 |                 | **CIFAR-100**       |                 |                 | **ImageNet-1k**          |                 |                 |
> |-|-|-|-|-|-|-|-|-|-|
> |                             | **CIFAR-100**      | **SVHN**        | **LSUN**        | **CIFAR-10**        | **SVHN**        | **LSUN**        | **Texture**              | **iNaturalist** | **LSUN**        |
> | **AROS**                   | 88.2/80.1          | 93.0/86.4       | 90.6/82.4       | 74.3/67.0           | 81.5/70.6       | 74.3/68.1       | 78.3/69.2                | 84.6/75.3       | 79.4/69.0       |
> | **Random Gaussian Noise**   | 85.3/76.5          | 89.4/78.1       | 82.5/77.1       | 70.5/61.3           | 74.4/62.5       | 71.5/64.6       | 76.1/67.4                | 81.3/72.7       | 75.8/67.3       |
> | **Random Uniform Noise**    | 81.2/74.3          | 87.1/79.5       | 84.8/75.4       | 65.6/59.6           | 73.8/63.5       | 65.8/60.1       | 70.0/63.8                | 78.8/67.3       | 71.6/62.8       |
>
>
> *The perturbation budget $\epsilon$ is set to $\frac{8}{255}$ for low-resolution datasets and $\frac{4}{255}$ for high-resolution datasets. Table cells present results in the 'Clean/PGD$_{1000}$' format. We evaluate alternative OOD data synthesis strategies, such as random Gaussian and uniform noise instead of crafted fake OOD, while keeping other components fixed. Results are measured by AUROC (%).*

---

> ### Author Response · Authors · 2024-11-22
>
> >**W3:**
>
>
> **Hyperparameters $\gamma_1$, $\gamma_2$, $\gamma_3$:**
>
> In our proposed method, we introduce the empirical loss function $\mathcal{L}_{\text{SL}}$ as follows:
>
> $$
> L_{SL} = \min_{\phi, \eta} \frac{1}{\left| X_{\text{train}} \right|} \Bigg(\ell_{\text{CE}}(  B_{\eta} (h_{\phi}(X_{\text{train}}), y)) + \gamma_1 \| \| h_{\phi}(X_{\text{train}})\|\|_2
> $$
>
> $$
> \quad \quad \quad \quad \quad \quad + \gamma_2 \text{exp} \bigg(-\sum_{i=1}^n [\nabla h_{\phi}(X_{\text{train}})]_{ii} \bigg)
> $$
>
> $$
> \quad \quad \quad \quad \quad \quad \quad \quad \quad \quad \quad \quad + \gamma_3 \text{exp} \bigg( \sum_{i=1}^n (-|[\nabla h_{\phi}(X_{\text{train}})]_{ii}|
> $$
>
> $$
> \quad \quad \quad \quad \quad \quad \quad \quad \quad \quad \quad \quad + \sum_{j \neq i} |[\nabla h_{\phi}(X_{\text{train}})]_{ij}| ) \bigg) \Bigg)
> $$
>
> ### Here:
> - $X_{\text{train}}$ is the set of training embeddings.
> - $y$ are the corresponding labels.
> - $h_{\phi}$ is the NODE function parameterized by $\phi$.
> - $B_{\eta}$ is the orthogonal binary layer parameterized by $\eta$.
> - $\ell_{\text{CE}}$ denotes the cross-entropy loss.
> - $\nabla h_{\phi}(X_{\text{train}})$ is the Jacobian matrix of $h_{\phi}$ with respect to $X_{\text{train}}$.
>
> ---
>
> **Roles of the Hyperparameters**
>
> ### 1. $\gamma_1$
> Controls the regularization term $\| h_{\phi}(X_{\text{train}})\|_2$, which encourages the system's state to be near the equilibrium point. This term helps mitigate the effect of perturbations by ensuring that trajectories remain close to the equilibrium.
>
> ### 2. $\gamma_2$ and $\gamma_3$
> These hyperparameters weight the exponential terms designed to enforce Lyapunov stability conditions by ensuring the Jacobian matrix satisfies strict diagonal dominance (as per Theorem 3 in the paper).
>
> - **$\gamma_2$** weights the term $\exp\left( -\sum_{i=1}^n [\nabla h_{\phi}(X_{\text{train}})]_{ii} \right)$, encouraging the diagonal entries of the Jacobian to be negative with large magnitudes.
>
> - **$\gamma_3$** weights the term $\text{exp} \bigg( \sum_{i=1}^n (-|[\nabla h_{\phi}(X_{\text{train}})]_{ii}| \bigg)$
>
>   promoting strict diagonal dominance by penalizing large off-diagonal entries relative to the diagonal entries.
>
> ---
>
> **Reasoning Behind Default Values**
>
> - **$\gamma_1 = 1$**: Balances the influence of the regularization term with the primary cross-entropy loss. This ensures sufficient regularization without overwhelming the classification objective.
> - **$\gamma_2 = \gamma_3 = 0.05$**: Preliminary experiments indicate these values effectively enforce stability without causing numerical issues or hindering optimization. We choose $\gamma_2 = \gamma_3 = 0.05$ Inspired by related works [1-10] that incorporate stability terms into the training process.
>
> ---
>
> **Why $\gamma_2$ and $\gamma_3$ Are Equal**
>
> 1. **Balanced Contribution**: Both regularization terms serve complementary purposes in enforcing the stability conditions. Equal weighting ensures neither term dominates, maintaining a balanced emphasis on both negative diagonal dominance and strict diagonal dominance.
> 2. **Simplified Hyperparameter Tuning**: Setting $\gamma_2$ and $\gamma_3$ equal reduces the hyperparameter search space, simplifying the tuning process without sacrificing performance.
>
>
> ---
>
> **Impact of Hyperparameters on Performance**
>
> **Effect of $\gamma_1$**
>
>
> - **Low Values ($\gamma_1 < 1$)**: Reduces the emphasis on keeping the state near the equilibrium, making the system more susceptible to perturbations.
> - **High Values ($\gamma_1 > 1$)**: Overly constrains the state to the equilibrium point, potentially limiting the model's capacity to learn discriminative features.
> - **Chosen Value ($\gamma_1 = 1$)**: Offers a good balance, ensuring sufficient regularization without compromising learning.
>
> ---
>
> **Effect of $\gamma_2$ and $\gamma_3$**
>
>
>
> - **Low Values ($< 0.05$)**: Diminish the impact of the stability constraints, reducing robustness.
> - **High Values ($> 0.05$)**: Overemphasize the stability terms, potentially hindering the optimization of the primary classification loss.
> - **Equal Values ($\gamma_2 = \gamma_3$)**: Ensure balanced enforcement of both stability conditions, leading to optimal performance.
>
>
> To address the reviewers' concerns, we conducted extensive ablation studies, keeping all components fixed while varying the values of $\gamma_1$, $\gamma_2$, and $\gamma_3$. These experiments demonstrate that AROS consistently performs well across a broad range of hyperparameter values, including extreme cases (e.g., $\gamma_1 = 2$). The results of this analysis are presented in Table 13a of our paper and are also included below. Notably, this table was part of our submitted manuscript, and we have now extended it with additional experiments to further address any remaining concerns. These results confirm the robust performance of AROS under varying configurations of $\gamma_1$, $\gamma_2$, and $\gamma_3$.

---

> ### Author Response · Authors · 2024-11-22
>
> >**W3:**
>
> | Hyperparameter                     | **CIFAR-10**        |           |           | **CIFAR-100**       |           |           | **ImageNet-1k**     |           |           |
> |-|-|-|-|-|-|-|-|-|-|
> |                                    | **CIFAR-100**      | **SVHN**  | **LSUN**  | **CIFAR-10**        | **SVHN**  | **LSUN**  | **Texture**         | **iNaturalist** | **LSUN** |
> | **$\gamma_1=1, \gamma_2=\gamma_3=0.05$ (default)**     | 88.2/80.1          | 93.0/86.4 | 90.6/82.4 | 74.3/67.0           | 81.5/70.6 | 74.3/68.1 | 78.3/69.2          | 84.6/75.3       | 79.4/69.0 |
> | **$\gamma_1=1, \gamma_2=\gamma_3=0.025$**              | 86.4/78.2          | 90.4/86.3 | 88.6/80.4 | 74.9/64.1           | 78.8/68.7 | 72.8/68.3 | 79.2/69.7          | 83.3/76.0       | 78.8/67.5 |
> | **$\gamma_1=1, \gamma_2=\gamma_3=0.075$**              | 87.5/80.3          | 92.1/87.2 | 88.9/81.6 | 72.7/67.7           | 80.8/68.7 | 73.2/68.8 | 76.4/69.9          | 83.4/73.4       | 80.2/69.9 |
> | **$\gamma_1=1, \gamma_2=\gamma_3=0.1$**                | 86.9/78.6          | 93.6/82.1 | 85.8/78.2 | 70.0/65.8           | 80.6/66.2 | 72.6/67.3 | 79.3/69.6          | 82.8/74.0       | 78.9/67.2 |
> | **$\gamma_1=1, \gamma_2=\gamma_3=0.25$**               | 86.2/78.7          | 94.3/82.3 | 87.1/77.9 | 70.2/65.2           | 81.3/65.8 | 73.9/66.7 | 78.6/70.1          | 82.8/74.1       | 78.9/68.2 |
> | **$\gamma_1=1, \gamma_2=\gamma_3=0.5$**                | 85.2/76.9          | 93.6/81.5 | 85.0/76.0 | 69.7/64.2           | 78.1/64.4 | 72.6/65.2 | 77.0/66.7          | 83.6/74.4       | 78.2/67.3 |
> | **$\gamma_1=\gamma_2=\gamma_3=1$**                     | 84.4/75.3          | 90.5/80.5 | 81.5/73.4 | 69.8/62.2           | 77.5/65.3 | 70.1/64.4 | 73.5/67.3          | 82.7/70.9       | 76.7/65.4 |
> | **$\gamma_1=\gamma_2=\gamma_3=0.25$**                  | 85.4/77.4          | 94.6/82.9 | 86.3/76.9 | 69.9/64.4           | 79.9/65.1 | 74.4/65.3 | 77.2/68.6          | 85.4/76.0       | 79.2/66.8 |
> | **$\gamma_1=0.5, \gamma_2=\gamma_3=0.05$**             | 86.0/78.6          | 92.7/82.5 | 86.3/77.7 | 69.7/66.1           | 81.2/65.5 | 70.9/67.1 | 78.8/68.6          | 83.0/73.7       | 78.8/67.1 |
> | **$\gamma_1=1, \gamma_2=0.1, \gamma_3=0.05$**          | 84.0/76.3          | 91.5/81.5 | 81.7/76.4 | 70.1/63.4           | 76.2/63.8 | 69.8/67.7 | 77.4/69.5          | 83.2/71.0       | 78.5/64.8 |
> | **$\gamma_1=1, \gamma_2=0.05, \gamma_3=0.1$**          | 83.5/77.5          | 90.2/78.0 | 81.4/75.1 | 67.2/63.5           | 76.7/66.2 | 67.5/60.4 | 74.3/70.0          | 83.6/74.1       | 74.2/61.7 |
> | **$\gamma_1=2, \gamma_2=0.5, \gamma_3=0.5$**           | 80.4/74.0          | 93.5/80.7 | 81.8/70.9 | 65.2/66.0           | 77.3/60.0 | 65.0/60.1 | 78.4/63.3          | 80.3/70.6       | 71.6/59.5 |
>
>
> ---
>
> ---
>
>
>
> [1] Zonghan Yang, Tianyu Pang, and Yang Liu. A closer look at the adversarial robustness of deep equilibrium models. Advances in Neural Information Processing Systems, 35:10448–10461, 2022.
>
> [2] Ivan Dario Jimenez Rodriguez, Aaron Ames, and Yisong Yue. Lyanet: A lyapunov framework for training neural odes. In International conference on machine learning, pages 18687–18703. PMLR, 2022.
>
> [3] Xiyuan Li, Zou Xin, and Weiwei Liu. Defending against adversarial attacks via neural dynamic system. Advances in Neural Information Processing Systems, 35:6372–6383, 2022.
>
> [4] Qiyu Kang, Yang Song, Qinxu Ding, and Wee Peng Tay. Stable neural ode with lyapunov-stable equilibrium points for defending against adversarial attacks. Advances in Neural Information Processing Systems, 34:14925–14937, 2021.
>
> [5] Fabio Carrara, Roberto Caldelli, Fabrizio Falchi, and Giuseppe Amato. On the robustness to adversarial examples of neural ode image classifiers. In 2019 IEEE International Workshop on Information Forensics and Security (WIFS), pages 1–6. IEEE, 2019.
>
> [6] Jan Svoboda, Jonathan Masci, Federico Monti, Michael Bronstein, and Leonidas Guibas. Peernts: Exploiting peer wisdom against adversarial attacks. In International Conference on Learning Representations, 2019.
>
> [7] Arash Rahnama, Andre T Nguyen, and Edward Raff. Robust design of deep neural networks against adversarial attacks based on lyapunov theory. In Proceedings of the IEEE/CVF Conference on Computer Vision and Pattern Recognition, pages 8178–8187, 2020.
>
> [8] Mingjie Li, Lingshen He, and Zhouchen Lin. Implicit euler skip connections: Enhancing adversarial robustness via numerical stability. In International Conference on Machine Learning, pages 5874–5883. PMLR, 2020.
>
>
> [9] Sergey Dashkovskiy, Oleksiy Kapustyan, and Vitalii Slynko. Robust stability of a nonlinear ode-pde system. SIAM Journal on Control and Optimization, 61(3):1760–1777, 2023.
>
> [10] Mustafa Zeqiri, Mark Niklas Müller, Marc Fischer, and Martin Vechev. Efficient certified training and robustness verification of neural odes. arXiv preprint arXiv:2303.05246, 2023.

---

> ### Author Response · Authors · 2024-11-27
>
> >**W1&Q1&W3 (a):**
>
> Dear Reviewer x9hq,
>
> Thank you for your constructive comments. To thoroughly address the questions raised in W1&Q1&W3, we have added a new section (A.7) to our appendix. This section aims to make our manuscript clearer and directly address your concerns. Specifically, we have included the following updates:
>
> * **Expanded Background:** We provide a detailed explanation of the theorems used from control theory, including their proofs (e.g., the proof of the Hartman-Grobman Theorem).
>
> * **Practical Details:** Additional practical details about our proposed loss function are discussed to enhance understanding.
> Theoretical Guarantees: We offer theoretical guarantees of Lyapunov stability when applied to Neural Ordinary Differential Equations (NODE).
>
> * **Training Stability:** To address concerns about the practical effects of $\mathcal{L}_{SL}$ loss function on training stability, we have plotted the loss values over several iterations, demonstrating its robust performance (please kindly see Figure 9).
>
> * **Eigenvalue Analysis:** To further support the efficacy of our proposed loss function, we provide a plot of the eigenvalues of the Jacobian matrix. This illustrates how our objective function ensures that the eigenvalues remain negative (please kindly see Figure 8).
> Additionally, we offer justifications for the theoretical guarantees of Lyapunov stability within the NODE framework, particularly for high-dimensional embeddings. For more details, we kindly refer you to Appendix A.7 of our paper.

---

> ### Author Response · Authors · 2024-11-27
>
> >**W1&Q1&W3 (b):**
>
>
> Here, we aim to establish a theoretical foundation explaining how making clean instances asymptotically stable equilibrium points in an ODE ensures that perturbed instances converge back to them over time:
>
>
> ---
>
>
>
>
>
>
> Consider an ODE problem:
> $$
> \frac{d\mathbf{z}(t)}{dt} = \mathbf{h}(\mathbf{z}(t), t), t \geq t_0;\quad \mathbf{z}(t_0) = \mathbf{z}_0.
> $$
>
> Let $\mathbf{s}(\mathbf{z}_0, t_0, t)$ denote the solution of the ODE corresponding to the initial input $\mathbf{z}_0$ at time $t_0$.
>
> **Definition 1** *(Equilibrium)*[vidyasagar2002nonlinear]  A vector $ \mathbf{x}^* $ is called an equilibrium of a system if $ \mathbf{h}(\mathbf{x}^*, t) = 0, \, \forall t \geq 0 $.
>
> **Definition 2** *(Stability)*[bhatia2002stability] A constant vector $ \mathbf{x}^* \in \mathbb{R}^d $ is a stable equilibrium point for a system if, for every $ \epsilon > 0 $ and every $ t_0 \in \mathbb{R}^+ $, there exists $ \delta(\epsilon, t_0) $ such that for each $\mathbf{z}_0$ in  ball of radius $\delta$ , it holds that $ \|\mathbf{s}(\mathbf{z}_0, t_0, t) - \mathbf{x}^*\| < \epsilon, \, \forall t \geq t_0 $.
>
>
> (Ball of radius $\delta$ : $B_\delta(\mathbf{x}^*) = \{\mathbf{x} \in \mathbb{R}^d : \|\mathbf{x} - \mathbf{x}^*\| < \delta\}$)
>
>
>
>
>
>
>
>
> **Definition 3** *(Attractivity)* [bhatia2002stability] A constant vector $ \mathbf{x}^* \in \mathbb{R}^d $ is an attractive equilibrium point for ODE problem if for every $ t_0 \in \mathbb{R}^+ $, there exists $ \delta(t_0) > 0 $ such that for every $ \mathbf{z}_0 $ in  ball of radius $\delta$:
>
> $\lim_{t \to +\infty} \|\mathbf{s}(\mathbf{z}_0, t_0, t) - \mathbf{x}^*\| = 0.$
>
>
>
>
>
> **Definition 4** *(Asymptotic stability)* [bhatia2002stability]  A constant vector $ \mathbf{x}^* \in \mathbb{R}^d $ is said to be asymptotically stable if it is both stable and attractive. Note that if an equilibrium is exponentially stable, it is also asymptotically stable with exponential convergence.
>
> Our goal is to make the contaminated instance $ \hat{\mathbf{x}} $ converge to the clean instance $ \mathbf{x} $. To achieve this evolution, we impose constraints on the ODE to output $ \mathbf{z}(T) = \mathbf{x} $ when the input is $ \mathbf{z}(0) = \hat{\mathbf{x}} \in B_\delta(\mathbf{x}) $. To ensure that
> $$
> \lim_{t \to +\infty} \|\mathbf{s}(\hat{\mathbf{x}}, t) - \mathbf{x}\| = 0,
> $$
> where $ \hat{\mathbf{x}} \in B_\delta(\mathbf{x}) $, we make all $ \mathbf{x} \in \mathcal{X} $ asymptotically stable equilibrium points.
>
> **Theorem A** *Suppose the perturbed instance $ \hat{\mathbf{x}} $ is produced by adding a perturbation smaller than $ \delta $ to the clean instance. If all the clean instances $ \mathbf{x} \in \mathcal{X} $ are asymptotically stable equilibrium points of ODE equation, then there exists $ \delta > 0 $ such that for each contaminated instance $ \hat{\mathbf{x}} \in \{\hat{\mathbf{x}} : \hat{\mathbf{x}} \in \hat{\mathcal{X}}, \hat{\mathbf{x}} \not\in \mathcal{X}\} $, there exists $ \mathbf{x} \in \mathcal{X} $ satisfying*
> $$
> \lim_{t \to +\infty} \|\mathbf{s}(\hat{\mathbf{x}}, t) - \mathbf{x}\| = 0.
> $$
>
> **Proof:**
>
> According to the definition of asymptotic stability, a constant vector of a system is asymptotically stable if it is both stable and attractive. Based on the definition of stability of ODE problem, for every $ \epsilon > 0 $ and every $ t_0 \in \mathbb{R}^+ $, there exists $ \delta_1 = \delta(\epsilon, 0) $ such that
> $$
> \forall \hat{\mathbf{x}} \in B_{\delta_1}(\mathbf{x}) \implies \|\mathbf{s}(\hat{\mathbf{x}}, t) - \mathbf{x}\| < \epsilon, \, \forall t \geq t_0.
> $$
>
> Based on the attractivity definition, there exists $ \delta_2 = \delta(0) > 0 $ such that
> $$
> \hat{\mathbf{x}} \in B_{\delta_2}(\mathbf{x}), \, \lim_{t \to +\infty} \|\mathbf{s}(\hat{\mathbf{x}}, t) - \mathbf{x}\| = 0.
> $$
>
> We set $ \delta = \min\{\delta_1, \delta_2\} $. Since the perturbed instance $ \hat{\mathbf{x}} $ is produced by adding a perturbation smaller than $ \delta $ to the clean instance, then for each contaminated instance $ \hat{\mathbf{x}} \in \{\hat{\mathbf{x}} : \hat{\mathbf{x}} \in \hat{\mathcal{X}}, \hat{\mathbf{x}} \not\in \mathcal{X}\} $, there exists a clean instance $ \mathbf{x} \in \mathcal{X} $ such that $ \hat{\mathbf{x}} \in B_\delta(\mathbf{x}) $. Because the clean instance $ \mathbf{x} $ is an asymptotically stable equilibrium point of ODE equation, we have
> $$
> \lim_{t \to +\infty} \|\mathbf{s}(\hat{\mathbf{x}}, t) - \mathbf{x}\| = 0.
> $$
>
> This theorem guarantees that if we make the clean instance $ \mathbf{x} $ an asymptotically stable equilibrium point, the ODE can reduce the perturbation and cause the perturbed instance to approach the clean instance. This can help improve the robustness of our detector DNN and aid it in defending against adversarial attacks.

---

> ### Author Response · Authors · 2024-11-27
>
> >**W1&Q1&W3 (c):**
>
>
> Here we demonstrate that it is possible to design a smooth and continuous function (similar to DNNs) which maps data points to specific stable points while ensuring the mapping possesses desired stability and derivative characteristics. This is crucial for ensuring that the outputs of NODE behave predictably and remain stable, thereby enhancing robustness. By carefully constructing this function and proving its existence under certain mathematical conditions, we offer a theoretical foundation for designing neural network components with these stability properties.
>
>
>
>
> ---
>
>
> We denote by $ \lambda $ the pushforward measure, which is a probability distribution derived from the original distribution of the input data under the continuous feature extractor mapping $ h_\phi $. The conditional probability distribution for the embeddings of each class in the training set, $ l \in \{1, \ldots, L\} $, has compact support $ E_l \subset \mathbb{R}^n $, as $ E_l $ is closed and $ h_\phi(X) $ is bounded in $ \mathbb{R}^n $.
>
> **Premise.** The input data are sampled from a probability distribution defined over a compact metric space. The feature extractor $ f_\theta $ is injective and continuous. Furthermore, the supports of each class in the embedding space are pairwise disjoint.
>
> **Lemma .** *Given $ k $ distinct points $ \mathbf{z}_i \in \mathbb{R}^n $ and matrices $ \mathbf{A}_i \in \mathbb{R}^{n \times n} $, for $ i = 1, \ldots, k $, there exists a function $ g \in \mathcal{C}^1(\mathbb{R}^n, \mathbb{R}^n) $ such that $ g(\mathbf{z}_i) = 0 $ and $ \nabla g(\mathbf{z}_i) = \mathbf{A}_i $.**
>
> **Proof:** The set of finite points $ \{z_1, \ldots, z_k\} $ is closed, and this lemma is an immediate consequence of the Whitney extension theorem [mcshane1934extension].
>
> We restrict $ h_\phi $ to be in $ C^1(\mathbb{R}^n, \mathbb{R}^n) $ to satisfy the condition in Theorem 1. $C^1$ represents the function with first-order derivative. From [hornik1991approximation], we also know that standard multilayer feedforward networks with as few as a single hidden layer and arbitrary bounded and non-constant activation functions are universal approximators for $ C^1(\mathbb{R}^n, \mathbb{R}^n) $ functions with respect to some performance criteria, provided only that sufficiently many hidden units are available.
>
> Suppose for each class $ l = 1, \ldots, L $, the embedding feature set $ E_l = \{z_1^{(l)}, \ldots, z_k^{(l)}\} $ is finite. For each $ i = 1, \ldots, k $, let $ A_i \in \mathbb{R}^{n \times n} $ be a strictly diagonally dominant matrix with every main diagonal entry negative, such that the eigenvalues of $ A_i $ all have negative real parts. From Theorem 3 (please refer to the corresponding theorem in the paper) , each $ A_i $ is non-singular and every eigenvalue of $ A_i $ has negative real part. Therefore, from Theorem 2 (please refer to the corresponding theorem in the paper)  and Lemma , there exists a function $ h_\phi $ such that all $ z_i^{(l)} $ are Lyapunov-stable equilibrium points with corresponding first derivative $ \nabla h_\phi(z_i^{(l)}) = A_i $. This shows that if there exist only finite representation points for each class, we can find a function $ h_\phi $ such that all inputs to the neural ODE layer are Lyapunov-stable equilibrium points for $ h_\phi $ and
>
>
> (I) $E_{\lambda} \|h_\phi(X_{\text{train}})\|_2 = 0 $,
>
> (II)  $ E_{\lambda} \big[\nabla h_\phi(X_{\text{train}})\big]_{ii} < 0 $,
>
>
> (III)  $  E_{\lambda} \bigg[ \sum_{i=1}^n (|[\nabla h_{\phi}(X_{\text{train}})]_{ii}| $
>
> $  - \sum_{j \neq i} |[\nabla h_{\phi}(X_{\text{train}})]_{ij}| ) \bigg] >0 . $

---

> ### Author Response · Authors · 2024-11-27
>
> >**W1&Q1&W3 (d):**
>
>
>
> We will show that under mild conditions, for all $ \epsilon > 0 $, we can find a continuous function $ h_\phi $ with finitely many stable equilibrium points such that conditions (II) and (III) above hold and condition (I) is replaced by $ E_{\lambda} \|h_\phi(X_{\text{train}})\|_2 < \epsilon $. This motivates the optimization constraints in (I, II, III).
>
> **Theorem B.** *Suppose **Premise** hold. If $ \lambda $ is not a continuous measure on $ E_l $ for each $ l = 1, \ldots, L $, then the following holds:*
> 1. *The function space satisfying the constraints in (I, II, III) is non-empty for all $ \epsilon > 0 $.*
> 2. *If additionally the restriction of $ \lambda $ to any open set $ O \subset E_l $ is a continuous measure, then we can find such a function such that each support $ E_l $ almost surely satisfies the conditions in (I, II, III).*
>
>
>
>
> **Proof:** Consider $ g(z) = [g^{(1)}(z^{(1)}), \ldots, g^{(n)}(z^{(n)})] $ with each $ g^{(i)}(z^{(i)}) \in C^1(\mathbb{R}, \mathbb{R}) $. Since $ g^{(i)}(z^{(i)}) $ depends only on $ z^{(i)} $, $ \nabla g_\theta(z) $ is a diagonal matrix with all off-diagonal elements being $ 0 $. The constraint (III) is thus immediately satisfied, and it suffices to show that there exists such an $ f $ satisfying the constraints (II) and (III).
>
> Select a point $ z_l = (z_l^{(1)}, \ldots, z_l^{(n)}) $ from the interior of each $ E_l $, for $ l = 1, \ldots, L $. Let $ g^{(i)}(z^{(i)}) = -\nu(z^{(i)} - z_l^{(i)}) $ on each $ E_l $, where $ \nu > 0 $. Then $ g(z) $ satisfies (II) for all $ \nu > 0 $, and $ z_l $ is a Lyapunov-stable equilibrium point for each $ l $ since $ \nabla h_\phi(z) $ is a diagonal matrix with negative diagonal values. Since each $ E_l \subset \mathbb{R}^n $ is compact, we have that $ \forall \epsilon > 0, \exists \nu > 0 $ sufficiently small such that $ |f^{(i)}(z^{(i)})| < \epsilon $ for all $ z \in \bigcup_l E_l $. The constraint (I) is therefore satisfied for $ f(z) $ with a sufficiently small $ \nu $. Since $ \bigcup_l E_l $ is closed, the Whitney extension theorem [mcshane1934extension] can be applied to extend $ f(z) $ to a function in $ C^1(\mathbb{R}^n, \mathbb{R}^n) $.
>
>
> ---
>
> The assumption that the pushforward measure $\lambda$​ in the feature space is not a continuous uniform measure is fair because real-world data and feature extraction processes naturally lead to non-uniform distributions. Real-world data often forms clusters based on underlying patterns or classes—images of cats cluster separately from images of dogs, for example. Feature extractors like neural networks amplify this clustering by mapping similar inputs to nearby points in the feature space, resulting in concentrated, non-uniform embeddings. In high-dimensional spaces, uniform distributions are impractical due to the curse of dimensionality, where data points become sparse and tend to reside on lower-dimensional manifolds rather than filling the space uniformly. Additionally, supervised learning aims for class separability, mapping different classes to distinct regions in the feature space, which inherently creates a non-uniform distribution. Therefore, assuming that  $\lambda$​   is not a continuous uniform measure aligns with both theoretical principles and practical observations, making it a reasonable and realistic assumption in deep learning.

---

> ### Author Response · Authors · 2024-11-27
>
> Dear Reviewer x9hq,
>
> Thank you again for your review and time. Please let us know if you have any further questions, and we hope our updated PDF and new experiments that address your good points now fully address your questions!
>
> Sincerely, The Authors

---

> > ### Comment · Reviewer_x9hq · 2024-11-27
> > **Reply to the authors**
> >
> > Thanks to the author's response, most of my concerns have been addressed, so I will maintain my score here.

---

> > > ### Author Response · Authors · 2024-11-27
> > >
> > > Dear Reviewer x9hq,
> > >
> > > Thank you for taking the time to review our manuscript and responses. As you have chosen to maintain your score, we would be happy to discuss and address any remaining concerns you might have.
> > >
> > > Sincerely,
> > > The Authors

---

### Official Review · Reviewer_w7tq · 2024-11-03

**Soundness:** 2
**Presentation:** 3
**Contribution:** 3
**Rating:** 5
**Confidence:** 4

**Summary:**

This paper proposes the Adversarially Robust OOD Detection through Stability (AROS) method, aimed at enhancing robustness in out-of-distribution (OOD) detection, particularly under adversarial conditions. The authors base their approach on Neural Ordinary Differential Equations (NODE) and Lyapunov stability theory, designing a time-invariant dynamic system to train the model for stability in perturbed environments. The AROS method includes adversarial training, pseudo-OOD embedding generation, Lyapunov stability regularization, and an orthogonal binary layer to construct a robust feature space that improves the separation between in-distribution (ID) and OOD samples. Experimental results demonstrate that AROS outperforms other OOD detection methods across multiple benchmark datasets (e.g., CIFAR-10, CIFAR-100, and ImageNet-1k), especially under strong adversarial attacks such as PGD, AutoAttack, and Adaptive AutoAttack.

**Strengths:**

1. The paper is clearly written and easy to understand.
2. The authors conducted extensive experiments to validate the effectiveness of their method.

**Weaknesses:**

1. The loss function $L_{SL}$ includes multiple hyperparameters, $\gamma_1$, $\gamma_2$, and $\gamma_3$. The choice of these hyperparameters affects the model's focus on different regularization terms, but directly showing them in the formula can seem verbose.
2. A detailed explanation of the elements in $[\nabla h_{\phi}(X_{\text{train}})]_{ii}$

and $[\nabla h_{\phi}(X_{\text{train}})]_{ij}$ is needed to clarify their computation.

3. In the section on the orthogonal binary layer, the equation $z^T z = I$ expresses the orthogonalization condition. However, it is unclear if $z$ has been normalized and whether its inner product yielding $I$ is the required condition.
4. The adversarial attack methods used in experiments are not the latest. The authors are encouraged to add 2023 and 2024 adversarial attack methods, particularly those with stronger transferability and generalization. See this work [1] for reference.
5. The authors did not provide the code for the experiments.

**Reference:**

[1] https://github.com/Trustworthy-AI-Group/TransferAttack

**Questions:**

1. In the experiments in Table 1, the authors used different model architectures. What is the reason for this choice? Could a unified model architecture be used to ensure fairness in method comparison?
2. Tables 2, 3, and 4 do not specify the model architectures used. Only OOD detection performance on different datasets is shown, which reduces the reproducibility of the paper.
3. In the text, $\epsilon$ is set to 8/255 and 4/255, but in Table 8a, the authors set $\epsilon$ to 128/255 and 64/255. Such high perturbation rates would cause noticeable distortion in samples. Please explain the significance of this setting. Could results for perturbation rates of 16/255 and 32/255, especially 16/255 (the default parameter in most adversarial attack methods), be added?
4. The authors simply list the experimental results for $\beta$ without providing in-depth analysis. For instance, in Table 11a, why are the results for $\beta = 0.001$ identical to those for $\beta = 0.025$? The three hyperparameters $\gamma_1$, $\gamma_2$, and $\gamma_3$ in the loss function $L_{SL}$ are also listed without further analysis.
5. Could some generated pseudo-OOD data samples be provided for readers to view?
6. I am curious whether the proposed method in the paper can be applied to the detection of backdoor attack samples.

---

> ### Author Response · Authors · 2024-11-22
>
> Dear Reviewer w7tq,
> Thanks for your constructive and valuable comments on our paper. Below, we provide a detailed response to your questions and comments. If any of our responses fail to address your concerns sufficiently, please inform us, and we will promptly follow up.
>
>
> >**Q1:**
>
> The architecture of our proposed method, AROS, is detailed in the implementation section of the manuscript (please see line 506). In our main experiments, we employ a WideResNet-70-16 model as the default backbone feature extractor.
>
> To benchmark AROS, we selected representative OOD detectors as competitors. Our criteria for choosing these methods fall into two categories: (1) methods designed for robust OOD detection (e.g., ALOE) and (2) methods known for high clean performance in OOD detection tasks (e.g., DHM).
>
>
> For each competitor, we adhered to their original architectures as described in their respective papers. For instance, CATEX integrates both text and image inputs using a CLIP-transformer backbone, making it unsuitable to replace its backbone with WideResNet due to its dependency on a text encoder. Notably, our competitors span a wide range of architectures, including transformers and CNNs, underscoring the superior robust performance of AROS across diverse methods and backbones.
> Additionally, we conducted an ablation study (Table 11a) to evaluate the impact of different backbone architectures on AROS, including PreActResNet, ResNet18, ResNet50, and ViT-B-16, while keeping other components fixed. The results demonstrate AROS’s consistent superiority and robustness across various backbone architectures.
>
>
> >**Q2:**
>
>
> Table 1 provides details of each method's backbone as specified in their respective papers. For the other tables, the backbone remains consistent with the information in Table 1, and we omitted repeating it to avoid redundancy. We have revised the manuscript to clarify this point and address the reviewers' concerns. (please see line 322)
>
>
> >**Q3:**
>
> The perturbation added to an input is constrained by a norm bound $ \epsilon $.
>
> For the $ L_\infty $ norm, which measures the maximum absolute change in pixel intensity, typical $ \epsilon $ values are often set to **0.015** or **0.03** (approximately $ \frac{8}{255} $). On the other hand, the $ L_2 $ norm measures the Euclidean distance between the original and perturbed inputs, aggregating changes across all dimensions. Consequently, $ \epsilon $ values for the $ L_2 $ norm are generally larger, often set to **0.25**, **0.5**, **1**, or even higher.
>
> Specifically, the relationship between the $ L_2 $ and $ L_\infty $ norms in $ n $-dimensional space is given by:
>
> $L_2= \sqrt{n} L_\infty$, where  $n$ represents the dimensionality.
>
>
>
>
> Our main results in the manuscript are reported based on the $ L_\infty $ norm (with $ \epsilon = \frac{8}{255} $), and to further demonstrate the robustness of our method, we also present the performance of AROS under the $ L_2 $ norm ($ \epsilon = \frac{128}{255} $) attack, as shown in Table 8-a. Moreover, in response to your concern, we included a figure in the manuscript illustrating different perturbation levels. This confirms that the chosen perturbation sizes neither alter the semantic meaning of the samples nor cause noticeable distortion. (please see figure 3)
>
>
>
>
>
>
>
>
> We also note that the performance of our method under the $L_\infty $ norm with $ \epsilon = \frac{16}{255} $ has been reported in Figure 1. To additionally address the reviewer’s concern, we have included the results of AROS under PGD-1000 $\ell_\infty $ with $ \epsilon = \frac{16}{255} $, which are provided here and can be found in Table 8-b of the revised manuscript.
>
> | Attack          | CIFAR-10        |                     |                     | CIFAR-100       |                     |                     | ImageNet-1k       |                     |                     |
> |-|-|-|-|-|-|-|-|-|-|
> |                  | CIFAR-100       | SVHN                | LSUN                | CIFAR-10        | SVHN                | LSUN                | Texture           | iNaturalist         | LSUN                |
> | $\text{PGD}^{1000}(l_{\infty}), \epsilon = \frac{16}{255} $| 70.3           | 78.4                | 72.6                | 58.2           | 63.9                | 60.4                | 60.5              | 66.8                | 62.7                |
>
>
> >**W1:**
>
> We understand the reviewers' concern regarding verbosity and have addressed it by providing additional clarification  and intuition (e.g. Section A.3.3.2) for each term in the loss function, along with further analysis. For more information on the choice of hyperparameters, we kindly direct you to our response to Weakness 4.

---

> ### Author Response · Authors · 2024-11-22
>
> >**W2:**
>
>
>  Considering our loss function: $$
> L_{SL} = \min_{\phi, \eta} \frac{1}{\left| X_{\text{train}} \right|} \Bigg(\ell_{\text{CE}}(  B_{\eta} (h_{\phi}(X_{\text{train}}), y)) + \gamma_1 \| \| h_{\phi}(X_{\text{train}})\|\|_2
> $$
>
> $$
> \quad \quad \quad \quad \quad \quad + \gamma_2 \text{exp} \bigg(-\sum_{i=1}^n [\nabla h_{\phi}(X_{\text{train}})]_{ii} \bigg)
> $$
>
> $$
> \quad \quad \quad \quad \quad \quad \quad \quad \quad \quad \quad \quad + \gamma_3 \text{exp} \bigg( \sum_{i=1}^n (-|[\nabla h_{\phi}(X_{\text{train}})]_{ii}|
> $$
>
> $$
> \quad \quad \quad \quad \quad \quad \quad \quad \quad \quad \quad \quad + \sum_{j \neq i} |[\nabla h_{\phi}(X_{\text{train}})]_{ij}| ) \bigg) \Bigg)
> $$
>
> in our method, $h_\phi$ represents the dynamics function of the Neural Ordinary Differential Equation (NODE), which maps the input vector $X_{\text{train}}$ to a new representation in the feature space. The Jacobian matrix $\nabla h_\phi(X_{\text{train}})$ is the matrix of all first-order partial derivatives of $h_\phi$ with respect to $X_{\text{train}}$. Specifically, each element of the Jacobian is defined as $[\nabla h_\phi(X_{\text{train}})]_{ij}=$:
>
> $ = \frac{\partial h_{\phi, i}(X_{\text{train}})}{\partial [X_{\text{train}}]_j}$
>
>
>
>
>
> $X_{\text{train}}$
> where:
>
>
> $h_{\phi, i}(X_{\text{train}})$ is the $i$-th component of the output vector $h_\phi(X_{\text{train}})$ and $[X_{\text{train}}]_j$ is the $j$-th component of the input vector.
>
>
>  ($X_{\text{train}}$ denotes the input vector)
>
>
>
>
> The diagonal elements of the Jacobian matrix represent the sensitivity of each output component to its corresponding input component. Mathematically, for each $i$,   $[\nabla h_\phi(X_{\text{train}})]_{ii} $would be :=
>
> $=\frac{\partial h_{\phi, i}(X_{\text{train}})}{\partial [X_{\text{train}}]_i}$
>
> These terms quantify how a small change in the $i$-th input dimension affects the $i$-th output dimension. In the context of Lyapunov stability, the sum of these diagonal elements is critical because it relates to the trace of the Jacobian, which influences the real parts of the eigenvalues.
>
>
> The off-diagonal elements capture the cross-interactions between different input and output dimensions. i.e. $[\nabla h_\phi(X_{\text{train}})]_{ij}$ :=
>
>
> $\frac{\partial h_{\phi, i}(X_{\text{train}})}{\partial [X_{\text{train}}]_j}, \quad \text{for } i \neq j$.
>
>
>
> These terms measure how changes in the $j$-th input dimension influence the $i$-th output dimension. In our loss function, we sum the absolute values of these off-diagonal elements to assess the degree of coupling between different dimensions.
>
>
>
> ### Role in the Loss Function ($\mathcal{L}_{\text{SL}}$)
>
> In Equation (3) of our paper, we incorporate these elements into two regularization terms to enforce stability conditions based on the Lyapunov stability theorem:
>
> 1. **Encouraging Negative Diagonal Elements**
>
> $\gamma_2  \text{exp} ( -\sum_{i=1}^n [\nabla h_\phi(X_{\text{train}})]_{ii}  )$
>
>  This term penalizes the sum of the diagonal elements if they are not sufficiently negative. By promoting negative diagonal elements, we encourage the eigenvalues of the Jacobian to have negative real parts, which is essential for asymptotic stability according to Theorem 2 (Lyapunov Stability Theorem).
>
> 2. **Enforcing Strict Diagonal Dominance**
>
>  $$
>   \gamma_3 \text{exp} \Bigg( \sum_{i=1}^n (-|[\nabla h_{\phi}(X_{\text{train}})]_{ii}|
> $$
>
>
> $$
>  \quad \quad \quad \quad \quad + \sum_{j \neq i} |[\nabla h_{\phi}(X_{\text{train}})]_{ij}| ) \Bigg)
> $$
>
>  This term ensures that the magnitude of each diagonal element dominates the sum of the magnitudes of the off-diagonal elements in the same row, satisfying the strict diagonal dominance condition required by Theorem 3 (Levy--Desplanques Theorem). This condition further guarantees that all eigenvalues of the Jacobian have negative real parts, contributing to the system's stability.
>
> ### Practical Computation
>
> In practice, we compute $\nabla h_\phi(X_{\text{train}})$ using automatic differentiation tools available in deep learning frameworks (i.e., PyTorch). For each batch of input embeddings $X_{\text{train}}$:
>
> 1. **Forward Pass**
>    - Compute the output $h_\phi(X_{\text{train}})$.
>
> 2. **Jacobian Computation**
>    - Use automatic differentiation to compute the Jacobian matrix $\nabla h_\phi(X_{\text{train}})$.
>
>
>    - Extract the diagonal elements $[\nabla h_\phi(X_{\text{train}})]_{ii}$.
>
>    - Extract the off-diagonal elements $[\nabla h_\phi(X_{\text{train}})]_{ij}$.
>
> 3. **Loss Calculation**
>
>    - Evaluate the regularization terms in $\mathcal{L}_{\text{SL}}$ using the computed Jacobian elements.
>
>
>    - Combine with the cross-entropy loss and other terms to obtain the total loss.
>
> 4. **Backward Pass**
>    - Backpropagate the loss to update the network parameters $\phi$ and $\eta$.

---

> ### Author Response · Authors · 2024-11-22
>
> >**W3:**
>
> Thank you for bringing this point to our attention. There appear to be issues stemming from a typographical error in the equation you mentioned. Specifically, the term "output" should be replaced with "weights," and it is important to clarify that $Z$ in the context corresponds to the layer weights rather than the output.
>
> For the orthogonal binary layer, the orthogonality condition is applied to the weights of the orthogonal layer, not to the output of the Neural Ordinary Differential Equation (NODE). Additionally, we note that the output from the NODE is not normalized before being passed through the orthogonal layer.
>
>
> We have corrected this error and incorporated additional experiments and visualizations to further support the orthogonal binary layer. Please refer to Section A3.6 for more details.
>
> We apologize for any confusion caused by this typographical error and hope this clarification resolves your concern.
>
>
> >**W4:**
>
> We evaluated AROS against various strong and commonly used attacks, including PGD-1000 and the Adaptive Auto Attack (Liu et al., CVPR 2022), in comparison to previous works. To address reviewer concerns, we conducted additional experiments, specifically focusing on transfer-based attacks.
>
> To address the reviewer's concern, we evaluated our method, along with DHM and RODEO, against transfer-based attacks, including DeCoWA (Lin et al., AAAI 2024), DTA (Yang et al., NeurIPS 2023), and SASD-WS (Wu et al., CVPR 2024), utilizing the repository suggested by the reviewer. The results of these experiments are presented in Table 9b of the paper and are also included here. As the results demonstrate, AROS achieves superior performance under transfer attacks.
>
> We selected DHM and RODEO for comparison due to their strong performance in clean detection and robust detection, respectively.
>
>
> | Attack   | Methods | CIFAR-10 |        |        | CIFAR-100 |        |        | ImageNet-1k |          |        |
> |-|-|-|-|-|-|-|-|-|-|-|
> |          |         | CIFAR-100 | SVHN   | LSUN   | CIFAR-10  | SVHN   | LSUN   | Texture     | iNaturalist | LSUN  |
> | DTA      | AROS    | **84.7** | **86.4**   | 88.2   | **71.5**  | **76.6**   | **71.7**   | **76.2**    | **83.5**  | **75.9**  |
> |          | RODEO   | 61.6     | 68.5   | **89.3** | 51.1     | 63.1   | 70.9   | 61.1        | 60.6      | 59.2  |
> |          | DHM     | 72.9     | 78.3   | 78.9   | 75.4      | **79.1** | 74.1   | 55.9        | 59.2      | 57.6  |
> | DeCoWA   | AROS    | **83.8** | **85.2**   | **86.8**   | **69.6**  | **77.2** | **69.4**   | **72.6**    | **79.5**  | **74.8** |
> |          | RODEO   | 57.8     | 64.6   | 80.4   | 43.0      | 57.7   | 67.5   | 54.7        | 52.5      | 52.5  |
> |          | DHM     | 60.1     | 64.3   | 64.9   | 68.9      | 62.1   | 60.8   | 54.9        | 58.4      | 56.0  |
> | SASD-WS  | AROS    | **82.3** | **84.9**   | **81.7**   | **67.6**  | **75.0** | **67.7** | **72.4**    | **77.5**  | **71.2** |
> |          | RODEO   | 49.5     | 53.4   | 79.4   | 44.7      | 56.5   | 66.1   | 53.2        | 46.9      | 47.0  |
> |          | DHM     | 64.6     | 69.3   | 61.3   | 56.3      | 67.8   | 59.7   | 53.1        | 51.3      | 49.2  |
>
>
>
>
> >**W5:**
>
> We would like to clarify that the code for our experiments has been included in our submission. Please kindly refer to the attached zip file for details.

---

> ### Author Response · Authors · 2024-11-22
>
> >**Q4&W1-a:**
>
> >The authors simply list the experimental results for $\beta$ without providing in-depth analysis. For instance, in Table 11a, why are the results for $\beta = 0.001$ identical to those for $\beta = 0.025$?
>
>
>
> We sincerely apologize for the oversight in Table 11a. The identical results for $\beta = 0.001$ and $\beta = 0.025$ were due to an editorial mistake. This has been corrected, and the table now accurately reflects the correct values. Thank you for bringing this to our attention. The corrected results, along with an extended version of the table, have been provided to address the mentioned concern. Additionally, we note that the revised manuscript now includes a detailed hyperparameter ablation study in Tables 12a and 13a.
>
>
>
>
>
>
>
>
> In the following, we provide a detailed response regarding the hyperparameters of our method:
>
>
>
>
> **Hyperparameter $\beta$:**
>
>
> To craft fake samples, we fit a GMM to the embedding space of ID samples. The objective is to sample from the GMM such that the likelihood of the samples is low, ensuring they do not belong to the ID distribution and are located near its boundaries. This approach generates near-OOD samples, which are valuable for providing insights into the distribution manifold and improving the detector's performance. To achieve this, we introduced the following equation in the manuscript:
>
> $$
> \frac{1}{(2\pi)^{d/2} |\hat{\Sigma}_j|^{1/2}} \exp\left( -\frac{1}{2} (r - \hat{\mu}_j)^{T} \hat{\Sigma}_j^{-1} (r - \hat{\mu}_j) \right) < \beta,
> $$
>
> where
>
> * $r$: The embedding vector of a sample in the feature space.
>
>  * $d$: The dimensionality of the embedding vector $r$.
>
>   *   $\hat{\mu}_j$: The mean vector of the $j$-th class in the embedding space.
>
>    * $\hat{\Sigma}_j$: The covariance matrix of the $j$-th class in the embedding space.
>
>   *   $|\hat{\Sigma}_j|$: The determinant of the covariance matrix for the $j$-th class.
>
>    *  $\hat{\Sigma}_j^{-1}$: The inverse of the covariance matrix for the $j$-th class.
>
>    * $\beta$: A small threshold value indicating low-likelihood regions of the ID distribution.
>
> In practice, we use an encoder as a feature extractor to obtain the embeddings of ID samples. A GMM is then fitted to these embeddings, and the likelihood of each training sample under the GMM is computed. Next, we sort the embeddings based on their likelihoods and use the $\beta$-th minimum likelihood of the training samples as a threshold. We then randomly sample from the GMM and compare the likelihood of the sampled data against this threshold. If the likelihood is lower than the threshold, the sampled data is retained as fake OOD; otherwise, it is discarded.
>
> For example, setting $\beta = 0.1$ means that the crafted fake OOD samples will have a likelihood of belonging to the ID distribution that is lower than 90\% of the ID samples. An ablation study provided in our manuscript explores the effects of different $\beta$ values.
>
> To address reviewer concerns and provide further insights, we have expanded this study and included the results here:
>
>
> | Hyperparameter | **CIFAR-10**        |           |           | **CIFAR-100**       |           |           | **ImageNet-1k**     |           |           |
> |-|-|-|-|-|-|-|-|-|-|
> |                | **CIFAR-100**      | **SVHN**  | **LSUN**  | **CIFAR-10**        | **SVHN**  | **LSUN**  | **Texture**         | **iNaturalist** | **LSUN** |
> |  $\beta$= 0.001 *(default)* | 88.2/80.1         | 93.0/86.4 | 90.6/82.4 | 74.3/67.0         | 81.5/70.6 | 74.3/68.1 | 78.3/69.2          | 84.6/75.3       | 79.4/69.0 |
> |  $\beta$ = 0.01            | 87.3/80.2         | 92.2/85.3 | 89.4/80.7 | 74.7/65.8         | 80.2/68.9 | 73.1/66.6 | 78.1/69.5          | 83.0/74.2       | 79.0/68.5 |
> |  $\beta$ = 0.025            | 86.3/79.6         | 93.8/85.0 | 89.3/83.1 | 72.7/67.5         | 81.7/70.8 | 74.3/68.0 | 76.7/67.7          | 84.7/75.3       | 79.8/69.7 |
> |  $\beta$ = 0.05             | 86.5/80.1         | 92.5/86.1 | 90.8/82.7 | 74.5/66.4         | 81.7/69.6 | 72.8/68.3 | 77.9/67.4          | 82.7/75.1       | 77.9/68.5 |
> | $\beta$ = 0.075            | 88.4/78.4         | 92.7/85.2 | 89.1/82.3 | 72.4/65.7         | 81.5/69.4 | 72.5/67.9 | 78.6/69.7          | 84.8/73.5       | 78.8/69.3 |
> | $\beta$ = 0.1            | 86.6/79.5         | 92.0/85.4 | 90.9/80.9 | 72.3/65.8         | 80.3/69.4 | 73.3/66.7 | 77.8/67.5          | 84.1/74.9       | 78.2/67.8 |
> |  $\beta$ = 0.25            | 79.4/67.5         | 89.0/79.5 | 78.7/69.4 | 67.7/54.3         | 69.5/60.3 | 70.1/62.7 | 62.6/65.0          | 65.2/57.9       | 70.9/62.0 |
> |  $\beta$ = 0.5              | 65.4/53.0         | 77.4/62.3 | 64.7/54.6 | 57.5/40.9         | 52.4/49.2 | 49.1/41.9 | 55.8/48.9          | 67.4/59.7       | 57.9/53.8 |

---

> ### Author Response · Authors · 2024-11-22
>
> >**Q4&W1-b:**
>
> As the results show, AROS demonstrates consistent performance for small values of $\beta$. Notably, setting $\beta = 0.5$ corresponds to randomly sampling from the ID distribution instead of focusing on low-likelihood regions, which, as the results indicate, leads to poor performance. Based on our findings, we recommend choosing $\beta$ values in the range $[0.0, 0.1]$ to achieve the desired objective.
>
>
>
> **Hyperparameters $\gamma_1$, $\gamma_2$, $\gamma_3$:**
>
> In our proposed method, we introduce the empirical loss function $\mathcal{L}_{\text{SL}}$ as follows:
>
> $$
> L_{SL} = \min_{\phi, \eta} \frac{1}{\left| X_{\text{train}} \right|} \Bigg(\ell_{\text{CE}}(  B_{\eta} (h_{\phi}(X_{\text{train}}), y)) + \gamma_1 \| \| h_{\phi}(X_{\text{train}})\|\|_2
> $$
>
> $$
> \quad \quad \quad \quad \quad \quad + \gamma_2 \text{exp} \bigg(-\sum_{i=1}^n [\nabla h_{\phi}(X_{\text{train}})]_{ii} \bigg)
> $$
>
> $$
> \quad \quad \quad \quad \quad \quad \quad \quad \quad \quad \quad \quad + \gamma_3 \text{exp} \bigg( \sum_{i=1}^n (-|[\nabla h_{\phi}(X_{\text{train}})]_{ii}|
> $$
>
> $$
> \quad \quad \quad \quad \quad \quad \quad \quad \quad \quad \quad \quad + \sum_{j \neq i} |[\nabla h_{\phi}(X_{\text{train}})]_{ij}| ) \bigg) \Bigg)
> $$
>
> ### Here:
> - $X_{\text{train}}$ is the set of training embeddings.
> - $y$ are the corresponding labels.
> - $h_{\phi}$ is the NODE function parameterized by $\phi$.
> - $B_{\eta}$ is the orthogonal binary layer parameterized by $\eta$.
> - $\ell_{\text{CE}}$ denotes the cross-entropy loss.
> - $\nabla h_{\phi}(X_{\text{train}})$ is the Jacobian matrix of $h_{\phi}$ with respect to $X_{\text{train}}$.
>
> ---
>
> **Roles of the Hyperparameters**
>
> ### 1. $\gamma_1$
> Controls the regularization term $\| h_{\phi}(X_{\text{train}})\|_2$, which encourages the system's state to be near the equilibrium point. This term helps mitigate the effect of perturbations by ensuring that trajectories remain close to the equilibrium.
>
> ### 2. $\gamma_2$ and $\gamma_3$
> These hyperparameters weight the exponential terms designed to enforce Lyapunov stability conditions by ensuring the Jacobian matrix satisfies strict diagonal dominance (as per Theorem 3 in the paper).
>
> - **$\gamma_2$** weights the term $\exp\left( -\sum_{i=1}^n [\nabla h_{\phi}(X_{\text{train}})]_{ii} \right)$, encouraging the diagonal entries of the Jacobian to be negative with large magnitudes.
>
> - **$\gamma_3$** weights the term $\text{exp} \bigg( \sum_{i=1}^n (-|[\nabla h_{\phi}(X_{\text{train}})]_{ii}| \bigg)$
>
>   promoting strict diagonal dominance by penalizing large off-diagonal entries relative to the diagonal entries.
>
> ---
>
> **Reasoning Behind Default Values**
>
> - **$\gamma_1 = 1$**: Balances the influence of the regularization term with the primary cross-entropy loss. This ensures sufficient regularization without overwhelming the classification objective.
> - **$\gamma_2 = \gamma_3 = 0.05$**: Preliminary experiments indicate these values effectively enforce stability without causing numerical issues or hindering optimization. We choose $\gamma_2 = \gamma_3 = 0.05$ Inspired by related works [1-10] that incorporate stability terms into the training process.
>
> ---
>
> **Why $\gamma_2$ and $\gamma_3$ Are Equal**
>
> 1. **Balanced Contribution**: Both regularization terms serve complementary purposes in enforcing the stability conditions. Equal weighting ensures neither term dominates, maintaining a balanced emphasis on both negative diagonal dominance and strict diagonal dominance.
> 2. **Simplified Hyperparameter Tuning**: Setting $\gamma_2$ and $\gamma_3$ equal reduces the hyperparameter search space, simplifying the tuning process without sacrificing performance.
>
>
> ---
>
> **Impact of Hyperparameters on Performance**
>
> **Effect of $\gamma_1$**
>
>
> - **Low Values ($\gamma_1 < 1$)**: Reduces the emphasis on keeping the state near the equilibrium, making the system more susceptible to perturbations.
> - **High Values ($\gamma_1 > 1$)**: Overly constrains the state to the equilibrium point, potentially limiting the model's capacity to learn discriminative features.
> - **Chosen Value ($\gamma_1 = 1$)**: Offers a good balance, ensuring sufficient regularization without compromising learning.
>
> ---
>
> **Effect of $\gamma_2$ and $\gamma_3$**
>
>
>
> - **Low Values ($< 0.05$)**: Diminish the impact of the stability constraints, reducing robustness.
> - **High Values ($> 0.05$)**: Overemphasize the stability terms, potentially hindering the optimization of the primary classification loss.
> - **Equal Values ($\gamma_2 = \gamma_3$)**: Ensure balanced enforcement of both stability conditions, leading to optimal performance.

---

> ### Author Response · Authors · 2024-11-22
>
> >**Q4&W1-c:**
>
>
> To address the reviewers' concerns, we conducted extensive ablation studies, keeping all components fixed while varying the values of $\gamma_1$, $\gamma_2$, and $\gamma_3$. These experiments demonstrate that AROS consistently performs well across a broad range of hyperparameter values, including extreme cases (e.g., $\gamma_1 = 2$). The results of this analysis are presented in Table 13a of our paper and are also included below. Notably, this table was part of our submitted manuscript, and we have now extended it with additional experiments to further address any remaining concerns. These results confirm the robust performance of AROS under varying configurations of $\gamma_1$, $\gamma_2$, and $\gamma_3$.
>
>
>
> **Hyperparameters $\gamma_1$, $\gamma_2$, $\gamma_3$:**
>
>
> | Hyperparameter                     | **CIFAR-10**        |           |           | **CIFAR-100**       |           |           | **ImageNet-1k**     |           |           |
> |-|-|-|-|-|-|-|-|-|-|
> |                                    | **CIFAR-100**      | **SVHN**  | **LSUN**  | **CIFAR-10**        | **SVHN**  | **LSUN**  | **Texture**         | **iNaturalist** | **LSUN** |
> | **$\gamma_1=1, \gamma_2=\gamma_3=0.05$ (default)**     | 88.2/80.1          | 93.0/86.4 | 90.6/82.4 | 74.3/67.0           | 81.5/70.6 | 74.3/68.1 | 78.3/69.2          | 84.6/75.3       | 79.4/69.0 |
> | **$\gamma_1=1, \gamma_2=\gamma_3=0.025$**              | 86.4/78.2          | 90.4/86.3 | 88.6/80.4 | 74.9/64.1           | 78.8/68.7 | 72.8/68.3 | 79.2/69.7          | 83.3/76.0       | 78.8/67.5 |
> | **$\gamma_1=1, \gamma_2=\gamma_3=0.075$**              | 87.5/80.3          | 92.1/87.2 | 88.9/81.6 | 72.7/67.7           | 80.8/68.7 | 73.2/68.8 | 76.4/69.9          | 83.4/73.4       | 80.2/69.9 |
> | **$\gamma_1=1, \gamma_2=\gamma_3=0.1$**                | 86.9/78.6          | 93.6/82.1 | 85.8/78.2 | 70.0/65.8           | 80.6/66.2 | 72.6/67.3 | 79.3/69.6          | 82.8/74.0       | 78.9/67.2 |
> | **$\gamma_1=1, \gamma_2=\gamma_3=0.25$**               | 86.2/78.7          | 94.3/82.3 | 87.1/77.9 | 70.2/65.2           | 81.3/65.8 | 73.9/66.7 | 78.6/70.1          | 82.8/74.1       | 78.9/68.2 |
> | **$\gamma_1=1, \gamma_2=\gamma_3=0.5$**                | 85.2/76.9          | 93.6/81.5 | 85.0/76.0 | 69.7/64.2           | 78.1/64.4 | 72.6/65.2 | 77.0/66.7          | 83.6/74.4       | 78.2/67.3 |
> | **$\gamma_1=\gamma_2=\gamma_3=1$**                     | 84.4/75.3          | 90.5/80.5 | 81.5/73.4 | 69.8/62.2           | 77.5/65.3 | 70.1/64.4 | 73.5/67.3          | 82.7/70.9       | 76.7/65.4 |
> | **$\gamma_1=\gamma_2=\gamma_3=0.25$**                  | 85.4/77.4          | 94.6/82.9 | 86.3/76.9 | 69.9/64.4           | 79.9/65.1 | 74.4/65.3 | 77.2/68.6          | 85.4/76.0       | 79.2/66.8 |
> | **$\gamma_1=0.5, \gamma_2=\gamma_3=0.05$**             | 86.0/78.6          | 92.7/82.5 | 86.3/77.7 | 69.7/66.1           | 81.2/65.5 | 70.9/67.1 | 78.8/68.6          | 83.0/73.7       | 78.8/67.1 |
> | **$\gamma_1=1, \gamma_2=0.1, \gamma_3=0.05$**          | 84.0/76.3          | 91.5/81.5 | 81.7/76.4 | 70.1/63.4           | 76.2/63.8 | 69.8/67.7 | 77.4/69.5          | 83.2/71.0       | 78.5/64.8 |
> | **$\gamma_1=1, \gamma_2=0.05, \gamma_3=0.1$**          | 83.5/77.5          | 90.2/78.0 | 81.4/75.1 | 67.2/63.5           | 76.7/66.2 | 67.5/60.4 | 74.3/70.0          | 83.6/74.1       | 74.2/61.7 |
> | **$\gamma_1=2, \gamma_2=0.5, \gamma_3=0.5$**           | 80.4/74.0          | 93.5/80.7 | 81.8/70.9 | 65.2/66.0           | 77.3/60.0 | 65.0/60.1 | 78.4/63.3          | 80.3/70.6       | 71.6/59.5 |
>
>
> ---
>
> ---
>
>
>
> [1] Zonghan Yang,  . A closer look at the a . Advances in Neural Information Processing Systems, 35:10448–10461, 2022.
>
> [2] Ivan Dario Jimenez Rodriguez, Aaron Ames,  . Lyanet: A lyapunov framework for training neural odes. In International conference on machine learning,
>
> [3] Xiyuan Li, Zou Xin, and Weiwei Liu. Defending against adversarial attacks via  . Advances in Neural Information Processing Systems,
>
> [4] Qiyu Kang, et al Stable neural ode with lyapunov-stable equilibrium points for defending against adversarial attacks. Advances in Neural Information Processing Systems
>
> [5] Fabio Carrara et al On the robustness to adversarial examples of neural ode image classifiers
>
> [6] Jan Svoboda, Jonathan, Federico Monti, Michael Bronstein, and Leonidas Guibas. Peernts: Exploiting peer wisdom against adversarial attacks. In International Conference on Learning Representations, 2019.
>
> [7] Arash Rahnama,  et al. Robust design of deep neural networks against adversarial attacks based on lyapunov theory.  Computer Vision and Pattern Recognition,
>
> [8] Mingjie Li, Lingshen He, and Zhouchen Lin. Implicit euler skip connections: Enhancing adversarial robustness via  PMLR, 2020.
>
>
> [9] Sergey Dashkovskiy, Oleksiy Kapustyan, and Vitalii Slynko. Robust stability of a nonlinear ode-pde system.
>
> [10] Mustafa Zeqiri, Efficient certified training and robustness verification of neural odes.

---

> ### Author Response · Authors · 2024-11-22
>
> >**Q5:**
>
> Although AROS is a robust OOD detection method in the image domain, it operates by crafting fake OOD data in the embedding space (i.e., vectors) rather than in the input image space. Consequently, directly visualizing these vectors is challenging. However, to address the reviewer’s concern, we utilized dimensionality reduction techniques (i.e. t-SNE) to project the fake data into a 2D space for visualization. Please refer to Figure 5,6 in the appendix.
>
>
>
> >**Q6:**
>
> To address the reviewers' concerns, we conducted an experiment where our method was evaluated on detecting clean samples (without triggers) as in-distribution versus poisoned samples (with triggers) as out-of-distribution. The results are presented here as well as Table 10a. We included STRIP (Gao et al. 2019) as a baseline method for identifying Trojaned samples.
>
> It is important to note that in the literature on OOD detection, which forms the foundation of our study, in-distribution (ID) and OOD datasets generally differ semantically (e.g., CIFAR-10 versus SVHN). In contrast, spotting backdoored samples involves the addition of triggers to images, causing them to differ from clean samples at the pixel level. This distinction primarily reflects a difference in texture rather than semantics and typically requires a model specifically designed for this task to effectively leverage such an inductive bias.
>
> Nonetheless, as our results demonstrate, AROS achieves satisfactory performance in detecting poisoned samples, underscoring its effectiveness even in this challenging scenario.
>
>
>
>
>
>
>
> *Used metric:AUROC%*
> | Method | Backdoor Attack | CIFAR-10 | CIFAR-100 | GTSRB |
> |-|-|-|-|-|
> | **AROS** | Badnets        | **80.3** | 67.5      | 72.8  |
> |        | Wanet          | **62.7** | **58.9**  | 54.4  |
> |        | SSBA           | **57.2** | 72.6      | 66.0  |
> |-|-|-|-|-|
> | **STRIP** | Badnets       | 79.2     | **86.0**  | **87.1** |
> |        | Wanet          | 39.5     | 48.5      | 35.6  |
> |        | SSBA           | 36.4     | 68.5      | 64.1  |

---

> ### Author Response · Authors · 2024-11-26
> **Reviewer Feedback Requested**
>
> Dear Reviewer w7tq,
>
> thank you again for your review. We wanted to check in to see if there are any further clarifications we can provide. We hope our updated PDF with new experiments and explanations help convince you this work is above the threshold.
>
> Best,
> the authors

---

> > ### Author Response · Authors · 2024-12-02
> > **Feedback requested**
> >
> > Dear w7tq,
> >
> > Apologies for pinging you again, but we are eager to receive your feedback before the rebuttal period ends. Briefly, we have included extensive new experiments, extensive ablation studies, new baselines, updated figures & text, and demonstrated that AROS continues to perform favorably. Your inputs were invaluable, and we hope the new experiments and manuscript revisions enable you to reassess your score.
> >
> > Thank you!

---

### Official Review · Reviewer_e2wJ · 2024-11-03

**Soundness:** 2
**Presentation:** 3
**Contribution:** 3
**Rating:** 6
**Confidence:** 4

**Summary:**

This paper introduces a novel adversarially robust out-of-distribution (OOD) detection method, AROS, which leverages Neural Ordinary Differential Equations (NODE) and Lyapunov stability theory to enhance model stability against adversarial perturbations. AROS employs a pseudo-OOD data generation strategy, sampling low-likelihood regions in the embedding space to create pseudo-OOD samples, thereby improving the model's generalization capability. Additionally, it incorporates an orthogonal binary layer to strengthen the separation between in-distribution (ID) and OOD samples. Experiments on CIFAR-10, CIFAR-100, and ImageNet-1k demonstrate that AROS maintains high detection accuracy across various attacks, including strong adversarial PGD and adaptive AutoAttack. Ablation studies indicate that Lyapunov stability regularization and pseudo-OOD embedding generation are particularly important for enhancing robustness.

**Strengths:**

1. The paper proposes an OOD detection method based on Lyapunov stability and NODE, representing an innovative attempt to integrate control theory with OOD detection to improve model robustness under adversarial perturbations.
2. The proposed orthogonal binary layer is effective in enhancing the separation of ID and OOD samples.

**Weaknesses:**

1. In the formula for the pseudo-embedding generation strategy, $p(r | y = j)$ denotes the conditional probability of generating pseudo-OOD embeddings. Here, the conditional probability notation $p(\cdot | \cdot)$ is used to describe feature vectors that meet a given condition, but in reality, it defines an inequality rather than a complete probabilistic expression. To avoid misunderstandings, it is recommended to rewrite this as an inequality, preventing the impression that a probability density function is defined here.

2. The definition and application of the orthogonal binary layer $B_{\eta}$ are somewhat complex. This layer aims to enhance the separation of ID and OOD samples by enforcing orthogonality in specific directions of the feature space. However, in the formula $L_{SL}$, directly using $B_{\eta}$ as a layer with multiple parameters may cause confusion, especially when multiple hyperparameters are involved. The expression might be clearer if $B_{\eta}$ is decomposed or its function is clearly annotated.

3. The model architecture used in the experimental section is not clearly described in the paper. Please clarify the model architecture.

4. Most of the models used in the experiments are based on traditional CNN architectures. Could the authors consider using Transformer-based models to evaluate performance differences between different methods?

5. It seems the experiments use untargeted attacks. Has the author considered the impact of targeted attacks on OOD detection?

6. In Section A3.2, "Extended Ablation Study," the authors provide results from ablation experiments but do not analyze the reasons for choosing the default parameters in the paper or discuss how different parameters might affect the results.

**Questions:**

See Weaknesses.

---

> ### Author Response · Authors · 2024-11-22
>
> Dear Reviewer  e2wJ,
>
> We sincerely appreciate your valuable feedback on our paper. Below, we present detailed responses to your comments:
>
> >**W1:**
>
> Thank you for pointing out the potential ambiguity in our use of $p(r∣y=j)$. We agree that the current notation could be misinterpreted as defining a conditional probability density function, while in reality, it describes an inequality condition for selecting pseudo-OOD embeddings. To clarify this and avoid any misunderstanding, we have revised the corresponding text and notation in the manuscript
>
>
> >**W2:**
>
>
> Thank you for your suggestion. We have revised our manuscript based on your comments. Specifically, we have both decomposed and further clarified the functionality $B_{\eta}$, and we have also included additional supportive experiments and visualizations for the orthogonal binary layer in the manuscript. Kindly refer to Section A3.6 for the details. Our explanation for further clarification is as follows:
>
> The orthogonal binary layer $B_{\eta}$ is designed to apply a transformation to the NODE output $h_{\phi}(z)$, where the weights $W$ of the layer are constrained to be orthogonal ($W^T W = I$). This constraint encourages maximal separation between the equilibrium points of ID and OOD data by ensuring that the learned representations preserve distinct directions in the feature space. Specifically, the layer operates as:
>
> $$
> B_{\eta}(z) = Wz + b, \text{ subject to } W^T W = I,
> $$
>
> where $W$ represents the weight matrix, $b$ is the bias term, and the orthogonality constraint is enforced during training using a regularization term.
>
> To clarify its role in $L_{SL}$, we now explicitly annotate $B_\eta$ as the mapping responsible for projecting the NODE output $h_{\phi}(z)$ into a binary classification space (ID vs. OOD).
>
> The $L_{SL}$ with explicit reference to $W$ and $b$ would be as follows:
>
> $$
> L_{SL} = \min_{\phi, w} \frac{1}{\left| X_{\text{train}} \right|} \Bigg(\ell_{\text{CE}}( (W h_{\phi}(X_{\text{train}}) + b), y) + \gamma_1 \| \| h_{\phi}(X_{\text{train}})\|\|_2
> $$
>
> $$
> \quad \quad \quad \quad \quad \quad + \gamma_2 \text{exp} \bigg(-\sum_{i=1}^n [\nabla h_{\phi}(X_{\text{train}})]_{ii} \bigg)
> $$
>
> $$
> \quad \quad \quad \quad \quad \quad \quad \quad \quad \quad \quad \quad + \gamma_3 \text{exp} \bigg( \sum_{i=1}^n (-|[\nabla h_{\phi}(X_{\text{train}})]_{ii}|
> $$
>
> $$
> \quad \quad \quad \quad \quad \quad \quad \quad \quad \quad \quad \quad + \sum_{j \neq i} |[\nabla h_{\phi}(X_{\text{train}})]_{ij}| ) \bigg) \Bigg)
> $$
>
> where the orthogonality constraint $W^T W = I$ is enforced via regularization during optimization.

---

> ### Author Response · Authors · 2024-11-22
>
> >**W3:**
>
>
> The architecture of our proposed method, AROS, is detailed in the implementation section of the manuscript (please see line 506). In response to the reviewer’s concern, we provide additional details here and have revised the manuscript accordingly to highlight  this information.
>
> AROS consists of three main components: a feature extractor, a Neural Ordinary Differential Equations (NODE) layer, and an orthogonal layer. Specifically, we employ a WideResNet-70-16 model as the backbone feature extractor. Following this, a NODE layer, implemented as a multi-layer perceptron (MLP) with dimensions 128 X 128, models the system dynamics. Finally, an orthogonal layer, represented by an MLP with dimensions \28 X 2, is utilized.
>
> We also conducted an ablation study (Table 11a) to evaluate the impact of different architectures used as the backbone feature extractor, including PreActResNet, ResNet18, ResNet50, and ViT-B-16, while keeping the other components fixed. The results demonstrate the consistent superior performance of AROS and its robustness across various backbone architectures. We hope this clarification and the revisions made to the manuscript adequately address this concern.
>
>
> >**W4:**
>
> We evaluated the performance of AROS using different architectures as encoders, as shown in Table 10a, including Transformer-based models such as ViT-B/16. Additionally, we compared AROS against various methods using the default backbones reported in their respective studies (see Table 1). These methods span a range of architectures, including CNN-based models (e.g., the CSI method) and the CATEX method, which primarily employs CLIP-B/16 in its experiments. Notably, CLIP-B/16 integrates a ViT-B/16 Transformer as the image encoder and a masked self-attention Transformer as the text encoder.
>
>
> To address your concern more clearly, we conducted new experiments comparing the performance of AROS against ViT-MSP (Fort et al. , Exploring ,NeurIPS 2021), a strong detector model, and present the results here. Notably, while ViT-MSP uses ViT as its backbone and achieves high performance on clean data, it still lacks adversarial robustness against perturbations.
>
>
>
>
> | **Method** | **CIFAR-10**  |      |     | **CIFAR-100** |     |     | **ImageNet-1k** |     |     |
> |-|-|-|-|-|-|-|-|-|-|
> |            | CIFAR-100     | SVHN | LSUN | CIFAR-10      | SVHN| LSUN| Texture         | iNaturalist | LSUN |
> | **AROS**   | 88.2/80.1     | 93.0/86.4 | 90.6/82.4 | 74.3/67.0 | 81.5/70.6 | 74.3/68.1 | 78.3/69.2 | 84.6/75.3 | 79.4/69.0 |
> | **ViTMSP** | 97.3/0.8      | 99.5/1.5  | 98.4/1.1  | 94.8/4.1  | 96.2/0.4  | 91.6/0.0  | 93.4/2.8  | 98.5/0.7  | 93.1/0.0  |
>
> Comparison of AROS and ViTMSP under clean and $\text{PGD}^{1000}(l_{\infty})$ evaluation}, measured by AUROC (\%). The table cells denote results in the Clean/$\text{PGD}^{1000}$ format. The perturbation budget $\epsilon$ is set to $\frac{8}{255}$ for low-resolution datasets and $\frac{4}{255}$ for high-resolution datasets.
>
>
> >**W5:**
>
> All adversarial attacks against OOD detection models in our study are **targeted**, as detailed in the Preliminaries and Appendix A4 (please see lines 114–123 and 1647-1670) .
>
> Specifically, all existing OOD detection methods involve defining an indicator, referred to as the OOD score, which is used to distinguish OOD samples from ID samples by assigning likelihoods.  Ideally, an OOD detector assigns scores to test inputs such that the resulting score distributions for OOD and ID samples do not overlap—OOD samples receive higher scores compared to ID samples.
>
> In our evaluation, we employed attacks designed to manipulate the OOD score directly, either maximizing or minimizing it depending on the sample's label (ID or OOD). These fully end-to-end attacks specifically target the OOD score to achieve misclassification:
>
> * For OOD samples: The attack minimizes the OOD score, causing the detector to incorrectly classify OOD samples as ID (false negatives).
>
> * For ID samples: The attack maximizes the OOD score, leading the detector to incorrectly classify ID samples as OOD (false positives).
>
> By directly altering the OOD score, these attacks effectively act as targeted adversarial attacks in the OOD detection context. The adversary's goal is to push samples across the decision boundary between ID and OOD classifications, inducing the most detrimental type of misprediction for the detection method.

---

> ### Author Response · Authors · 2024-11-22
>
> >**W6-a:**
>
> **Hyperparameter $\beta$:**
>
>
> To craft fake samples, we fit a GMM to the embedding space of ID samples. The objective is to sample from the GMM such that the likelihood of the samples is low, ensuring they do not belong to the ID distribution and are located near its boundaries. This approach generates near-OOD samples, which are valuable for providing insights into the distribution manifold and improving the detector's performance. To achieve this, we introduced the following equation in the manuscript:
>
> $$
> \frac{1}{(2\pi)^{d/2} |\hat{\Sigma}_j|^{1/2}} \exp\left( -\frac{1}{2} (r - \hat{\mu}_j)^{T} \hat{\Sigma}_j^{-1} (r - \hat{\mu}_j) \right) < \beta,
> $$
>
> where
>
> * $r$: The embedding vector of a sample in the feature space.
>
>  * $d$: The dimensionality of the embedding vector $r$.
>
>   *   $\hat{\mu}_j$: The mean vector of the $j$-th class in the embedding space.
>
>    * $\hat{\Sigma}_j$: The covariance matrix of the $j$-th class in the embedding space.
>
>   *   $|\hat{\Sigma}_j|$: The determinant of the covariance matrix for the $j$-th class.
>
>    *  $\hat{\Sigma}_j^{-1}$: The inverse of the covariance matrix for the $j$-th class.
>
>    * $\beta$: A small threshold value indicating low-likelihood regions of the ID distribution.
>
> In practice, we use an encoder as a feature extractor to obtain the embeddings of ID samples. A GMM is then fitted to these embeddings, and the likelihood of each training sample under the GMM is computed. Next, we sort the embeddings based on their likelihoods and use the $\beta$-th minimum likelihood of the training samples as a threshold. We then randomly sample from the GMM and compare the likelihood of the sampled data against this threshold. If the likelihood is lower than the threshold, the sampled data is retained as fake OOD; otherwise, it is discarded.
>
> For example, setting $\beta = 0.1$ means that the crafted fake OOD samples will have a likelihood of belonging to the ID distribution that is lower than 90\% of the ID samples. An ablation study provided in our manuscript explores the effects of different $\beta$ values.
>
> To address reviewer concerns and provide further insights, we have expanded this study and included the results here:
>
>
> | Hyperparameter | **CIFAR-10**        |           |           | **CIFAR-100**       |           |           | **ImageNet-1k**     |           |           |
> |-|-|-|-|-|-|-|-|-|-|
> |                | **CIFAR-100**      | **SVHN**  | **LSUN**  | **CIFAR-10**        | **SVHN**  | **LSUN**  | **Texture**         | **iNaturalist** | **LSUN** |
> |  $\beta$= 0.001 *(default)* | 88.2/80.1         | 93.0/86.4 | 90.6/82.4 | 74.3/67.0         | 81.5/70.6 | 74.3/68.1 | 78.3/69.2          | 84.6/75.3       | 79.4/69.0 |
> |  $\beta$ = 0.01            | 87.3/80.2         | 92.2/85.3 | 89.4/80.7 | 74.7/65.8         | 80.2/68.9 | 73.1/66.6 | 78.1/69.5          | 83.0/74.2       | 79.0/68.5 |
> |  $\beta$ = 0.025            | 86.3/79.6         | 93.8/85.0 | 89.3/83.1 | 72.7/67.5         | 81.7/70.8 | 74.3/68.0 | 76.7/67.7          | 84.7/75.3       | 79.8/69.7 |
> |  $\beta$ = 0.05             | 86.5/80.1         | 92.5/86.1 | 90.8/82.7 | 74.5/66.4         | 81.7/69.6 | 72.8/68.3 | 77.9/67.4          | 82.7/75.1       | 77.9/68.5 |
> | $\beta$ = 0.075            | 88.4/78.4         | 92.7/85.2 | 89.1/82.3 | 72.4/65.7         | 81.5/69.4 | 72.5/67.9 | 78.6/69.7          | 84.8/73.5       | 78.8/69.3 |
> | $\beta$ = 0.1            | 86.6/79.5         | 92.0/85.4 | 90.9/80.9 | 72.3/65.8         | 80.3/69.4 | 73.3/66.7 | 77.8/67.5          | 84.1/74.9       | 78.2/67.8 |
> |  $\beta$ = 0.25            | 79.4/67.5         | 89.0/79.5 | 78.7/69.4 | 67.7/54.3         | 69.5/60.3 | 70.1/62.7 | 62.6/65.0          | 65.2/57.9       | 70.9/62.0 |
> |  $\beta$ = 0.5              | 65.4/53.0         | 77.4/62.3 | 64.7/54.6 | 57.5/40.9         | 52.4/49.2 | 49.1/41.9 | 55.8/48.9          | 67.4/59.7       | 57.9/53.8 |
>
>
> As the results show, AROS demonstrates consistent performance for small values of $\beta$. Notably, setting $\beta = 0.5$ corresponds to randomly sampling from the ID distribution instead of focusing on low-likelihood regions, which, as the results indicate, leads to poor performance. Based on our findings, we recommend choosing $\beta$ values in the range $[0.0, 0.1]$ to achieve the desired objective.

---

> ### Author Response · Authors · 2024-11-22
>
> >**W6-b:**
>
> **Hyperparameters $\gamma_1$, $\gamma_2$, $\gamma_3$:**
>
> In our proposed method, we introduce the empirical loss function $\mathcal{L}_{\text{SL}}$ as follows:
>
> $$
> L_{SL} = \min_{\phi, \eta} \frac{1}{\left| X_{\text{train}} \right|} \Bigg(\ell_{\text{CE}}(  B_{\eta} (h_{\phi}(X_{\text{train}}), y)) + \gamma_1 \| \| h_{\phi}(X_{\text{train}})\|\|_2
> $$
>
> $$
> \quad \quad \quad \quad \quad \quad + \gamma_2 \text{exp} \bigg(-\sum_{i=1}^n [\nabla h_{\phi}(X_{\text{train}})]_{ii} \bigg)
> $$
>
> $$
> \quad \quad \quad \quad \quad \quad \quad \quad \quad \quad \quad \quad + \gamma_3 \text{exp} \bigg( \sum_{i=1}^n (-|[\nabla h_{\phi}(X_{\text{train}})]_{ii}|
> $$
>
> $$
> \quad \quad \quad \quad \quad \quad \quad \quad \quad \quad \quad \quad + \sum_{j \neq i} |[\nabla h_{\phi}(X_{\text{train}})]_{ij}| ) \bigg) \Bigg)
> $$
>
> ### Here:
> - $X_{\text{train}}$ is the set of training embeddings.
> - $y$ are the corresponding labels.
> - $h_{\phi}$ is the NODE function parameterized by $\phi$.
> - $B_{\eta}$ is the orthogonal binary layer parameterized by $\eta$.
> - $\ell_{\text{CE}}$ denotes the cross-entropy loss.
> - $\nabla h_{\phi}(X_{\text{train}})$ is the Jacobian matrix of $h_{\phi}$ with respect to $X_{\text{train}}$.
>
> ---
>
> **Roles of the Hyperparameters**
>
> ### 1. $\gamma_1$
> Controls the regularization term $\| h_{\phi}(X_{\text{train}})\|_2$, which encourages the system's state to be near the equilibrium point. This term helps mitigate the effect of perturbations by ensuring that trajectories remain close to the equilibrium.
>
> ### 2. $\gamma_2$ and $\gamma_3$
> These hyperparameters weight the exponential terms designed to enforce Lyapunov stability conditions by ensuring the Jacobian matrix satisfies strict diagonal dominance (as per Theorem 3 in the paper).
>
> - **$\gamma_2$** weights the term $\exp\left( -\sum_{i=1}^n [\nabla h_{\phi}(X_{\text{train}})]_{ii} \right)$, encouraging the diagonal entries of the Jacobian to be negative with large magnitudes.
>
> - **$\gamma_3$** weights the term $\text{exp} \bigg( \sum_{i=1}^n (-|[\nabla h_{\phi}(X_{\text{train}})]_{ii}| \bigg)$
>
>   promoting strict diagonal dominance by penalizing large off-diagonal entries relative to the diagonal entries.
>
> ---
>
> **Reasoning Behind Default Values**
>
> - **$\gamma_1 = 1$**: Balances the influence of the regularization term with the primary cross-entropy loss. This ensures sufficient regularization without overwhelming the classification objective.
> - **$\gamma_2 = \gamma_3 = 0.05$**: Preliminary experiments indicate these values effectively enforce stability without causing numerical issues or hindering optimization. We choose $\gamma_2 = \gamma_3 = 0.05$ Inspired by related works [1-10] that incorporate stability terms into the training process.
>
> ---
>
> **Why $\gamma_2$ and $\gamma_3$ Are Equal**
>
> 1. **Balanced Contribution**: Both regularization terms serve complementary purposes in enforcing the stability conditions. Equal weighting ensures neither term dominates, maintaining a balanced emphasis on both negative diagonal dominance and strict diagonal dominance.
> 2. **Simplified Hyperparameter Tuning**: Setting $\gamma_2$ and $\gamma_3$ equal reduces the hyperparameter search space, simplifying the tuning process without sacrificing performance.
>
>
> ---
>
> **Impact of Hyperparameters on Performance**
>
> **Effect of $\gamma_1$**
>
>
> - **Low Values ($\gamma_1 < 1$)**: Reduces the emphasis on keeping the state near the equilibrium, making the system more susceptible to perturbations.
> - **High Values ($\gamma_1 > 1$)**: Overly constrains the state to the equilibrium point, potentially limiting the model's capacity to learn discriminative features.
> - **Chosen Value ($\gamma_1 = 1$)**: Offers a good balance, ensuring sufficient regularization without compromising learning.
>
> ---
>
> **Effect of $\gamma_2$ and $\gamma_3$**
>
>
>
> - **Low Values ($< 0.05$)**: Diminish the impact of the stability constraints, reducing robustness.
> - **High Values ($> 0.05$)**: Overemphasize the stability terms, potentially hindering the optimization of the primary classification loss.
> - **Equal Values ($\gamma_2 = \gamma_3$)**: Ensure balanced enforcement of both stability conditions, leading to optimal performance.
>
>
> To address the reviewers' concerns, we conducted extensive ablation studies, keeping all components fixed while varying the values of $\gamma_1$, $\gamma_2$, and $\gamma_3$. These experiments demonstrate that AROS consistently performs well across a broad range of hyperparameter values, including extreme cases (e.g., $\gamma_1 = 2$). The results of this analysis are presented in Table 13a of our paper and are also included below. Notably, this table was part of our submitted manuscript, and we have now extended it with additional experiments to further address any remaining concerns. These results confirm the robust performance of AROS under varying configurations of $\gamma_1$, $\gamma_2$, and $\gamma_3$.

---

> ### Author Response · Authors · 2024-11-22
>
> >**W6-c:**
>
> **Hyperparameters $\gamma_1$, $\gamma_2$, $\gamma_3$:**
>
>
> | Hyperparameter                     | **CIFAR-10**        |           |           | **CIFAR-100**       |           |           | **ImageNet-1k**     |           |           |
> |-|-|-|-|-|-|-|-|-|-|
> |                                    | **CIFAR-100**      | **SVHN**  | **LSUN**  | **CIFAR-10**        | **SVHN**  | **LSUN**  | **Texture**         | **iNaturalist** | **LSUN** |
> | **$\gamma_1=1, \gamma_2=\gamma_3=0.05$ (default)**     | 88.2/80.1          | 93.0/86.4 | 90.6/82.4 | 74.3/67.0           | 81.5/70.6 | 74.3/68.1 | 78.3/69.2          | 84.6/75.3       | 79.4/69.0 |
> | **$\gamma_1=1, \gamma_2=\gamma_3=0.025$**              | 86.4/78.2          | 90.4/86.3 | 88.6/80.4 | 74.9/64.1           | 78.8/68.7 | 72.8/68.3 | 79.2/69.7          | 83.3/76.0       | 78.8/67.5 |
> | **$\gamma_1=1, \gamma_2=\gamma_3=0.075$**              | 87.5/80.3          | 92.1/87.2 | 88.9/81.6 | 72.7/67.7           | 80.8/68.7 | 73.2/68.8 | 76.4/69.9          | 83.4/73.4       | 80.2/69.9 |
> | **$\gamma_1=1, \gamma_2=\gamma_3=0.1$**                | 86.9/78.6          | 93.6/82.1 | 85.8/78.2 | 70.0/65.8           | 80.6/66.2 | 72.6/67.3 | 79.3/69.6          | 82.8/74.0       | 78.9/67.2 |
> | **$\gamma_1=1, \gamma_2=\gamma_3=0.25$**               | 86.2/78.7          | 94.3/82.3 | 87.1/77.9 | 70.2/65.2           | 81.3/65.8 | 73.9/66.7 | 78.6/70.1          | 82.8/74.1       | 78.9/68.2 |
> | **$\gamma_1=1, \gamma_2=\gamma_3=0.5$**                | 85.2/76.9          | 93.6/81.5 | 85.0/76.0 | 69.7/64.2           | 78.1/64.4 | 72.6/65.2 | 77.0/66.7          | 83.6/74.4       | 78.2/67.3 |
> | **$\gamma_1=\gamma_2=\gamma_3=1$**                     | 84.4/75.3          | 90.5/80.5 | 81.5/73.4 | 69.8/62.2           | 77.5/65.3 | 70.1/64.4 | 73.5/67.3          | 82.7/70.9       | 76.7/65.4 |
> | **$\gamma_1=\gamma_2=\gamma_3=0.25$**                  | 85.4/77.4          | 94.6/82.9 | 86.3/76.9 | 69.9/64.4           | 79.9/65.1 | 74.4/65.3 | 77.2/68.6          | 85.4/76.0       | 79.2/66.8 |
> | **$\gamma_1=0.5, \gamma_2=\gamma_3=0.05$**             | 86.0/78.6          | 92.7/82.5 | 86.3/77.7 | 69.7/66.1           | 81.2/65.5 | 70.9/67.1 | 78.8/68.6          | 83.0/73.7       | 78.8/67.1 |
> | **$\gamma_1=1, \gamma_2=0.1, \gamma_3=0.05$**          | 84.0/76.3          | 91.5/81.5 | 81.7/76.4 | 70.1/63.4           | 76.2/63.8 | 69.8/67.7 | 77.4/69.5          | 83.2/71.0       | 78.5/64.8 |
> | **$\gamma_1=1, \gamma_2=0.05, \gamma_3=0.1$**          | 83.5/77.5          | 90.2/78.0 | 81.4/75.1 | 67.2/63.5           | 76.7/66.2 | 67.5/60.4 | 74.3/70.0          | 83.6/74.1       | 74.2/61.7 |
> | **$\gamma_1=2, \gamma_2=0.5, \gamma_3=0.5$**           | 80.4/74.0          | 93.5/80.7 | 81.8/70.9 | 65.2/66.0           | 77.3/60.0 | 65.0/60.1 | 78.4/63.3          | 80.3/70.6       | 71.6/59.5 |
>
>
> ---
>
> ---
>
>
>
> [1] Zonghan Yang, Tianyu Pang, and Yang Liu. A closer look at the adversarial robustness of deep equilibrium models. Advances in Neural Information Processing Systems, 35:10448–10461, 2022.
>
> [2] Ivan Dario Jimenez Rodriguez, Aaron Ames, and Yisong Yue. Lyanet: A lyapunov framework for training neural odes. In International conference on machine learning, pages 18687–18703. PMLR, 2022.
>
> [3] Xiyuan Li, Zou Xin, and Weiwei Liu. Defending against adversarial attacks via neural dynamic system. Advances in Neural Information Processing Systems, 35:6372–6383, 2022.
>
> [4] Qiyu Kang, Yang Song, Qinxu Ding, and Wee Peng Tay. Stable neural ode with lyapunov-stable equilibrium points for defending against adversarial attacks. Advances in Neural Information Processing Systems, 34:14925–14937, 2021.
>
> [5] Fabio Carrara, Roberto Caldelli, Fabrizio Falchi, and Giuseppe Amato. On the robustness to adversarial examples of neural ode image classifiers. In 2019 IEEE International Workshop on Information Forensics and Security (WIFS), pages 1–6. IEEE, 2019.
>
> [6] Jan Svoboda, Jonathan Masci, Federico Monti, Michael Bronstein, and Leonidas Guibas. Peernts: Exploiting peer wisdom against adversarial attacks. In International Conference on Learning Representations, 2019.
>
> [7] Arash Rahnama, Andre T Nguyen, and Edward Raff. Robust design of deep neural networks against adversarial attacks based on lyapunov theory. In Proceedings of the IEEE/CVF Conference on Computer Vision and Pattern Recognition, pages 8178–8187, 2020.
>
> [8] Mingjie Li, Lingshen He, and Zhouchen Lin. Implicit euler skip connections: Enhancing adversarial robustness via numerical stability. In International Conference on Machine Learning, pages 5874–5883. PMLR, 2020.
>
>
> [9] Sergey Dashkovskiy, Oleksiy Kapustyan, and Vitalii Slynko. Robust stability of a nonlinear ode-pde system. SIAM Journal on Control and Optimization, 61(3):1760–1777, 2023.
>
> [10] Mustafa Zeqiri, Mark Niklas Müller, Marc Fischer, and Martin Vechev. Efficient certified training and robustness verification of neural odes. arXiv preprint arXiv:2303.05246, 2023.

---

> ### Author Response · Authors · 2024-11-27
> **Reviewer feedback requested**
>
> Dear Reviewer e2wJ,
>
> Thank you for your insightful review of our manuscript. We have addressed your feedback and submitted a revised version of the paper. We would greatly value it if you could review our updates and share any additional comments or concerns. Thank you once again for your valuable input!
>
> Yours sincerely,
> The Authors

---

> > ### Comment · Reviewer_e2wJ · 2024-12-02
> >
> > Thank you for your detailed response. Most of my concerns have been resolved, and I will keep my score unchanged.

---

> > > ### Author Response · Authors · 2024-12-02
> > >
> > > Dear e2wJ, thanks for reading our rebuttal (we know it's long, but we aimed to fully satisfy your concerns). Therefore, is there a specific concern we missed that could help you improve the score of our work? Thank you!

---

### Official Review · Reviewer_dAkz · 2024-11-04

**Soundness:** 4
**Presentation:** 3
**Contribution:** 3
**Rating:** 8
**Confidence:** 2

**Summary:**

This paper proposes a new out-of-distribution detector that is robust against adversarial attacks. Specifically, the authors point out the problems of adversarial training with auxiliary datasets and propose a novel approach based on neural ordinary differential equation (ODE) and Lyapunov stability. The authors force the neural ODE model’s dynamical system to have stable equilibrium points on both in-distribution (ID) and out-of-distribution (OOD) samples. The authors also present a theoretical background that supports the proposed method. With experiments, the authors present the performance of the proposed method and some ablation studies.

**Strengths:**

1. The proposed method's theoretical background is solid, and the main trick (i.e., the Levy-Desplanques Theorem) seems useful for other neural ODE models.
2. With an orthogonal layer, the authors effectively overcame practical problems, such as exploding gradients and vanishing gradients.
3. The experiments are well done. The authors used very strong attacks and many baseline models to compare. The proposed method's performance is really impressive.

**Weaknesses:**

1. The proposed method generates fake OOD data in the embedding space, which might expose other vulnerabilities in ID embedding extraction. For example, the adversary can try poisoning the training ID data by injecting OOD samples so that the crafted OOD data does not include those injected OOD samples.
2. If I understand correctly, the fake OOD embedding is sampled from the low-likelihood space. This makes sense because OOD samples will be embedded in the low-likelihood part. However, this could be an ineffective method to generate embeddings of “realistic” OOD samples because even random noise would be mapped to the low-likelihood part.

**Questions:**

1. I’m curious about the sampling methods. Do you have some justification that the low-likelihood part would represent the embeddings of the realistic OOD samples, or do you just not care about the realisticness of the embeddings?

---

> ### Author Response · Authors · 2024-11-22
>
> Dear Reviewer dAkz,
>
> Thank you for your positive feedback! We have provided detailed responses to your questions and comments below:
>
> >**W1:**
>
>
> There are two primary types of malicious attempts to deceive deep neural networks. The first involves data poisoning attacks, such as backdoor or Trojan attacks, which represent a significant category of threats. These attacks occur when adversaries have the ability to manipulate the training data or the training process, fundamentally altering the behaviour of the model.
> The second type pertains to adversarial robustness against inference-time attacks, where adversaries attempt to deceive the model by manipulating inputs during the inference phase. In this scenario, the adversary does not have access to or influence over the training data or process. The primary concern here is defending against attacks that occur when the model is deployed in real-world scenarios without compromising the integrity of the training pipeline.
> While we appreciate and understand the reviewer’s concern regarding potential vulnerabilities related to training data poisoning, we emphasise that our threat model specifically addresses scenarios where the training data is secure and free from malicious modifications. It is important to note that these two types of attacks are generally addressed in separate areas of the literature.
> Our primary goal in this study was to enhance the adversarial robustness of OOD detectors against inference-time attacks, which is critical for deploying models in real-world environments. Addressing training-time attacks is indeed an important direction for future work, and we acknowledge its significance. However, it falls outside the scope of our current study, which assumes a clean and secure training pipeline.

---

> ### Author Response · Authors · 2024-11-22
>
> >**W2&Q1:**
>
> It has been shown that utilizing auxiliary realistic OOD samples is generally effective for improving OOD detection performance. However, this strategy comes with several challenges, as discussed in our manuscript (see lines 79–89 and 190-200).
> First, in certain scenarios, access to an external realistic OOD dataset may not be feasible, and acquiring such data can be challenging. Even when a suitable dataset is available, it must be processed to remove ID concepts to prevent the detector from being misled. This preprocessing step is both time-consuming and computationally expensive. Additionally, studies highlight a potential risk of bias being introduced into the detector when trained on specific auxiliary datasets. Such reliance on a particular realistic dataset may undermine the detector's ability to generalize effectively to diverse OOD samples. These issues become even more pronounced in adversarial training setups, where the complexity of the required data is significantly higher. Motivated by these challenges, this study proposes an alternative strategy that does not depend on the availability of an auxiliary OOD dataset. Notably, our approach is flexible and can incorporate auxiliary OOD datasets as additional information if they are available.
> To validate this, we conducted an experiment assuming access to a realistic OOD dataset (i.e., Food-101). In this scenario, we computed embeddings of the real OOD samples and used them alongside crafted fake OOD embeddings during training. The results, presented in Table 4 (Setup A), demonstrate improved performance compared to using fake OOD embeddings alone.
> Furthermore, related studies have shown that in adversarial training, using samples near the decision boundary of the distribution improves robustness by encouraging compact representations. This boundary modeling is critical for enhancing the model's robustness, especially against adversarial attacks that exploit vulnerabilities near the decision boundary.
> In light of this, our approach shifts focus from generating “realistic” OOD data to estimating low-likelihood regions of the in-distribution. We generate fake "near" OOD data that is close to the ID boundary, which is particularly beneficial for adversarial robustness.
>  For better intuition regarding usefulness of auxiliary near OOD samples here we will provide a simple example that highlights the effectiveness of near-distribution crafted OOD samples in the adversarial setup.  We assume that the feature space is one-dimensional, i.e. $\mathbb{R}$, and the ID class is sampled according to a uniform distribution $U(0, a - \epsilon)$, with $a > 0$, and $\epsilon < a $. We assume that the OOD class is separable with a safety margin of $2\epsilon$ from the ID class to allow a perfectly accurate OOD detector under input perturbations of at most $\epsilon$ at inference. For instance, we let $U(a+\epsilon, b)$ be the feature distribution under the OOD class. The goal is to leverage crafted fake OOD samples to find a robust OOD detector under the $\ell_2$ bounded perturbations of norm $\epsilon$. We assume that the crafted OOD samples data distribution is not perfectly aligned with the anomaly data, e.g. crafted OOD samples comes from $U(a+r, c)$, with $r \geq \epsilon$. It is evident that the optimal robust decision boundary under infinite training samples that separates the ID and crafted OOD samples would be a threshold $k$ satisfying $a \leq k \leq a + r - \epsilon$. The test adversarial error rate to separate ID and OOD classes is $\frac{1}{2}. \mathbb{I}(k \geq a+\epsilon).\left(\frac{k-a-\epsilon}{b-a-\epsilon}  + \frac{\min(k+\epsilon, b)-k}{b - a - \epsilon} \right) + \frac{1}{2}.\mathbb{I}(a < k < a + \epsilon) \frac{\min(k+\epsilon, b) - a - \epsilon}{b - a - \epsilon} $, assuming that the classes are equally probable a prior. It is obvious the adversarial error rate would be zero for $k = a$. But otherwise, if $k \geq a + \epsilon$ the classifier incurs classification error in two cases; in intervals $(a+\epsilon, k)$ (even without any attack), and $(k, \min(k+\epsilon, b))$ in which a perturbation of $-\epsilon$ would cause  classification error. Also if $a < k < a + \epsilon$, classification error only happens at $(a+\epsilon, \min(k+\epsilon, b))$. Now, for the crafted OOD samples to be near-distribution, $r \rightarrow \epsilon$, which forces $k$ to be $a$ in the training, and makes the test adversarial error zero. Otherwise, the adversarial error is proportional to $k$, for $k$ being distant from $b$. Therefore, in the worst case, if $k = a + r - \epsilon$, we get an adversarial error proportional to $r$. As a result, minimizing $r$, which makes the crafted OOD samples near-distribution, would be an effective strategy in making the adversarial error small. We added this new example to our updated manuscript PDF along with a new figure for further clarification. We kindly ask the reviewer to see A3.5 section of the Appendix.

---

> ### Author Response · Authors · 2024-11-27
>
> Dear Reviewer dAkz,
>
> We sincerely appreciate the thoughtful feedback you’ve offered on our manuscript. We have carefully reviewed each of your comments and have made efforts to address them thoroughly. We kindly ask you to review our responses and share any additional thoughts you may have on the paper or our rebuttal. We would be more than happy to accept all your criticisms and incorporate them into the paper.
>
> Sincerely, The Authors

---

### Comment · Area_Chair_JHUo · 2024-11-22

Dear Authors and Reviewers,

The discussion phase has passed 10 days. If you want to discuss this with each other, please post your thoughts by adding official comments.

Thanks for your efforts and contributions to ICLR 2025.

Best regards,

Your Area Chair

---

### Author Response · Authors · 2024-11-27
**Summary of Revisions**

Dear AC & Reviewers,

Firstly, we would like to thank the early overall positive reviews and scores. Notably,
- [dAkz](https://openreview.net/forum?id=GrDne4055L&noteId=wKnbtHnr5t) notes “The experiments are well done. The authors used very strong attacks and many baseline models to compare. The proposed method's performance is really impressive.”;
- [e2wJ](https://openreview.net/forum?id=GrDne4055L&noteId=VZ6pRvJazY) notes “The paper proposes an OOD detection method based on Lyapunov stability and NODE, representing an innovative attempt to integrate control theory with OOD detection to improve model robustness under adversarial perturbations.”
- [w7tq](https://openreview.net/forum?id=GrDne4055L&noteId=AO1KPtxMSI) notes “Experimental results demonstrate that AROS outperforms other OOD detection methods across multiple benchmark datasets (e.g., CIFAR-10, CIFAR-100, and ImageNet-1k), especially under strong adversarial attacks such as PGD, AutoAttack, and Adaptive AutoAttack”
- [x9hq](https://openreview.net/forum?id=GrDne4055L&noteId=UwLwydwwGy) highlights our innovations and strong results: “The integration of Lyapunov stability within NODEs for OOD detection is novel and well-motivated. The application of stability principles to mitigate adversarial perturbations shows theoretical grounding.”

**Revisions TL;DR: we have performed more experiments and updated our PDF to incorporate feedback from the reviewers.** In summary, we included:
- new analysis of hyperparameters,
- new ablation studies,
- clarifications on our synthetic OOD data including new figures and visualizations,
- clarified any remaining questions on our theoretical foundations.

We kindly invite the reviewers to review our responses and edits, and hope these new improvements can positively affect their ratings. In more detail, here is a summary of changes:

**Hyperparameters and Ablation Studies**: We have included new extensive ablation studies that explore the impact of various hyperparameters in our pipeline, which includes $\gamma_1$, $\gamma_2$, $\gamma_3$ related to the weights in our loss function, and $\beta$, which intuitively acts as a threshold in our pipeline. We provided the rationale behind choosing these values and proposed ranges for them, along with comprehensive experimental results showing the model's robustness across a wide range of values. **We demonstrate the model's consistent performance despite changes in hyperparameters**.

- Please refer to Appendix Section A3.3, Figures 4, 5, and 6 and the updated Tables 12a and 13a

**Generation and Representativeness of Synthetic OOD Data**: We clarified our data-centric strategy, which involves generating synthetic OOD embeddings from low-likelihood regions near the ID boundary, emphasizing that near-distribution samples are particularly beneficial for enhancing adversarial robustness. We provided theoretical justifications, including illustrative examples, to demonstrate the effectiveness of this approach. We discussed that our goal was to design a pipeline that is computationally cost-efficient and avoids bias instead of relying on realistic datasets, enabling it to work even in scenarios without such information. However, we show that if such sources of information are available, AROS can incorporate them, highlighting our method's adaptability. Moreover, we conducted new experiments incorporating real OOD datasets alongside our synthetic embeddings, which showed improved performance. **Ablation studies comparing different OOD data synthesis strategies were also added to validate our method empirically.**

- Please see Appendix, new Section A3.5, Table 15a

**Clarification of Theoretical Foundations and Mathematical Justifications**: We have expanded the background section to include detailed explanations of the control theory concepts utilized in our work. Specifically, we provided comprehensive derivations and proofs of the Lyapunov stability theorems and their application within the NODE framework. **An additional section in the Appendix now offers proofs of key theorems, practical details about our loss function, and theoretical guarantees of stability in high-dimensional embeddings.** These additions aim to make the theoretical aspects more approachable and enhance understanding for all readers.

- Please see Appendix, Figure 4 and Section A7

**Orthogonal Binary Layer**: We have revised the manuscript to provide a clearer explanation of the orthogonal binary layer. We decomposed its mathematical formulation, clarified how it enhances separation between in-distribution (ID) and out-of-distribution (OOD) samples, and annotated its function within the model architecture. **New experiments and visualizations have been added to demonstrate empirically how this layer maintains separation, especially under adversarial scenarios, and to analyze its sensitivity to different perturbation levels.**

- Please see Appendix Figure 7 and Section A3.6

---

### Comment · Area_Chair_JHUo · 2024-11-30
**Need engagement in discussion**

Dear reviewers,

Many thanks for your efforts in reviewing this paper. As the discussion deadline is fast approaching, could you check if the authors' responses and the revision address your major concerns?

Your engagement for this paper is important and necessary.

Best,

Your Area Chair

---

### Meta-Review · Area_Chair_JHUo · 2024-12-21

**Metareview:**

Out-of-distribution detection aims to improve the systems' robustness by letting the system say no to something unknown to it. However, the safety of OOD detection algorithm is rarely discussed or studied. This paper focuses on this important problem and makes a further step towards robust OOD detection algorithms. The novelty and contribution of this paper are enough to this venue, agreed by all reviewers. However, relevant comments must be merged into the final version of this paper.

**Additional Comments On Reviewer Discussion:**

Concerns from three reviewers are addressed well. One reviewer does not engage to discussion, but, after checking the responses, the concerns are addressed as well.

---

### Decision · Program_Chairs · 2025-01-22

Accept (Poster)